



**Reviews and syntheses: Physical and biogeochemical processes associated with upwelling in the**
**Indian Ocean**
Puthenveettil Narayana Menon Vinayachandran[1*], Yukio Masumoto[2], Mike Roberts[3], Jenny Hugget[4],
Issufo Halo[5], Abhisek Chatterjee[6], Prakash Amol[7], Garuda V. M. Gupta[8], Arvind Singh[9], Arnab
Mukherjee[10], Satya Prakash[6], Lynnath E. Beckley[11], Eric Jorden Raes[12], Raleigh Hood[13]
[1]Centre for Atmospheric and Oceanic Sciences, Indian Institute of Science, Bengaluru, 560012, India
[2] Graduate School of Science, University of Tokyo, Tokyo, Japan
[3] Nelson Mandela University, Port Elizabeth, South Africa
[4]Oceans and Coasts Research, Department of Environment, Forestry and Fisheries, Private Bag X4390, Cape Town 8000,
South Africa
[5]Department of Conservation and Marine Sciences, Cape Peninsula University of Technology, PO Box 652, Cape Town
8000, South Africa
[6]Indian National Centre for Indian Ocean Services, Ministry of Earth Sciences, Hyderabad, India
[7]CSIR-National Institute of Oceanography, Regional Centre, Visakhapatnam, 530017, India
[8]Centre for Marine Living Resources and Ecology, Ministry of Earth Sciences, Kochi, India
[9]Physical Research Laboratory, Ahmedabad, 380009, India
[10] National Centre for Polar and Ocean Research, Ministry of Earth Sciences, Goa, India
[11] Environmental and Conservation Sciences, Murdoch University, Perth, Western Australia 6150, Australia
[12]CSIRO Oceans and Atmosphere, GPO Box 1538, Hobart, TAS, 7001 Australia
[13] University of Maryland Center for Environmental Science, Cambridge, MD, USA
*Correspondence to*: P. N. Vinayachandran (vinay@iisc.ac.in)



**Abstract.**
The Indian Ocean presents two distinct climate regimes. The North Indian Ocean is dominated by the monsoons, whereas
the seasonal variation is less pronounced in the south.   The prevailing wind pattern produces upwelling along different parts
of the coast in both hemispheres during different times of the year. Additionally, dynamical processes and eddies either
cause or enhance upwelling. This paper reviews the phenomena of upwelling along the coast of the Indian Ocean extending
from the tip of South Africa to the southern tip of the west coast of Australia. Observed features, underlying mechanisms,
and the impact of upwelling on the ecosystem are presented.
In the Agulhas Current region, cyclonic eddies associated with the Natal pulses drive slope upwelling and enhances
chlorophyll concentration along the continental margin. The Durban break-away eddy spun-up by the Agulhas upwells cold
nutrient-rich water. Besides, topographically induced upwelling occurs along the inshore edges of Agulhas Current. Wind-
driven coastal upwelling occurs along the South coast of Africa and augments the dynamical upwelling in the Agulhas
Current. Upwelling hotspots along Mozambique are present in the northern and southern sectors of the channel, and they
are ascribed to dynamical effects of ocean circulation in addition to wind forcing. Interaction of mesoscale eddies with the
western boundary, anticyclonic eddy pair interactions, and passage of cyclonic eddies cause upwelling. Upwelling along the
southern coast of Madagascar is caused by Ekman wind-driven mechanism and by eddy generation and inhibited by
Southwest Madagascar Coastal Current. The seasonal upwelling that occurs along the East African coast is primarily driven
by the Northeast monsoon winds and enhanced by topographically induced shelf-breaking and shear instability between the
East African Coastal Current and the island chains. Somali coast presents a strong case for the classical Ekman type of
upwelling. This upwelling can be inhibited by the arrival of deeper thermocline signals generated in the offshore region by
wind stress curl. The upwelling is nearly uniform along the coast of Arabia, it is caused by the alongshore component of the
summer monsoon winds and modulated by the arrival of Rossby waves generated in the offshore region by cyclonic wind
stress curl. Along the west coast of India, upwelling is driven by coastally trapped waves together with the alongshore
component of the southwesterlies. Along the southern tip of India and Sri Lanka, the strong Ekman transport dives
upwelling.  Upwelling is feeble along the east coast of India and occurs during the summer, caused by alongshore winds. In
addition, mesoscale eddies lead to upwelling but the arrival of river water plumes inhibits upwelling along this coast.
Southeasterly winds drive upwelling along the coast of Sumatra and Java during summer. Kelvin wave propagation
originating from the Equatorial Indian Ocean affects the magnitude and extent of Sumatra and Java upwelling. Both ENSO
and IOD events cause large variability of upwelling here. Along the west coast of Australia, southerly winds can dominate
over the Leeuwin Current, causing sporadic upwelling, which is prominent along the southwest, central, and Gascoyne
coasts during summer. The open ocean upwelling in the southern tropical Indian Ocean and within the Sri Lanka Dome is
driven primarily by the wind stress curl but also impacted by Rossby wave propagations.



Upwelling is a key driver in enhancing biological productivity in all sectors of the coast, as indicated by enhanced sea
surface chlorophyll concentrations. Additional knowledge at varying levels has been gained through in situ observations
and model simulations. In the Mozambique channel, upwelling simulates new production, and circulation redistributes the
production generated by upwelling and mesoscale eddies leading to observations of higher ecosystem impact along the
edges of eddies.  Similarly, along the southern Madagascar coast, biological connectivity is influenced by the transport of
phytoplankton from upwelling zones. Along the coast of Kenya, both productivity rates and zooplankton biomass are higher
during the upwelling season. Along the Somali coast, accumulation of upwelled nutrients in the northern part of the coast
leads to spatial inhomogeneity in productivity. On the other hand, productivity is more uniform along the coasts of Yemen
and Oman.  Upwelling along the west coast of India has several biogeochemical implications, including oxygen depletion,
denitrification, and high production of CH4 and dimethyl sulfide. Though feeble, wind-driven upwelling leads to significant
enhancement of phytoplankton in the northwest Bay of Bengal during the summer monsoon. Along the Sumatra and Java
coasts, upwelling affects the phytoplankton composition and assemblages. Dissimilarities in copepod assemblages occur
during the upwelling periods along the west coast of Australia. Phytoplankton abundance characterizes inshore edges of the
slope during upwelling season, and upwelling eddies are associated with abundance in Krill.
The review identifies the northern coast of the Arabian Sea and eastern coasts of the Bay of Bengal as the least observed
sectors. Further, sustained long-term observations with high temporal and spatial resolutions along with high-resolution
modeling efforts are suggested for a proper description of upwelling, its variability, and its relationship to the ecosystem.



## 1 Introduction

**1 Introduction**
Tangential winds that blow parallel to the coast result in the transport of water away from the coast (Ekman, 1905). The
water that is transported from near the coast must be replaced by water from below, usually from a depth range of 100 -
300m. This upward motion of water from below is termed as (coastal) upwelling (Sverdrup, 1937). While the dynamics of
the system primarily concerns the upward flow, the change in properties of the water near the surface is what concerns most
for the ecosystem and biogeochemistry. The water that upwells comes from below the Ekman layer, and therefore it is
cooler, denser, and rich in nutrients. The transport away from the coast is governed by Ekman dynamics, and owing to the
higher density of the upwelled water near the coast, a current that is parallel to the coast gets sets-up. The existence of a
physical boundary, the coast, is a necessary condition for the upwelling to take place. The absence of Coriolis force at the
equator and its change in sign across the equator creates a dynamical boundary at the equator that supports upwelling. Thus,
easterlies drive poleward Ekman transport on both sides of the equator giving rise to equatorial upwelling. Upwelling is
possible in the open ocean as well, even in the absence of a physical or dynamic boundary when the surface winds possess
strong positive vorticity. Cyclonic wind stress curl leads to divergence within the surface layer leading to upward vertical
velocity known as Ekman suction, which is often represented by a 'thermocline dome'. Other mechanisms, such as planetary
waves can cause upwelling, and they will be described later in this paper. Upwelling has great significance in ocean science
owing to its enormous potential to (1) cool the sea surface by several degrees and (2) increase the productivity of near-
surface water by several orders of magnitudes, compared to regions unaffected by upwelling. California, Iberian, Canary,
Humboldt, and Benguela are the well-known eastern boundary current upwelling systems in the world oceans. These
classical eastern boundary upwelling systems are driven by winds blowing towards the equator.
The upwelling process connects the upper wind-driven part of the ocean with the relatively quiescent sub-surface regimes.
Upwelling brings cold, nutrient-rich bottom waters to the surface layer, which significantly supports the primary production
and hence the higher food web. This connection is vital for cycling tracers and nutrients and invigorating marine life across
all states of the food chain. Water upwelled along the eastern boundaries of the major continents is known to harbor some
of the world's largest marine ecosystems (Hutchings et al., 2009, Montecino and Lange, 2009, Checkley and Barth, 2009,
Arístegui et al., 2009). Globally, the upwelling systems occupy less than 2% of the total oceanic area, but they alone
contribute to ~20% of the total fish catch (Pauly and Christensen, 1995). Upwelling links the ocean interior with the surface
where the ocean and atmosphere interact, exchanging heat, water, and gases, and serves as the source for major
biogeochemical and ecological transformations. Though the impact of upwelling is most pronounced regionally, its impact
could affect basin-scale circulation and regional climate.
The Indian Ocean is different from the Atlantic and Pacific due to its unique geographical setting marked by the northern
land boundary located in the tropics itself. The vast landmass situated to the north of the ocean gives rise to the region's
monsoon climate, which is characterized by seasonally reversing winds and copious rainfall during summer. The monsoon



winds (**Figure 1**) are southwesterlies during May-September and Northeasterly during December-February. The transition
from one Monsoon to the other occurs during the spring and autumn months of March - April and October - November,
respectively (Schott and McCreary, 2001). Therefore, the most striking characteristic of the upwelling in the Indian Ocean,
particularly concerning other typical Eastern boundary upwelling systems, is its seasonality.
A review of the coastal currents in the Indian Ocean was carried out by Shetye and Gouveia (1998).  Schott and McCreary
(2001) provide a comprehensive review of the monsoon circulation in the Indian Ocean, and an update of this review has
been given in Schott et al. (2009). Shankar et al. (2002) has presented a detailed description of the monsoon currents and a
synthesis of their dynamics. More recently, Hood et al. (2015) have reviewed the boundary currents in the Indian Ocean and
their impact on biogeochemistry.  The Indian Ocean science has witnessed a surge in activities in the last decade. Several
multidisciplinary research programs that cut across institutional and national boundaries have been deployed towards
developing new data sets and testing hitherto unknown hypotheses. Concurrently, numerical models have evolved into
higher levels of sophistication, resolution, accuracy, and complexity. Motivated by the rapid progress that the Indian Ocean
has witnessed in the last few years, this paper aims to synthesize the knowledge that has accumulated in recent times.
Additionally, the review covers upwelling regions that have not received enough attention in past reviews.  It is expected
that the review would assist in developing future programs in the Indian Ocean coastal oceanography such as those outlined
in the UN Decade of the Oceans.

The alignment of the coastline with respect to the winds offers favorable conditions for upwelling along several parts of the
Indian Ocean boundaries.  The southwesterlies are favourable for upwelling along the western boundary of the North Indian
Ocean, particularly along the coast of Somalia and Oman. As they approach the west coast of India, the southwesterlies turn
towards the equator and blow nearly parallel to the west coast of India, owing to the presence of the Sahyadri (Western
Ghats) mountain ranges.  The summer monsoon winds are also favorable for upwelling along the southern coast of Sri Lanka
and along the east coast of India.  Persistent wind stress curl in the Southern Tropical Indian Ocean (STIO) leads to a very
shallow thermocline (Xie et al., 2002) and makes it one of the strongest upwelling open ocean upwelling regions.  In the
southern hemisphere, upwelling has been observed in the Agulhas Current region, Mozambique channel, in the region of
the East African Current, and along the coast of Java and Australia.  In the section that follows, upwelling along each of the
above regions is described.

**2. Coastal Upwelling Systems**

In this section, each of the coastal upwelling systems in the Indian Ocean are described in detail. We first present an overview
using historic portrayal followed by recent observations; these sections render characteristics of the upwelling and its impact
on physical parameters.  A review of the present status of modeling these upwelling systems is presented next, along with



mechanisms that drive upwelling. The impact of upwelling on the marine ecosystem and biogeochemistry is discussed next,
including those on the fisheries. Progress made during the IIOE-2 (2015 – 2010) era are paid particular attention, major
outstanding issues are listed, and plausible approaches are suggested. While this general content remains the same for all
regions, no effort has been made to wrap subject matter for each of the regions into the same packaging but follow the
progress of science because the advancement of knowledge in each of these regions have progressed differently in terms of
both the time when major progress was made and also in terms of focus on features that are specific to the region.
**2.1 Agulhas Current**
**2.1.1. Characteristics and importance**

The warm, fast-flowing Agulhas Current is the western-most outflow of the Indian Ocean. In the form of a 1000 km-long
western boundary current along the southeastern side of the African continent, it transports an average of 84 Sv of upper IO
water into the south Atlantic and Subtropical Convergence (STC; Lutjeharms, 2006; Beal et al., 2015). It is considered the
largest of the WBCs. As such, the Agulhas Current plays a critical role in the planet's global circulation of thermocline water
and the MOC (Rahmstorf, 2003; Donners and Drijfhout, 2004; Biastoch et al., 2008; Beal et al., 2011).

It's origin lie in the convergent flows from the Mozambique Channel, the East Madagascar Current, and the southern Indian
Ocean subtropical gyre (**Figure 2**) carrying water masses from the Arabian Sea, Red Sea, and the equatorial Indian Ocean
on the shoreward side, while offshore waters comprise Atlantic Ocean, Southern Ocean, and southeast Indian Ocean
(Lutjeharms 2006; Beal et al., 2006). This convergence occurs in the vicinity of the Delagoa Bight in the southern part of
Mozambique. With a volume transport that can reach 160 Sv, it is one of the most powerful WBCs. Typically, the narrow
core (~200 km wide) has a velocity of ~2 ms$^{-1}$ with maximums reaching 3.5 m s$^{-1}$ (Lutjeharms, 2006; Beal et al., 2015). The
core closely follows the African continent's steep slope once south of the Delagoa Bight at 27°S. The very narrow shelf (~3
km) here on the off northern KwaZulu-Natal (also known as Maputoland) means that the warm subtropical surface waters
reach the coast and consequently extend the subtropical IO fauna and flora towards the poles (Turpie et al., 2000; Griffiths
et al., 2010).

South of Cape St Lucia, the coastline retracts northwards away from the shelf edge for ~120 km forming the KZN Bight
(**Figure 2**). Mid-bight, the Agulhas Current is ~40 km from the coast following the undeviating continental edge/slope. The
small KZN Bight, which has a shelf edge depth of around 100 m and mid-shelf depth of 50 m, offers the only refuge from
the strong Agulhas Current flow in the upper half of its trajectory.

Further downstream, more or less mid-length, the core again moves away from the coast as the continental shelf gradually
widens at 27°E (near Port Alfred; see **Figure 2**) to become the Agulhas Bank — an area of great importance for spawning
and the early life cycle of many of South Africa's commercially fisheries (Hutchings et al., 2002). The Agulhas Bank is the



most expansive shelf on the African continent and has a shelf edge at 200 m depth with typical mid-shelf depths around
120-150 m. The eastern part of the bank up to 22°E is commonly influenced by plumes of warm water from current meanders
which extend northward (Lutjeharms, 1989; Krug et al., 2017). The Agulhas Bank has some of the most intense thermoclines
found world-wide (Swart and Largier, 1987). At the Agulhas Bank's southern tip, the jet-like Agulhas Current becomes
unstable and undergoes several fates (Lutjeharms, 2006). Ordinarily, the core retroflects south then eastwards, forming the
Agulhas Return Current which flows along to the north of the Subtropical Convergence (STC) — a divide between the IO
and colder Southern Ocean. A temporary northward displacement of the Return Current around the Agulhas Plateau (**Figure**
**2**) at times causes a fusion (occlusion) of the ARC with the Agulhas Current resulting in the formation of warm Agulhas
Rings which propagate westwards into the south Atlantic Ocean — a critical contribution to the MOC (Biastoch et al., 2008;
Beal et al., 2011). Occasionally the end of the Agulhas Current turns northwards and follows the steep slope of the Western
Agulhas Bank.

Surface temperatures of the Agulhas Current range between 22 and 30°C in the northern reaches, reflecting seasonal
oscillations but these decrease with southward latitude along the current's length in both seasons (Lutjeharms, 2006). Being
of subtropical origins, the surface waters of the Agulhas Current are oligotrophic, but at depth, reflect nutrient concentrations
of those typical of the SEC. As with all WBCs, isopycnals slope upwards across the current towards the shelf-slope moving
nutrient-rich, cooler water to shallower depths (Lutjeharms et al., 2000; **Figure 3**).

Notwithstanding the current's planetary importance, it is also a major driver of local processes that underpin the shelf
ecosystems along the east and southern shelf region of South Africa. This is underscored in the composite image shown in
**Figure 4** where several important productivity features are highlighted by enhanced surface chlorophyll levels along the
current's boundary. Some are bathymetrically fixed — others transient. All are underpinned by some form of upwelling of
cooler, nutrient-rich water.

**2.1.2. Transient meanders and cyclonic eddies (core upwelling)**

A range of transient meanders and associated cyclonic eddies on the inshore boundary of the Agulhas Current commonly
occur, promoting shelf-edge upwelling which does not usually break the surface. The most well-known is the Natal pulse
which is observed on average 1.6 times a year, but this appearance ranges anywhere between 0 and 6 events a year
(Lutjeharms and Roberts, 1988; de Ruijter et al., 1999; Bryden et al., 2005; Rouault and Penven, 2011; Beal et al., 2015;
Leber and Beal, 2015). These large solitary meander events do not have a discernible seasonal cycle but, as pointed out,
display considerable interannual variability (Krug & Tournadre, 2012). Natal pulses are of the order of 100 km in diameter
and originate in the upper reaches of the current, usually due to the interaction of the core flow with adjacent anticyclonic
eddies (Tsugawa and Hasumi, 2010). Natal pulses propagate down the east coast of South Africa at approximately 10-20
km/day and grow in size (amplitude) (upstream ~30 km, downstream ~200 km) (Lutjeharms et al., 2003), extending the full



depth of the Agulhas Current, i.e., ~2000 m (Lutjeharms et al., 2001, 2003; Elipot and Beal (2015); Pivan et al., 2016). The
passage of a Natal pulse is often followed by the spawning of an Agulhas ring which moves off into the south Atlantic (Van
Leeuwen et al., 2000; Lutjeharms 2006; Elipot and Beal, 2015).
Natal pulses drive slope upwelling with an order of magnitude of 50–100 m per day (Bryden et al., 2005; Pivan et al., 2016),
and given their slow propagation, are associated with relatively long residence times. Their cold cyclonic cores temporarily
move deeper water onto the narrow continental slope along the Transkei shelf and are coincident with enhanced surface
chlorophyll (**Figure 5**). Their influence on the coastal waters is perhaps greatest between Port Alfred and Algoa Bay on the
far eastern Agulhas Bank, where they facilitate cross-shelf exchange (Jackson et al., 2012: Krug et al., 2014; Pivan et al.,
2016). Goschen et al. (2015) observed the dynamics of six Natal pulses here using in situ moorings, and found slope
upwelling-induced cold bottom water events (10–12°C) to extend over the entire shelf reaching the inshore areas of Algoa
Bay. These lasted last 1–3 weeks during the passing of the pulse, but the cold water on the shelf could linger a further three
weeks. During upwelling, the isotherms ascended at an average rate of 1.8m day1 as the cold bottom layer increased in
thickness to 40–60 m, although upwelled water did not break the surface in all cases. Cold water remained in the area for a
further 2–3 weeks.
These results were recently contextualized by Malan et al. (2018) using a combination of two ocean models (INALT01 and
AGU HYCOM). They showed that large meander events in the Agulhas Current drive strong shear with the shelf waters on
the meander leading and trailing edges. This induces strong negative vorticity areas, which promotes upwelling events in
the bottom boundary layer, resulting in a significant decrease in subsurface temperatures at 100 m at the shelf edge. This is
particularly prevalent along the slope of the eastern Agulhas Bank. They used a tracer experiment to observe the uplift of
water from 400 m beneath the surface of the Agulhas Current, on the leading edge of a large meander. **Figure 6** depicts the
tracer results at the onset of the meander and follows the dynamics for four weeks. Important to note is the cold-nutrient-
rich Central Water left on the shelf after the meander has moved west.
Another common recurring cold-core cyclonic eddy is the Durban break-away eddy first noted by Lutjeharms and Connell
(1989) and more recently studied by Guastella and Roberts (2016). This is a semi-permanent feature of smaller proportions
than the Natal pulse (~ 60 km). It is considered to be lee-trapped during its early development as a result of a submerged
bight off Durban in the 100 m depth contour configuration. It is hypothesized that the cyclone is spun-up by the strong
southwestward flowing Agulhas Current offshore of the regressed shelf edge near Durban. Analysis of ADCP data and
satellite imagery shows the eddy to be present off Durban approximately 55% of the time with an average lifespan of 8.6
days. After spin-up, the eddy breaks loose from its lee position and propagates downstream on the inshore boundary of the
Agulhas Current (**Figure 2**). The eddy is highly variable in occurrence, strength, and downstream propagation speeds. There
is no detectable seasonal cycle in the eddy occurrence, with the Natal pulse causing more variability than any seasonal



signal. Moorings and ship data confirm upward doming of the thermal structure in the eddy core associated with cooler
water and nutrients being moved higher in the water column, stimulating primary production. Gaustella and Roberts (2016)
also noted a second mechanism of upwelling by this feature, viz. divergent upwelling in the northern limb of the eddy (where
the cyclonic radial flow separates from the shelf). Moreover, satellite-tracked surface drifters released in the eddy
demonstrated the potential for nutrient-rich eddy water to be transported northwards along the inshore regions of the
KwaZulu-Natal (KZN) Bight, thus contributing to the functioning of the bight ecosystem, as well as southwards along the
KZN and Transkei coasts – both by the eddy migrating downstream and by eddy water being recirculated into the inshore
boundary of the Agulhas Current itself.

**2.1.3 Dynamic boundary upwelling**

Another form of upwelling also occurs at two bathymetric points along the inshore boundary of the Agulhas Current.
Historically referred to as dynamic or divergent upwelling, surface upwelling expressions (isotherm outcropping) occur west
of Cape Lucia (near Richards Bay), where the very narrow Maputoland shelf (3 km) widens to become the KZN Bight, and
near Port Alfred (27°E) further downstream where the Transkei shelf widens into the Agulhas Bank (**Figure 2**).

Both Lutjeharms et al. (2000) and Meyer et al. (2002) showed that low water temperatures of <19 °C, high salinities (c.
35.30), and nitrate levels (c. 15 µmol kg$^{-1}$) indicated upwelling in the northern KZN Bight with an epicenter between Cape
St Lucia and Richards Bay. This is the prime source of upwelled water and nutrients for the KZN Bight. This upwelling is
responsible for elevated chlorophyll levels commonly observed in the northern part of the Bight (c. 1.5 mg m$^{-3}$, cf. c. 0.5
mg m$^{-3}$. More recent work by Roberts and Nieuwenhuys (2016) showed upwelling events to last 5–10 days, with
temperatures commonly dropping by 7°C. The earlier studies (Lutjeharms et al., 2000; Meyer et al., 2002) suggested this
upwelling was topographically and dynamically driven by the juxtaposition of the Cape St Lucia offset and the Agulhas
Current (a solitary mechanism). However, Roberts and Nieuwenhuys (2016) showed almost all major and minor cold-water
intrusions on the shelf coincided with upwelling-favorable north-easterly winds that simultaneously force a southwesterly
coastal current. Analysis of in situ mooring data indicates the strongest upwelling events here are driven by a coupled
mechanism of Ekman bottom veering on the continental slope and upwelling-favorable wind.

Some 150 km south of Durban, the coastline again undergoes a small northward retraction from the shelf edge ― which
begins the slowly southward expanding Transkie shelf (at Port St Johns; see **Figure 2**). The shelf north of here is very
narrow, as is the case north of the KZN Bight. South of Durban (and the Durban Eddy), the Agulhas Current flows close to
the coast. But at Port St Johns, the Current begins to move offshore following the smooth continental slope. Roberts et al.
(2010), using S–ADCP data and satellite SST demonstrated the existence of cyclonic flow in the Port St Johns–Waterfall
Bluff coastal inset, with a northward coastal current similarly ranging in velocity between 20 and 60 cm s$^{-1}$. CTD data
indicated that this was associated with shelf-edge upwelling, with surface temperatures 2–4 °C cooler than the adjacent core



temperature (24–26 °C) of the Agulhas Current (**Figure 7**). Vertical profiles of the S–ADCP data showed that the
countercurrent, about 7 km wide, extends down the slope to at least 600 m, where it appeared to link with the deep Agulhas
Undercurrent at 800 m. It is not known how often this feature exists. Satellite images at times show enhanced surface
chlorophyll on the narrow shelf here, but often this is overtaken by passing turbulent features on the inshore boundary of
the current.

Surface upwelling near Port Alfred occurs on a much grander scale than the KZN Bight or Port St Johns, at times stretching
from East London (29°E) to Port Elizabeth (80-300 km in length **Figure 4**), and is considered the most important upwelling
on the southeast coast of South African. Lutjeharms et al. (2000), using cruise data, showed the upwelled water to originate
from a depth of 200-300 m in the Agulhas Current resulting in the water of 8-11°C moving up onto the continental shelf,
which has an edge break at 100 m depth. This colder, nutrient-rich water is derived from the upper to middle levels of South
Indian Central Water and forms a thermocline, which at times breaks the surface here, resulting in extensive chlorophyll
blooms that propagate westwards well onto the Eastern Agulhas Bank (e.g., **Figure 4**). Lutjeharms et al. (2000) suggested
that topographically induced changes in the Agulhas current structure underpin the mechanism for this 'dynamic' upwelling.
The intermittent outcropping of upwelled water occurs more than 40% of the time and dramatically changes the surface
temperatures (Lutjeharms et al., 2000). Moreover, Lutjeharms (2006) suggested that the cold, nutrient-rich bottom layer on
the eastern Agulhas Bank has its origins from upwelling in the Port Alfred region underpinning the intense thermoclines
found here (Swart and Largier, 1987). However, Leber et al. (2017) found that meanders act in combination with upwelling-
favorable winds to force the strongest cold events, while upwelling-favorable winds alone, possibly primed by Ekman
veering, force weaker cold events. This is not unlike that found near Cape St Lucia. It is found that the frontal curvature of
warm Agulhas Current meanders link with the atmosphere to drive local wind stress curl anomalies that reinforce upwelling.
[see below]

**2.1.4. Wind-driven coastal upwelling**

Surface coastal upwelling is also found along the south coast of South Africa (i.e., eastern and central Agulhas Bank), some
far removed from the Agulhas Current which is some 200 km away off Mossel Bay. This coastal upwelling is driven by the
easterly winds that dominate during the austral summer months (Walker, 1986). It has been shown that the dynamic
upwelling near Port Alfred is also augmented with easterly wind-driven coastal upwelling (Leber et al. 2016).

While the upwelling is found on the westward sides of prominent capes that reach deeper water, the epicenter occurs along
the 100 km Tsitsikamma Coast (**Figure 4**), where the coastal bathymetry is steep (Roberts and van den Berg, 2005). A 100
km-long, thin offshore extension of this upwelling is commonly observed in satellite data during the summer months. This
banana-shaped feature, known as the 'cold ridge', is associated with high levels of chlorophyll (**Figure 4**). Roberts (2005)



suggested that the cold ridge is an upwelling filament drawn out by the shelf circulation; however, this hypothesis is still
under investigation.

**2.2 The Mozambique Channel**

**2.2.1 Historical perspectives**

Oceanographic sampling within the Mozambique Channel was limited before the first International Indian Ocean Expedition
(IIOE; 1959-1965), with merely six voyages and fewer than 100 stations recorded between 1913 and 1952 (Jorge da Silva
et al., 1981). The Commandant Robert Giraud conducted extensive sampling throughout the Mozambique Channel during
October and November 1957 as part of the International Geophysical Year (Menaché, 1963), but few of the 65 stations were
located close to the coast. It seems likely that prior to the IIOE, coastal upwelling processes in this region were unknown,
as the Somali upwelling system was the only upwelling area in the western Indian Ocean to be investigated during the
expedition.
The first hydrographic data used to report on upwelling phenomena in the Mozambique Channel, as inferred from sloping
isotherms and isohalines in the upper 500 m of the water column, were collected onboard RV *Dr. Fridtjof Nansen*, which
surveyed the entire coast of Mozambique four times between August 1977 and June 1978 (IMR 1977a; IMR 1978a, b, c).
Saetre and de Paula e Silva (1979) concluded that, during the NE monsoon (Nov-April), wind-induced upwelling occurs in
a narrow strip of the ocean along the northern Mozambique coast between 11 and 16°S. Although they did not observe any
associated low temperatures or high nutrient concentrations in the surface waters, they observed cyclonic eddies off Angoche
in September and November 1977 and further south off Inhambane and along a transect off ~27°S during the September
1977 and January-March 1978 surveys. A special effort to investigate the upwelling in the northern section of the channel
was subsequently undertaken onboard the RV *Alexander von Humboldt* in February and March 1980 to determine whether
the upwelling was due mainly to wind or current effects (Nehring, 1984). Hydrographic sampling was conducted along nine
transects normal to the coast between Cabo Delgado and Angoche. During this survey, dynamic topography revealed a
cyclonic eddy in the Angoche region, with high $NO_3-$ and chlorophyll concentrations associated with the core of the eddy
(Nehring, 1984; Nehring et al. 1987).
More detailed hydrographic surveys within the Delagoa Bight by the RV *Dr. Fridtjof Nansen* in October 1980 (Brinca et
al., 1981) and RV *Ernst Haeckel* in January 1982 (Lutjeharms and Jorge da Silva, 1988) provided further information on
upwelling and circulation in this southernmost part of the Mozambique coast. Lutjeharms and Jorge da Silva (1988) used
data from all these cruises, in conjunction with satellite remote sensing SST imagery from AVHRR for the period spanning
from 1975 to 1985, to study the region in detail. Their results suggested that there is an area in the Delagoa Bight, the
Inharrime terrace, where upwelling enhances biological productivity over the continental shelf. A later study by
Kyewalyanga et al. (2007) using satellite ocean color products and a biological model corroborated this finding. Lutjeharms



and Jorge da Silva (1988) also suggested that a cyclonic lee eddy present in the Delagoa Bight during the 1980 and 1982
cruises was topographically driven and a relatively consistent feature. Between 2004 and 2006, a series of four cruises on
the RV *Algoa* was undertaken to investigate the persistence of this lee eddy, as well as the influence of passing eddies on
upwelling in the Bight, as part of the African Coelacanth Ecosystem Project (ACEP), with hydrographic and biological
sampling conducted along a series of shore-normal transects within the Bight (Lamont et al., 2010). The lee eddy was
documented only once during these cruises, leading Lamont et al. (2010) to suggest that the Delagoa Bight eddy is more
transient than previously thought.
The RV *Dr. Fridtjof Nansen* returned to the region almost three decades later in 2007 for a comprehensive ecosystem survey
of the entire Mozambique coast (Johnsen et al., 2007), and again in 2009 to survey the Angoche upwelling area during the
Agulhas and Somali Large Marine Ecosystem (ASCLME) program (Olsen et al., 2009). These efforts complemented several
hydrographic surveys within the Mozambique Channel between 2002 and 2010, driven largely by a French–South African
partnership through the multidisciplinary MESOBIO (Influence of mesoscale dynamics on biological productivity at
multiple trophic levels in the Mozambique Channel) research programme (Ternon et al., 2014), which focused on the
mesoscale eddies. Detailed information about the Angoche and Delagoa Bight upwelling events, based on hydrographic
data collected during MESOBIO, has been documented by Malauene et al. (2014), Roberts et al. (2014), and Lamont et al.

373 (2014).

**2.2.2. Mechanisms**
The northern part of the Mozambique Channel is influenced by the monsoonal wind system, with wind stress predominantly
from the north to north-east during austral summer and the south to southeast during austral winter (Saetre and Jorge da
Silva, 1982; Schott et al., 2009). The influence of the monsoon winds in the Mozambique Channel is halted at about 20°S
(Tomczak and Godfrey, 1994; Schott et al., 2009). South of this latitude, the winds are southeasterly (known as the trade
winds) almost all year round and are unfavourable for Ekman upwelling along the Mozambican coast.
The monthly mean wind stress (vectors) and wind stress curl (shading) within the Mozambique Channel and around
Madagascar are shown for different seasons in **Figure 8.** January (**Figure 8a**) represents typical austral summer conditions,
corresponding to the boreal northeast Monsoon (NEM) regime. April (fall; **Figur 8b**) represents the period of the transition
from the NEM towards the austral southeast Monsoon (SEM), shown for July, corresponding to the austral winter season
(**Figure 8c**). October (**Figure 8d**) represents the reversal of the Monsoon from the SEM to the NEM. In the southern
hemisphere, negative and positive wind stress curl correspond to Ekman suction and pumping, respectively. Ekman suction
in general leads to the emergence of upward vertical velocities within the water column, resulting in upwelling (blue areas),
whereas Ekman pumping leads downward vertical velocities, leading to downwelling events (red areas). The strongest
upwelling is predicted around Madagascar, especially during July and October.



With over 30 cruises in the Mozambique Channel since the late 1970s, there is now a clear picture of where upwelling
hotspots are located along the Mozambique coast. In the northern sector, upwelling develops at Angoche, off the coast of
Nampula between 15°S and 18°S, around the narrows of the Channel (**Figure 9**). Upwelling in the southern sector of the
Mozambique Channel is more variable with regards to location, but several hotspot regions are evident, such as on the Sofala
Bank, at Ponta Zavora, around Inhambane, and at the Delagoa Bight, directly offshore from the Mozambican capital Maputo.
Upwelling within the Mozambique Channel, both in the northern and southern sectors, can be ascribed to two dynamic
forcing mechanisms, one linked to the local characteristics of the oceanic circulation, and the other linked to the atmospheric
wind forcing that transfers its momentum into the ocean's interior (Nehring et al., 1987; Quartly and Srokosz, 2004;
Malauene et al., 2014; Roberts et al., 2014).
The drivers of upwelling at Angoche in the northern Mozambique Channel were recently investigated by Malauene et al.
(2014), who inferred dominance of both wind-stress and oceanic mesoscale current instabilities. Data from an in situ
underwater temperature recorder (UTR) deployed near Angoche between 2002 and 2007, combined with satellite data,
revealed intermittent "cool water" events between August and March, which coincides with the period of the northeast
monsoon winds. During this period, the alongshore winds in the northern Mozambique Channel are southward oriented and
upwelling favorable; hence they induce surface divergence in the upper water column, thereby establishing the onset of
wind-driven Ekman coastal upwelling (Malauene et al., 2014). This seasonal wind-driven coastal upwelling results in
elevated chlorophyll-a signatures over an area between 15 and 18°S (Malauene et al., 2014).
The other contribution to upwelling at Angoche has been attributed to the dynamics of anticyclonic-cyclonic eddy pair
interaction with the continental shelf (Malauene et al., 2014), due to the southward passage of large anticyclonic eddies and
rings along the western boundary of the Channel (**Figure 9**; de Ruijter et al., 2002; Ridderinkhof and de Ruiter, 2003; Halo
et al., 2014). The interaction of mesoscale eddies with the continental slope on the western side of the Mozambique Channel
has been shown to cause upwelling of cooler, nutrient-rich water, resulting in elevated phytoplankton biomass in the shelf
regions, as described further below (Lamont et al., 2014; Roberts et al., 2014). Malauene et al. (2014) suggested that the
cool surface, elevated chlorophyll-a waters off Angoche are primed and formed by favourable wind-driven Ekman-type
coastal upwelling during August and March, but may be further enhanced in chlorophyll-a by anticyclonic/cyclonic eddy
pairs interacting with the shelf.
The interaction between mesoscale eddies and the Mozambican western boundary is intense and a frequent occurrence. This
interaction also causes lateral divergence of the flow-field and has been regarded as an important driver of the observed
upwelling events through slope current topographic-driven upwelling occurring predominantly at Ponta Zavora and Sofala
Bank (Roberts et al., 2014; Lamont et al., 2014). Roberts et al. (2014) used in situ observations of ocean currents measured
by a ship-borne Acoustic Doppler Current Profiler (S-ADCP) and hydrographic data from Conductivity Temperature Depth
(CTD) casts to investigate the interaction of a dipole eddy (with the cyclone to the south of the anticyclone, tracked using



altimetry maps of sea level anomalies) with the western continental slope of the southern Mozambique Channel, near
Inhambane. They observed strong (>100 cm s$^{-1}$) southward currents over the slope adjacent to the anticyclone, with
horizontal divergence over the shelf at the southern edge of the anticyclone, and intense slope upwelling between the dipole
and the shelf. Nutrient and chlorophyll concentrations were enhanced in the near-surface waters over the shelf, although
there was no evidence of upwelling at the surface. Data from a nearby UTR confirmed prolonged bouts of slope upwelling
over several weeks until the dipole had moved further south. Combined altimetry and UTR data also slowed that both
cyclonic and anticyclonic independent eddies (not part of a dipole) along the Mozambique continental shelf may induce
slope upwelling, with divergence north of the contact zone in the case of cyclonic eddies (Roberts et al., 2014). Cyclonic
eddies are usually associated with vertical pumping in the eddy's interior, favouring upwelling of nutrient-rich deep waters
(i.e., new production) into the euphotic zone, particularly during the spin-up phase (Robinson, 1983; Tew-Kai and Marsac,

431    2009).

The southernmost upwelling region in the Mozambique Channel is the Delagoa Bight, centered around 26$^o$S and 34$^o$E
(Lutjeharms and Da Silva, 1988; Lamont et al., 2010). The region is one of the largest coastal indentations in the southwest
Indian Ocean, and the second richest area in terms of shrimp fisheries in the country, after the Sofala Bank. The oceanic
circulation in the Bight is dominated by a semi-permanent cyclonic lee eddy (Lutjeharms and Da Silva, 1988; Cossa et al.,
2016), which is topographically trapped and appears to occur about 25% of the time, with no clear seasonal signal (Cossa
et al., 2016). The formation of the lee eddy in the Bight has been linked to the characteristics of the flow-field offshore,
especially the Mozambique Channel rings. In particular, the passage of cyclonic eddies off the Inhambane region influences
the water masses of the Delagoa Bight through upwelling onto the shelf, resulting in enhanced productivity (Quartly and
Srokosz, 2004; Kyewalyanga et al., 2007; Lamont et al., 2010; Lamont et al., 2014). Kyewalyanga et al. (2007) recorded
high chlorophyll a and primary production values in the northern part of the Delagoa Bight (**Figure 10**), where pelagic fish,
mostly round herring (*Etrumeus teres*) have previously been recorded (Brinca et al., 1981).
**2.2.3 Ecosystem impacts**
In addition to stimulating primary production along the continental shelf of Mozambique, often in areas associated with
higher biomass or pelagic fish or shrimps, the mesoscale eddies play an important role in ecosystem dynamics in the
Mozambique Channel through the stimulation of new primary production via upwelling in cyclonic eddies, as well as the
broad distribution of both coastal upwelling-generated and eddy-generated production. Using isotopic tracers, Kolasinski et
al. (2012) showed that the new production is circulated throughout the mixed layer, while some cyclonic production may
also be exported horizontally into the frontal region. Strong currents at the perimeters of these eddies result in the
entrainment and offshore advection of this high biomass, dominated by siliceous diatoms, into the frontal regions (Kolasinski
et al., 2012). Huggett (2014) found mesozooplankton populations were significantly enriched within the cyclonic eddies and
divergence areas, with a higher abundance of copepod and euphausiid nauplii observed in the cyclonic eddies compared to



the anticyclonic eddies. This suggests that the divergence areas are constantly "fed" by production from within the cyclonic
eddies. This concentration of coastal production combined with the import of cyclonic production into the boundary region
might explain why it is often the boundaries of eddies that are targeted by consumers in the Mozambique Channel. Sabarros
et al. (2009) documented large aggregations of micronekton (small forage organisms including crustaceans, squid, and fish)
mainly in areas where the local horizontal gradient of sea level anomalies is strong, i.e. at the periphery of eddies, and
foraging frigatebirds tend to avoid the centre of cold-core (cyclonic) eddies, preferring the eddy edges (Weimerskirch et al.,
2004). Mesoscale eddies are also thought to provide better conditions for tuna aggregations throughout the water column,
not just at the surface, and high species diversity among longline catches (tunas and swordfish) in the MC suggests the
eddies may function as biodiversity hotspots (Tew-Kai & Marsac, 2010). Through upwelling in the core of cyclonic eddies
and offshore entrainment of shelf production in the inter-eddy frontal zones, mesoscale eddies are a major source and
distributor of production and organic matter in an otherwise oligotrophic system, and a key driver in supporting the high
biomass and diversity of pelagic consumers observed in this region.

**2.3 Madagascar**
**2.3.1 Historical perspectives**
The island of Madagascar received little attention both before and during the IIOE. The transect made by RV *Atlantis II* in
1963, departing from Maputo at the Delagoa Bight, simply crossed the southern Madagascar coast as a pathway to Reunion
and Mauritius Islands (Miller and Risebrough, 1963). No wonder not even the name Madagascar is mentioned in their
description (Wallen, 1964; Fye, 1965). If a potential upwelling zone off southern Madagascar upwelling had been known
of then, surely a drive to investigate it during the IIOE would have been easily motivated.
Even since the IIOE, relatively few large-scale hydrographic surveys have been conducted along the coastline of
Madagascar, which at ~48000 km is the longest in Africa. The first extensive oceanographic survey over the southern
continental shelf of Madagascar to provide evidence of upwelling was conducted in June 1983 onboard the RV *Dr. Fridtjof*
*Nansen* (IMR, 1983a; Lutjeharms 2006). In the south, inshore surface temperatures in the vicinity of Cap Sainte Marie, and
Taolagnaro (Fort Dauphin) at the southeastern corner of the shelf, were about 2°C lower than farther offshore, with salinities
indicating upwelled Subtropical Surface Water originating from depths of about 200 m. Just over a quarter of a century later,
the first "circumnavigation" of this large island was achieved through two "ecosystem surveys" in 2008 and 2009 by the
RV *Dr. Fridtjof Nansen* during the ASCLME programme. Between 24 August and 1 October 2008, the *Nansen* completed
115 CTD stations in total along 11 transects extending far offshore along the south and east coasts of Madagascar, ending
at the northern tip (Krakstad et al., 2008). Evidence was found of upwelled Subtropical Surface water at the southeastern
corner of the shelf (25°S), while relatively fresher and cooler water inshore at 16°S and 14°S was suggestive of upwelling



along the northeast coast (Krakstad et al., 2008). One year later, from 25 August to 3 October 2009, the Nansen revisited
the western sector of the south coast and continued sampling along the southwestern and north-western coasts, ending once
more at Antsiranana (Diego Suárez) in the north, completing 10 transects and 182 hydrographic stations (Alvheim et al.
2009). Once again, hydrographic sampling provided evidence of coastal upwelling on the southern coast (26°S), as well as
at two locations on the west coast, near Cap Sainte André (16°S) and Nosy Be Island (13°S), with salinity maxima indicating
upwelling of Subtropical Surface Water in the south and Equatorial Surface Water in the northern region (Pripp et al., 2014).
**2.3.2 Mechanisms**
Seasonal maps of wind stress curl indicate both strong upwelling and downwelling events around Madagascar are likely
during austral winter (July, **Figure 8c**) through to late spring (October, **Figure 8d**). In July, the strongest upwelling is
predicted to the northwest of Madagascar, around the Comoros basin. During this period, the winds are from the southeast.
In October, the strongest upwelling is predicted all around the south, southeast, and southwest coasts of Madagascar. During
this period, the winds have a northeast orientation along the southeastern coast, and a southeast orientation along the
southwestern coast of the Island, thus becoming upwelling favourable.
Since the first observation of upwelling off southern Madagascar, there has been considerable interest amongst the scientific
oceanographic community, both locally and internationally, to confirm this upwelling and understand the physical
mechanisms of its formation, frequency, characteristics, and spatial extension and temporal variability (Lutjeharms and
Machu, 2000; DiMarco et al., 2000; Machu et al., 2002; Ho et al., 2004; Srokosz and Quartly, 2013; Ramanantsoa et al.,
2018; Collins, 2020). Lutjeharms and Machu (2000) used a snapshot composed satellite SST imagery from Advanced High-
Resolution Radiometer (AVHRR) sensor onboard of NOAA satellite, with a spatial resolution of $1^o$x$1^o$ longitude and
latitude, in conjunction with chlorophyll-a concentrations retrieved by SeaWiFS satellite, and Scatterometer wind field data
from Quickscat satellite, to inspect the mechanisms of formation of this upwelling. Their finding suggested that this
upwelling was caused by current instabilities at the inshore edge of the South East Madagascar Current, as no correlation
was found with the local winds (Lutjeharms and Machu, 2000). In a parallel study using SST and wind field data from the
same sources, DiMarco et al. (2000) concluded that upwelling over the southern continental shelf and along the southeastern
continental slope, which extended over an area of  $2^o$ longitude by $1^o$ latitude (nearly 24,642 km$^2$) during February and
March (North-East Monsoon), was driven by both wind forcing and current interactions with the continental shelf and slope.
However, the paucity of in situ wind and current data prevented them from quantifying the relative contribution of each
process.
Machu and colleagues revisited the topic soon thereafter, and surveyed the southern and southeastern continental shelf of
Madagascar on board the Dutch RV *Pelagia*, during the second phase of the Agulhas Current Source Experiment (ACSEX-
2) project in March 2001. Hydrographic measurements conducted along three transects provided the first dedicated and



comprehensive hydrographic evidence of the upwelling cell inshore of the EMC. The combination of this dataset and
satellite imagery led the authors to conclude that the southeastern Madagascar upwelling occurs through a combination of
favourable wind stress in the area, enabling an Ekman wind-driven mechanism, and the dynamics of a cyclonic eddy
generated inshore of the current, favoured by the concave-shaped bathymetry as the shelf widens (Machu et al., 2002).
An attempt to study the long-term inter-annual variability of the upwelling events to the south and southeastern Madagascar
and their interaction with the EMC was conducted by Ho et al. (2004). Their analysis of monthly SeaWiFs Chlorophyll-a
imagery spanning from September 1997 to November 2001 revealed that the upwelling was generally enhanced in austral
winter and austral summer each year. They also concluded that the southern and southeastern upwelling boundary cells
interact, based on the movement and deformation of the boundary between them, with a mechanism that can be explained
by the shear wave propagation theory (Ho et al., 2004).
More recently, Ramanantsoa et al. (2018a) investigated the temporal and spatial variability of the coastal upwelling south
of Madagascar. Using a suite of satellite remote sensing data, in-situ observations, and numerical model simulations, they
provide new insight on the structure, variability, and drivers of this upwelling. Their results suggest that the southern and
southeastern upwelling cells already indicated in former studies (Ho et al., 2004), which they termed core 2 and core 1
respectively, are characterized by distinct seasonal variability, have different intensities and water mass origins, and are
formed by different physical mechanisms (Ramanantsoa et al., 2018a). The core in the southeastern sector is attributed to
dynamical upwelling in response to the detachment of the EMC from the continental slope, reinforced by favorable winds.
The southern core, situated to the west of the southern tip of Madagascar (Cap Ste Marie), is primarily attributed to Ekman-
driven upwelling by favourable winds, whilst being inhibited by the recently described warm poleward current along the
eastern boundary of the Mozambique Channel, the Southwest Madagascar Coastal Current, or SMACC (Ramanantsoa et
al., 2018b).
During the *Nansen* survey in 2009, Pripp et al. (2014) observed upwelling off Cap Ste Andre and Nosy Be along the
northwest coast, with elevated sea surface salinities indicative of upwelled Equatorial Surface Water. They suggested this
upwelling was most likely current-driven due to strong northeastward bottom currents associated with passing anticyclonic
eddies, which would have resulted in onshore bottom Ekman transport.
**2.3.3. Productivity and ecosystem effects**
As with other upwelling regions, the upwelling areas on the Madagascar shelf are associated with elevated biological
productivity. During the 2009 survey, Pripp et al. (2014) found all upwelling cells to be associated with relatively high
surface chlorophyll and satellite-derived net primary production (NPP), as well as higher acoustic estimates of pelagic fish,
elevated pelagic and demersal trawl catches, and greater whale sightings. Ockhuis et al. (2017) found the highest neuston



biovolume on the Madagascan shelf to be associated with relatively cool water (<22 °C) in the core upwelling areas, and
Ramanantsoa et al. (2018a) describe the coastal upwelling area south of Madagascar as a hotspot of marine biological
productivity. As has been observed for the Mozambique coast, the interaction of eddies with the continental shelf can lead
to the export of this shelf-based, upwelling-derived production into the open ocean. A young cyclonic eddy that formed off
southern Madagascar in 2013 was observed to entrain chlorophyll-rich shelf water around its perimeter (Barlow et al., 2017),
with the associated entrapment of plankton having implications for the dispersal and recruitment of larval stages and
biological connectivity between regions (Braby, 2014; Noyon et al., 2019).
The southeast core of current-driven upwelling has been proposed (Longhurst 2001; Lévy et al., 2007; Raj et al., 2010;
Srokosz & Quartly, 2013) to be the main driver of the South-East Madagascar Bloom, an extensive
phytoplankton/cyanobacteria bloom that has been shown by satellite imagery to occur to the southeast of Madagascar during
late austral summer). However, analysis of a 19-year time series of ocean color satellite data by Dilmahamod et al. (2019)
laid this as well as other theories to rest. Bloom occurrence was associated with La Niña conditions when upwelling intensity
south of Madagascar was reduced due to a stronger than average Southeast Madagascar Current detaching from the coast.
The resultant feeding of low-salinity water into the Madagascar Basin and enhanced stratification, along with ample light,
are suggested as ideal conditions for a nitrogen-fixing cyanobacterial bloom onset (Dilmahamod et al., 2019).
**2.4 East African Coastal Current system**
**2.4.1. Background**
The equatorward-flowing East African Coastal Current (EACC) is present along the coasts of Tanzania and Kenya between
11$^{o}$S and 3$^{o}$S (**Figure 11**). Transporting about 19.9 Sv, as estimated by Swallow et al. (1991), the EACC draws much of its
water from the westward-flowing South Equatorial Current. Even though it experiences the impact of the seasonally
reversing winds, the northeast Monsoon in austral summer (NEM, November to March) and southeast monsoon in austral
winter (SEM, April to October), the EACC is northward-oriented all year round, in contrast to the Somali Current located
in its downstream bounds, which reverses its southward – northward orientation in synchrony with the reversal of the
monsoons (Wyrtki, 1973; Schott, 1983; Tomczak and Godfrey, 1994). Downwelling is prevalent throughout the year,
particularly during the SEM when the coastal current is strongest, but irregular upwelling has been observed near the
northern Kenyan coast during the NEM when the EACC moves away from the coast in the region of the confluence with
the southward-flowing Somali Current (Heip et al., 1995; Jacobs et al., 2020).
Although upwelling off the East African coast was first documented by Newell (1959), later confirmed by Iversen et al.
(1984), Bakun et al. (1998), and Roberts et al. (2008), it is only recently that the importance of these coastal upwelling cells
have been given deserved consideration through various regional research initiatives, such as the Productivity of the East



African Coastal Current (PEACC) project, the Sustainable Oceans, Livelihoods and food Security Through Increased
Capacity in Ecosystem research in the Western Indian Ocean (SOLSTICE-WIO) programme (www.solstice-wio.org), and
the Western Indian Ocean Upwelling Research Initiative (WIOURI) flagship programme of the IIOE-2, due to their potential
to sustain food security to local coastal communities (Roberts, 2015). The dynamics of the overlying atmospheric wind
forcing (Varela et al., 2015) and the progression of the EACC through the chain of small scale islands (from south to north
- Mafia, Zanzibar and Pemba) along the coast of Tanzania (Roberts et al., 2008), combined with the varying local bottom
topography characterized by the presence of shallow banks along the coast of Kenya, have been identified as potential
drivers of upwelling events in the region (Roberts et al., 2008; Roberts 2015; Jacobs et al., 2020).
**2.4.2 Upwelling mechanisms**
The southern continental shelf off Kenya is very narrow (0-3 km wide), but in the northern sector the shelf widens to
approximately 45 km due to the presence of the North Kenya Banks (NKBs; Nguli 1995; Jacobs et al. 2020). Upwelling
events along the Kenyan coast are thought to be driven primarily by the northeast monsoonal winds that favor Ekman-driven
coastal upwelling and increased productivity during November -- April (Heip et al., 1995; Varela et al., 2015). However,
recent findings based on outputs from a high-resolution global biogeochemical model and satellite remote sensing
observations along the Kenyan coast suggest that, during the NEM, the Ekman wind-driven coastal upwelling is further
enhanced in the NKBs by a secondary dynamical process, topographically induced shelf-break upwelling, (Jacobs et al.,
2020). This shelf-break upwelling showed high levels of spatial and intensity variability at interannual timescales, related
to the confluence position between the EACC and the Somali Current (**Figure 12 a, b**). The model indicated that shelf-edge
upwelling and productivity were enhanced over the NKBs when the confluence was located further south.
Along the coast of Tanzania, both the North-East Monsoonal winds and shear instabilities between the EACC and the chain
of islands along the coast have been attributed as responsible physical mechanisms driving upwelling in the region, as
suggested by a modeling study by Halo et al. (in review). Roberts (2015) suggested elevated chlorophyll-a concentrations
in the lee (downstream) of Zanzibar Island, in particular, and to a lesser extent off Pemba Island, measured during a survey
in 2007, were a consequence of localized upwelling wake induced by an island wake (Roberts, 2015). A ROMS model
constructed by Zavala-Garay et al. (2015) also shows cool temperatures in the Zanzibar Channel during the NEM, potentially
caused by wind-induced upwelling north of Zanzibar Channel, followed by advection into the Zanzibar Channel. A small
but intense upwelling cell also develops around Tanga, between Pemba Island and the Tanzanian coast. This small upwelling
cell has been observed in both monsoons (**Figure 12**), suggesting it is a regular occurrence (Halo et al., in review).
**2.4.3 Productivity and ecosystem impacts**



The modeling study by Jacobs et al. (2020) found that upwelling of cold, nutrient-rich water along the Kenyan coast during
the NEM results in elevated chlorophyll, primary production, and phytoplankton biomass. This was particularly enhanced
over the NKBs (**Figure 12 c, d**) and likely to contribute to higher fishery potential in this area, which has been traditionally
low along the Kenyan coast. Interannual variability in wind strength during the NEM is likely to be an important factor
controlling upwelling intensity and subsequent phytoplankton production in the region (Painter, 2020). However, a recent
study by Varela et al. (2015) documented a long-term decline in coastal upwelling off Kenya during the NEM for 1982-
2010, which suggests that upwelling-related productivity may decline in the long-term if this trend continues. In contrast,
analysis of weather station data for the period 1977-2006 generally showed long-term increases in winds along the coast of
Tanzania, although the trends in mean and maximum wind speed varied with latitude and season (Mahongo et al., 2012).
Long-term trends were stronger during the SEM than during the NEM, with increased wind speeds for Tanga and Zanzibar
in the north, but a decline in maximum wind speed for Mtwara in the south, and constant maximum wind speeds for Dar es
Salaam. A coastal upwelling index (CUI) based on SST output from a coupled biophysical climatological model by Halo et
al. (in review) showed a moderate and steady linear increase in upwelling for Tanga over a 23-year period (1990-2013), in
line with the regional increase in wind speed observed by Mahongo et al. (2012).
The limited biogeochemical data for the EACC region were recently reviewed by Painter (2020), who noted that the warm
surface waters are permanently N – limited, with low $NO_3^-$:$PO_4^{3-}$, conditions that favor the nitrogen-fixing
cyanobacterium. *Trichodesmium* colonies are generally more abundant during the NEM off both Kenya and Tanzania
(Kromkamp et al., 1997; Lugomela et al., 2002), but this is unlikely to be related to upwelled nutrients, and more likely due
to wind-borne aeolian dust and land-based nutrient input during the rains, as well as and the warmer, more stable conditions
that prevail during the NEM compared to during the SEM. Sampling in Kenyan waters aboard RV *Tyro* in 1992, Kromkamp
et al. (1995) measured higher rates of primary production during the NEM than during the SW monsoon, with maximum
rates of 6 $gCm^{-2}d^{-1}$. Zooplankton biomass was also higher during the NEM, with maximum values of 18.6 mg C $m^{-3}$
(Mwaluma, 1995).
**2.5 Coast of Somalia**
Coastal currents off Somalia exhibit a strong annual cycle forced primarily by the seasonally reversing monsoon winds.
While during winter, alongshore currents are equatorward, during summer it is poleward and exhibits one of the strongest
coastal upwelling of the north Indian Ocean. Until the early 1990s', the Somali upwelling regime was the most studied
region of the Indian Ocean, and a comprehensive review of these studies was given in Shetye and Gouveia (1998), Schott



and McCreary (2001), and Hood et al. (2015).  Below the historical studies are highlighted briefly and focus primarily on
the recent research on this upwelling region over the last few decades.
**2.5.1 Background**
In early May, as the Intertropical-Convergence-Zone move north of the equator, the northward East African Coastal Current
crosses the equator and extends till about 3-4°N along the Somali coast and then recirculate to form a gyral circulation called
the Southern Gyre (SG) (Duing et al., 1980).  A portion of this SG meanders eastward and the rest flows southward to cross
the equator offshore (Chatterjee et al., 2013). During this process, a cold upwelling wedge forms along its western and
northern front. As the Monsoon progresses, currents north of the SG turn very complex. By June, the southwesterly monsoon
winds (Findlater Jet; Findlater, 1969) strengthen along the coast resulting in a strong alongshore current all along the Somali
coast extending up to a depth of 1000 m and the offshore Ekman transport induced by strong alongshore winds causes a
strong upwelling off the coast of Somalia. By July/August, currents along the Somali coast strengthen rapidly to reach up
to 250-300 cm/s with transport reaching up to 37 Sv (Fischer et al., 1996; Beal and Donohue, 2013) and thus forms the
strongest boundary current of the north Indian Ocean. In the process, another gyre forms towards the offshore side of the
northern part of the Somali coast between ~5-9°N, known as Great Whirl (GW) (Leetmaa et al., 1982). This time, a second
cold-wedge forms along the northern flank of the GW north of ~9°N, where SST falls below 20°C. Interestingly, the early
spin-up of GW can be traced to April, two months before the Findlater Jet initiation, and is linked to the annual Rossby
waves radiated from the west coast of India (Shankar et al., 2002; Beal and Donohue, 2013). As the summer monsoon winds
strengthen, the current along its northern front strengthens and transports up to 60 Sv during August. In some years, the
Somali current continues to flow north through the Socotra passage and across the mouth of the Gulf of Aden (Beal and
Chereskin, 2003). Occasionally the Somali current also flows offshore to the northeast of GW and forms a third anticyclonic
eddy east of Socotra known as Socotra Gyre (Bruce, 1979; Fischer et al., 1996). During the late summer monsoon
(August/September), SG weakens or even disappears in the south, but GW maintains its strength (Schott and McCreary,
2001; Chatterjee et al., 2019). Few observations and modeling study suggests that in late summer, SG migrate north to
coalesce with the GW (Evans and Brown, 1981; Schott, 1983; Swallow and Fieux, 1982; Swallow et al., 1983; Luther and
O'Brien, 1989), but the absence of such migration during WOCE 1995-96 observations (Schott et al., 1997) suggests that
such events may be rare and therefore, does not reflect in climatological maps as well (Chatterjee et al., 2019). Finally, by
late September, as the summer monsoon winds start to withdraw from the north Indian Ocean, the strength of the Somali



Current and GW weakens. GW typically survives another month after the southwesterly winds wane off from the Somalia
coast.
The summer monsoon upwelling off the coast of Somalia also drives one of the most productive zones of the north Indian
Ocean. As the southwesterly alongshore winds strengthen, Ekman transport pushes the coastal surface water offshore,
leading to cold subsurface water to upwell and then advect away offshore by the strong SG and GW fronts. This upwelled
water brings a bounteous amount of nutrients to the euphotic zone (more than 15 μM), which result in enhanced
phytoplankton concentration in the upper surface layer (Smith and Codispoti, 1980; Hitchcock and Olson, 1992; McCreary
et al., 1996a, Wiggert et al., 2005).
**2.5.2 Observations**
This is one of the regions of the Indian Ocean where the number of observations has been sparse and mainly dates back to
the early campaigns of the north Indian Ocean in the 1960s' and 1970s' (Chatterjee et al., 2012). In the last couple of
decades, observing networks have grown tremendously over most parts of the Indian Ocean, but owing to the infestation by
pirates (McPhaden et al., 2009), the Somali region remains a data void region.
The first modern description of hydrography and circulation across the Somali coast was provided based on cruise based
observations between August-September of 1964 (Warren et al., 1966; Swallow and Bruce, 1966) under the first
International Indian Ocean Expedition (IIOE); a series of cross-shore hydrographic sections were carried out between 3°S-
12°N. They observed upwelled cold surface temperature (reaching up to 12.8°C) north of 7°N, and these cold waters spread
offshore as cold tongues along the northern flank of the GW reaching up to 55°E. Later, an extensive survey of the Somali
basin and the western Indian Ocean was carried out in the summer of 1979 using a multi-ship observation campaign known
as the Indian Ocean Experiment (INDEX) under the framework of the Indian Ocean Panel of SCOR. Based on samples
collected during INDEX, two separate zones of upwelling were identified: one in the south at ~3-4°N associated with SG
and the another in the northern part of the coast north at ~9°N linked to the fronts associated to GW with a minimum SST
of ~ 17°C (Leetmaa et al., 1982, Quadfasel and Schott, 1982) and surface $NO_3^-$ concentration of ~15-20 μmole/liter (Smith
and Codispoti, 1980). This enhanced nutrients level increases productivity significantly to more than 300 g C m-2 yr-1
(Heileman and Scott, 2008). It was also observed that, in the middle part of the coast between ~5-8°N, the surface
concentration of the $NO_3^-$ is relatively much lower, with maximum concentration reaching up to 1.8 μmole/liter even during
the peak monsoon (July/August) (Smith and Codispoti, 1980). Veldhuis et al. (1997) also reported strong upwelling with
surface temperature no more than 20°C and dominance of diatoms between 7-11°N along the Somali coast in July 1992. By
the late 90s' the availability of the remotely sensed satellite observations provided an opportunity to observe long term SST

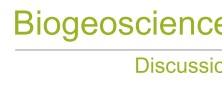

variability along this coast and is being used widely for understanding the seasonal variability and climatic trend of coastal
upwelling of this region (Goes et al., 2005; Wiggert et al., 2005; Prakash and Ramesh, 2007; Beal and Donohue, 2013).
Due to the large concentration of nutrients in the upper euphotic zone, the phytoplankton communities are mostly dominated
by large phytoplankton (diatoms) in the upwelled waters of the western Arabian Sea during the summer monsoon (Brown
et al., 1999; Shalapyonok et al., 2001; Wiggert et al., 2005). Despite the large abundance of nutrients in the upwelling
wedges off Somalia, the concentration of chlorophyll does not grow exponentially. Smith and Codispoti (1980) suggest that
the zooplankton grazing is the primary factor that limits the phytoplankton from growing exponentially. Few other studies
suggest that the swift Somali Current spreads these upwelled nutrients over a large part of the interior Arabian Sea and thus
enhances the productivity offshore (Keen et al., 1997; Hitchcock et al., 2000; Prasanna Kumar et al., 2001; Kawamiya,
2001). Later, based on in-situ measurements in the western Arabian Sea, Naqvi et al. (2010) suggested that dissolved iron
is one of the stressed micro-nutrient in this region and thus makes it an iron-limited High Nutrient Low Chlorophyll (HNLC)
zone during the summer monsoon.

### 2.5.3 Modelling and Mechanisms

Strong currents and double gyres off the Somali coast is one of the most studied phenomena in the 1970s to late 1990s. The
pioneering works by Lighthill (1969) and Cox (1979) were the first modeling study on the strong Somali currents during
the summer monsoon. Lighthill (1969) showed that as the westward propagating planetary waves excited by the offshore
negative wind stress curl reflect along the continental boundary off Somalia, they generate short-wavelength Rossby waves
that superpose to form the boundary currents. Thereafter, several papers studied the various aspects of this current system
and mainly focused on the dynamical mechanisms of the alongshore currents, the generation and decay of the two gyre
circulations off the Somalia coast, the impact of the slanted boundary in the propagation of these gyres (Anderson and
Moore, 1979; Cox, 1979; McCreary and Kundu; 1988; Luther and O'Brien, 1989; McCreary et al., 1993) and the impact of
internal instabilities (Wirth et al., 2002; Jochum and Murtugudde, 2005; Chatterjee et al., 2013). In a recent study using a
coupled ocean general circulation model, Chatterjee et al. (2019) showed that the upwelling of Somalia is limited to the
early phase of the summer monsoon when the low-level Findlater Jet sets in across the Arabian Sea (**Figure 13**). As the
Monsoon progresses, Ekman pumping induced by offshore negative wind stress curl deepens the thermocline in the interior
Arabian Sea. Subsequently, these downwelling signals propagate westward to interfere with the upwelling signals off
Somalia. As a response, the thermocline along the major part of the Somalia coast (~60%) deepens by about 40-60 m,
particularly in the central part of the Somali coast. Moreover, strong alongshore winds and weaker stratification allow more
mixing in the bottom of the mixed layer, which further deepens the thermocline and cools the surface mixed layer. As a





result, during the peak summer months, upwelling becomes limited primarily to the eddy dominated frontal flows in the
northern and to some extent in the southern part of the coast.
There are relatively much less modeling studies on the observed intense productivity in response to the upwelling. McCreary
et al. (1996) demonstrated the first reasonable simulation of the annual variability of the surface chlorophyll bloom of the
Arabian Sea based on a simple 2½ layer model coupled with an NPZD biological module. He showed that the phytoplankton
blooms in the northern and central Arabian Sea during summer monsoon is primarily driven by the lateral advection of
upwelled nutrients off the Somalia and Oman coasts and local entrainment. However, it was noted that the model
underestimates the lateral advection as it does not resolve the mesoscale features like filaments that transport nutrients
offshore in the real ocean. Later, Kawamiya (2001) studied extensively the role of this offshore advected nutrients from the
coastal upwelling region in the open ocean of the Arabian Sea and concluded that Somali upwelling is the primary source
of nutrient supply into the southcentral Arabian Sea and the Oman upwelling water supplies nutrient in the northern Arabian
Sea. These coupled physical-biogeochemical models were also used to identify the most limiting factors that suppress the
exponential growth of the phytoplankton in this region. McCreary et al. (1996) showed that nutrients and not the zooplankton
grazing primarily limit phytoplankton growth in the upwelling region, as was suggested earlier (Smith and Codispoti, 1980).
However, as they used a very simple 4 component NPZD model, it was not clear, which are the limiting nutrients that control
the phytoplankton growth. On the other hand, studies with the help of more complex models, in the last couple of decades,
suggest that phytoplankton growth in this region are prone to iron limitation (Wiggert et al., 2006; Wiggert and Murtugudde,
2007) and also likely to be silicate stressed (Kone´ et al., 2009; Resplandy et al., 2011). Recently, Lakshmi et al. (2020)
studied various limiting factors and distribution of phytoplankton along the coast of Somalia using a high-resolution
physical-biogeochemical coupled model. They showed that high values of chlorophyll concentration are limited to the
northern flank of the GW north of 9°N and exhibit moderate or low concentration in the south. The strong boundary currents
advect the upwelled nutrients from the southern region to the northern part of the coast and thereby accumulate the advected
nutrients. In contrast, the deepening of the thermocline and horizontal advection keep chlorophyll concentration low to the
south of 9°N. They further noted that dissolved iron concentration (~1.2-1.8 nM) and the NO3-:Fe ratio (<15000) do not
indicate iron-deficient conditions throughout the coast but suggests NO3- limited growth of phytoplankton communities
south of 9°N.
**2.6 Upwelling off the coast of Arabia**

Unlike the Somali coast, upwelling along Arabia (the coast of Yemen and Oman) is more uniform and exhibits classical
upwelling dynamics primarily driven by southwesterly alongshore winds during the summer monsoon. In the 1990's the
repeated multiple alongshore/crosshore ship-based transacts under the US Joint Global Ocean Flux Study (US JGOFS), and
the availability of the satellite observations of SLA, SST, and Chl-a led to a significant advancement in the understanding





of the coastal current system and its associated upwelling dynamics of this region. A detailed review based on these
observations is presented in Schott and McCreary (2001) and Hood et al. (2017). Here we briefly highlight some of these
results and review recent advances in our understanding of this upwelling system.

The alongshore wind off the coast of Arabia is much weaker than that off the Somali coast but significant enough to cause
coastal upwelling as early as May (Kindle et al., 2002), much before the development of southwest Monsoon. The upwelling
strengthens as the magnitude of the alongshore winds become stronger with the progression of the summer monsoon. During
the late summer (August/September), SST close to the coast decreases by about 5°C from the ambient offshore temperature
to fall below 23°C (Shi et al., 2000). Owing to the offshore Ekman transport driven by the alongshore winds, sea level also
drops by more than 30 cm along the coast. This is the time, owing to the crosshore sea level gradient, a northeastward coastal
current, Oman Coastal Current (OCC; Shi et al., 2000) which persists throughout the summer monsoon (Cutler and Swallow,
1984). Interestingly, the maximum strength of the alongshore winds does not coincide with the minimum SST and sea level:
while the alongshore wind reaches its peak in mid-June, the SST and sea level attain their minimum about one and a half
month later by the end of August or early September (Manghnani et al., 1998; Vic et al., 2017). The reason for this delay is
not very clear. However, Vic et al. (2017) indicated that remotely forced Rossby waves generated due to offshore Ekman
pumping by the upwelling favorable wind stress curl (Smith and Bottero, 1977) north of the Findlater Jet (Findlater, 1969)
axis drive this delay by modulating coastal stratification of the Arabian peninsula.

The strength of the OCC varies between 0.4-1.2 m/s during the SWM and attains its peak in the late summer (Lee et al.,
2000). Direct current ADCP measurements suggest that OCC is not coherent all along the coast: its direction frequently
changes in the south driven by eddy induced variability and predominantly southwestward in the north. As the mean current
of the OCC strengthens in the north, its transport reaches ~10 Sv at the northeastern coast of Oman close to the Ras al Hadd
(Lee et al., 2000). Subsequently, it turns offshore and meanders into the northern Arabian Sea to form the Ras al Hadd jet
or front (Böhm et al., 1999). This front advects cold and fresh coastally upwelled water offshore and thereby creates a large
SST gradient between the warm and saltier Gulf of Oman water and fresh and cooler Arabian Sea water and is visible clearly
in the satellite images (Fischer et al., 2002). Moreover, protruding capes in the southern part of the Oman and Yemen coast
generate topographically locked anticyclonic eddies close to the coast (Fagg and Kim, 1998; Lee et al., 2000). The non-
linear interactions between these eddies and OCC create filaments and squirts extending far offshore from the coast
(Manghnani et al., 1998; Wiggert et al., 2005). The width of these filaments is about 100 km with a current speed of ~0.5
m/s and transport ~2-6 Sv upwelled cold water offshore (Brink et al., 1998). This offshore transport by these filaments is
found to be much more than the estimated wind-driven offshore Ekman transport, suggesting possible offshore deflection
of alongshore mean currents which advect surface and upwelled subsurface water offshore. A part of this water then
recirculates back to the coast by the returning flow of the offshore gyres.





The first estimate of the intensity of upwelling along this coast was given by Smith and Bottero (1977) using hydrographic
observations and winds observed during 1963 under the first IIOE campaign. They estimated a vertical velocity of the order
of $2\times10^{-5}$ m/s with an upwelling transport of ~8 Sv through the 50 m depth along the 1000 km long coastline and from
the coast to 400 km offshore. Observations from the JGOFS cruises suggest that the upwelling signature on SST persists to
about 120 km offshore, whereas in the subsurface, upsloping of thermocline can be evident to about 260 km (Shi et al.,
2000). They find that, during the summer of 1995, the lowest SST recorded is 21oC close to the southern part of the Oman
coast in late August to early September, which upwelled from a depth approximately 100-150 m. However, note that the
coolest temperature is observed in the shelf of Oman, where SST starts to fall immediately with the onset of alongshore
winds and falls below 20°C in early July. The gradual increase of temperature away from the coast indicates that the
upwelling predominantly happens near the coast than offshore, where positive wind stress curl favors open ocean upwelling.
McCreary et al. (1996a) further noted that in the open ocean, offshore of the coast of Oman, their model-simulated vertical
velocity at the bottom of the mixed layer remains very small despite a large upwelling favorable Ekman pumping velocity.
This negligible vertical velocity is attributed to the state of Sverdrup balance via the radiation of Rossby waves. Therefore,
they advocated that the open ocean cooling off Oman and the associated biological response is primarily driven by advection
of cold nutrient-rich upwelled water from Oman coast and the wind-driven mixing entrainment at the bottom of the mixed
layer, which deepens the thermocline offshore.

This intense upwelling all along the coast of Yemen and Oman in the western Arabian Sea also drives one of the strongest
primary productivity of this region at a rate of more than 2.5 gCm$^{-2}$day$^{-1}$ (Marra et al., 1998; Morrison et al., 1998). Unlike,
Somali coast, here chlorophyll concentration is found to be more uniform and extend up to 400 km offshore.  The offshore
open ocean chlorophyll bloom is attributed to the advection of nutrient-rich upwelled water offshore by the several filaments
along the coast of Oman and wind-driven entrainment (McCreary et al., 1996; Wiggert et al., 2005).  A detailed discussion
on biophysical interactions along this coast is discussed in Section 2.7.5.

A major part of our understanding about the coastal upwelling off Oman is based on observations and modeling studies
carried out in 90s' and early 2000. Unfortunately, lack of observation and concerted modeling effort resulted in sluggish
progress of our understanding in the last couple of decades for this region. The dynamical reasons for the development of
offshore eddies and their impact on the coastal upwelling, coastal currents, SST, air-sea interactions, and finally, over the
biology is still not clear. Thus, considering the importance of this region in regional physical and ecological processes and,
most importantly its influence on the Indian Monsoon, a much focused effort is needed from the scientific community for a
complete understanding of oceanic processes of this coastal upwelling system.


**2.7 Upwelling along the West Coast of India**



The signatures of upwelling along the west coast of India begin to appear during March, peak during June, and weaken by
September. Though several studies have shown that upwelling could start between February and May, the surfacing of cold,
nutrient-rich water is more prominent during June-July.  Moreover, the upwelling is more intense along the southwest coast
of India than that along the northwest coast. For the remaining months, the sea level anomaly is positive, and the thermocline
is deeper, indicating conditions unfavorable for upwelling.  A major consequence of west coast upwelling is the formation
of anoxia that has a significant impact on the benthic ecosystem on the continental shelf (Banse 1959, Naqvi, 1991).
Although the upwelling along the west coast of India is weaker than that along the coast of Somalia, the region accounts for
70% of the Arabian Sea fish production (Luis and Kawamura 2004).
**2.7.1 Historical Background**
The earliest temperature observations along the west coast were collected by trading vessels and were confined to major
shipping routes. These observations were compiled into several atlases generated by different countries (Anonymous 1945,
1948, 1952). Though there were inconsistencies among the atlases, they showed the presence of cold water off the southwest
coast of India from June to October (Banse, 1959).  The decrease in temperature during the summer monsoon was also
evident in sea surface temperature (SST) data shown by Sewell (1929) along the southwest coast of India and Lakshadweep.
However, it was difficult to attribute this decrease in temperature to upwelling as the SST could also be controlled by other
factors like atmospheric fluxes, horizontal advection, or mixing.  He showed that SST increased during April-May when the
boreal summer is at its peak and dropped during June-July when the Monsoon picks up. After the Monsoon, the second
oscillation started: the temperature picked up again and dropped during the boreal winter when the winds were cooler.
Naturally, using available observations, Sewell (1929) linked the double oscillation of SST to air temperature change.
The first evidence for upwelling along the west coast of India was presented by Sastry and Myrland (1959); they showed
that the isotherms tilted upwards all along the southwest coast of India.  Both Sastry and Myrland (1959) and Banse (1959)
argued that the upwelling along the southwest coast of India is not completely driven by monsoon winds because the fall in
SST occurred in April-May, which is a month before the onset of the summer monsoon. They hypothesized that the
prevailing current-system caused the upward tilting of isotherms. The reversal in the West India Coastal Current (WICC)
appeared to coincide with the beginning of upwelling at the southern tip of India. Banse (1959) suggested that after the onset
of Monsoon, the winds could intermittently push the cold water to the surface. Banse (1959) further noted that the poorly
aerated bottom water on the shelf during the summer monsoon was linked to upwelling that takes place along the coast.
Hydrographic sections in the decades that followed showed that the upwelling signatures extended all along the west coast
of India and Pakistan (Banse 1968; Sarma, 1968; Ramamirtham and Rao, 1973) and revealed that upwelling sets in earlier
in the south and progresses slowly towards the north (Sharma, 1968, Longhurst and Wooster, 1990). Due to the boisterous
nature of monsoon winds, upwelling along the west coast of India was still considered to be driven by alongshore winds



(Ramamirtham and Rao, 1973). The role of wind in driving the upwelling was disputed again by Sharma (1978), using the
available wind data from the atlases that showed that the wind was onshore and poleward and not favorable for upwelling
during the summer monsoon. Notwithstanding, recent wind data sets show that the alongshore winds are not poleward but
equatorward (but weak) during the summer monsoon.
Johannessen et al. (1981) used an extensive data set (consisting of 1500 Nansen casts collected from 1971-1975) and
confirmed the upwelling features highlighted in previous studies. The seasonal upwelling was found to repeat every year,
albeit with a certain amount of variability. Upwelling signatures were not evident in salinity but in temperature and oxygen
data.  The upwelling process also increased phytoplankton and zooplankton production.  However, no such correlation was
evident for the higher trophic level of the food chain.  The calculated rate of upwelling was around 1.5 m/day, which was
consistent with the earlier observations.
The "wind-driven upwelling" hypothesis was again invoked in the mid-eighties.  Shetye et al. (1985) used a more recent
wind product (Hasternath and Lamb, 1978) and found that offshore Ekman transport, though weak, peaked during the
summer monsoon. Using ship-based observations, Shetye et al. (1990) confirmed that the upwelling intensity weakened
from south to north. The width of the surface current, which is related to the upwelling, extended about 150 km from the
coast. The signatures of upwelling were evident only in the top 100 m below which there were signatures of downwelling,
indicative of an undercurrent. They refuted the "current-induced upwelling" hypothesis using regression analysis between
Ekman transport and temperature gradient.
In the nineties, numerical models were used to provide insight into the seasonal cycle of north Indian Ocean circulation
(McCreary 1993; Shankar et al., 1996; McCreary et al., 1996; Vinayachandran et al., 1996; Shankar and Shetye 1997).
Using linear wave theory, McCreary et al. (1993) proposed that the upwelling along the west coast of India was primarily
driven by coastal-trapped waves generated by remote winds from the Bay of Bengal (see section 2.7.3 for details).  The
wind-driven upwelling was weaker than that caused by the propagation of these waves.
The dominance of these waves suggests that upwelling indices based on Ekman theory (Pankajakshan, 1997; Bakun et al.,
1998) do not provide a complete picture of coastal upwelling along the west coast of India. The weak alongshore winds,
however, would still contribute to upwelling and cannot be neglected (Shankar 2002; Suresh et al., 2013). It is to be noted
that, unlike the west coast of India, the southern tip of India is unique in the sense that the Findlater jet is parallel to the coast
and causes strong Ekman Transport (Bakun et al., 1998; Smitha et al. 2008). The wind-induced coastal upwelling index here
was almost five times that observed along the southwest coast of India (Bakun et al., 1998). The study also hypothesized
that the strong upwelling near the southern tip could generate coastal-trapped waves that could propagate along the west
coast of India.





### 2.7.2 Observations


The double oscillation, as observed by Sewell (1929), is evident in the SST climatology from the satellite data (**Figure 14a,**
**15a**). The temperature peaks during April and is highest along the southwest coast of India.  The area surrounding the
southwest coast of India, where the temperature remains above $30^{o}$C, is often referred to as the Arabian Sea mini warm pool,
and this region is thought to play an essential role in the onset of the summer monsoon (Vinayachandran and Shetye,
1991;Rao and Sivakumar, 1999;  Shenoi et al., 2005; Kurian and Vinayachandran, 2007; Vinayachandran et al., 2007).  The
increase in temperature is attributed to air-sea fluxes and is independent of the SST changes observed during the winter
monsoon.  The temperature begins to drop after April and is the lowest during July and August. The drop in temperature
starts in the south and progresses northwards within a month. The progression of SST towards the north, also observed in
hydrography data (Sarma, 1968 and Longhurst and Wooster, 1990), could be linked to the poleward propagation of coastal
Kelvin waves.  A typical first or second baroclinic mode Kelvin wave would cover the entire west coast of India within 7-
21 days (these waves are sometimes too fast to be detected by a satellite over a small domain).
As the satellite SST only permits the assessment of near-surface processes, we use the *North Indian Ocean Atlas* (Chatterjee
et al., 2011) to identify the variations in the thermocline along the west coast of India (**Figure 14**). The atlas includes data
from the Indian Exclusive Economic Zone (EEZ) and has more stable values in this region compared to the *World Ocean*
*Atlas* (Antonov et al., 2010; Locarnini et al., 2010). Along the southwest coast of India, the isotherms tilt upwards by April
(**Figure 14b-c**). By June, the cooler water starts touching the surface, and the upwelling becomes more intense by July and
August. The isotherms start lowering by September-October, and surface waters become warmer.  In the north, the surface
layers are cooler during April, and the downwelling of isotherms persists till June. The isotherms begin to rise by July-
August, but they are very weak compared to the southwest coast of India. Unlike the SST, the depth of $26^{o}$C isotherm shows
an annual cycle. The depth decreases during summer and increases during winter. The lag associated with poleward
propagation of the Kelvin wave is also evident in the isotherm depth (See Figure 7 in Shah et al., 2015 and Figure 6 in
Shankar et al., 2019). The larger width of the upwelling region in the south is also indicative of the westward propagation
of Rossby waves, whose westward phase speed decreases with latitude. The westward drift of chlorophyll along with Rossby
waves is evident in the satellite data but is not as prominent as seen during the winter monsoon (Amol, 2018; Amol et al.,

903    2020).

Since wind is the primary driving factor for upwelling around the world, it is essential to look at its behavior along the west
coast of India. Although the monsoon winds are strong (**Figure 15d,e**), they mainly blow perpendicular to the coast. The
alongshore component of the wind is weak and equatorward all around the year.  The winds peak during July along the
entire west coast and this increase in the magnitude of the alongshore wind intensifies the upwelling during the summer
monsoon. It is only at the southern tip of India that the alongshore winds reverse with the season. Upwelling indices
calculated using wind data show that the upwelling along the west coast of India is weak compared to the upwelling around





Somalia, Oman, Tanzania, and south Madagascar but is equivalent to that in Mozambique and west Madagascar (Bakun et
al., 1998). As the alongshore winds are weak compared to the cross-shore, there is also an ambiguity in the direction of the
wind reported by previous authors (Sarma, 1978; Shetye et al., 1985;Shah et al., 2015). For example, Shah et al., (2015)
showed that the alongshore winds were equatorward during the summer monsoon, but only south of 17°N. The difference
here lies in the angle of rotation applied to compute the alongshore component of wind. Shah et al. (2015) used coastline
angle, which is almost parallel to the longitude in the north. The wind vectors in **Figure 15** are pointing northeast, which
would lead to a poleward wind when rotated based on coastline angle. The slope angle, which is used to compute the
alongshore wind in **Figure 15**, is different from the coastline angle because of the widening of the continental shelf north
of 15°N (see 1000 m contour in **Figure 15a**).
Unlike the alongshore wind, which is unidirectional, the alongshore WICC and the sea level anomaly show a strong seasonal
cycle (**Figure 15,f**). The current (sea level anomaly) is equatorward (low) during summer and poleward (high) during winter.
The reversal in current follows the drop in sea level, and the flow is poleward in March, which is much before the onset of
the summer monsoon. The early reversal of current is also evident in direct current measurements (Amol et al., 2014;
Chaudhuri et al., 2020). Unlike the sea level, the currents, particularly along the southwest coast of India, have a significantly
larger intraseasonal component (Vialard et al., 2009; Amol et al., 2014; Chaudhuri et al., 2020). The current could flow in
either direction during a particular time of a year, and the frequent intraseasonal bursts would further make it difficult to
predict its direction. Still, it was the early reversal of current during March that prompted Banse (1959) to discard wind as
the driving factor for upwelling.
In response to the raising of the isotherms, chlorophyll also increases from April onwards (**Figure 15c**). The chlorophyll
concentration is highest along the southwest coast of India and peaks during July-August. During this time, the wind, the
sea level, and the SST are at their peak phase as well. The increase in chlorophyll concentration is weakest along the central
west coast of India and only extends over a few months. In the north, the chlorophyll is high all around the year because of
the winter convective mixing that follows the upwelling in the summer.
**2.7.3 Modelling and mechanisms**
McCreary et al. (1993) used a series of reduced-gravity model experiments to show that the upwelling along the west coast
was driven by remote forcing. They concluded that winds in the Bay of Bengal and the equator caused upwelling along the
west coast of India. The local winds, however, enhanced upwelling, but their contribution was weaker than that by remote
winds. They noted that the driving mechanism for upwelling was the generation of coastal Kelvin waves by winds along
the western boundary of the Bay of Bengal. These winds generated upwelling Kelvin waves that propagated equatorward
(with the coast on the right) along the east coast of India, turned around Sri Lanka, and propagated poleward along the west
coast of India. The poleward propagation explained why the upwelling is delayed in the north. Shankar and Shetye (1997)



further highlighted the mechanism for the early onset of upwelling using an analytical model. They showed that the
upwelling along the west coast of India and the Lakshadweep low formed in the southeastern Arabian Sea resulted from
poleward propagation of Kelvin waves and westward radiation of Rossby waves, which supported the results shown by
McCreary et al. (1993).
Modeling studies for upwelling using ocean general circulation models are very few. Haugen et al. (2002) used a 5-6 km
resolution coastal model that was nested into large-scale models. Their simulations were consistent with the observations
shown by Johannessen et al. (1981). The model upwelling began during April and persisted till October, and was most
intense along the southwest coast of India. Models run by Rao et al. (2005) and Rao et al. (2008) had restricted model
domains, which would not accurately represent the role of remote forcing on upwelling.
Lévy et al. (2007) used both satellite chlorophyll and a physical model to assess the timing of the phytoplankton bloom and
its related dynamics. They showed that the onset of summer blooms in the southeastern Arabian Sea occurred during March
and was primarily driven by upwelling. The horizontal currents had a limited role in driving the blooms. Koné et al. (2013)
arrived at the same conclusions using a biophysical model. They further showed high values of NO3-that were associated
with the vertical advection in this region.
A more recent and extensive study shows that differences in the strength of upwelling between north and south could affect
the nature of fisheries along the coast of India (Shankar et al., 2019). In the south, stronger upwelling permits the growth of
larger phytoplankton owing to a greater supply of nutrients, whereas in the north, phytoplankton tends to be smaller in size
owing to weaker upwelling. The large phytoplankton is directly fed by planktivorous fishes that are not common in the
north.
In summary, model simulations show that the upwelling is primarily driven by poleward propagation of coastal Kelvin
waves. The linear wave theory explains the early onset of upwelling and the progression of upwelling from south to north.
The alongshore winds also favor upwelling and could contribute significantly to its variability along the coast. A detailed
analysis using observations and numerical models would be required to delineate the relative contribution of the wind and
large-scale waves during the peak of the summer monsoon.
**2.7.4 Biogeochemistry**
Unlike most parts of the world ocean, the biophysical provinces of the Indian Ocean vary seasonally (Rixen et al., 2020;
Lévy et al., 2007). This is because during both the monsoons, the underlying mechanisms for nutrient intrusion that support
elevated primary productivity are different during summer and winter. During summer, there is strong coastal upwelling,
while cooler and dry air from the northern Indian subcontinent drives convective mixing in the Arabian Sea during winter
(Madhupratap et al., 1996).



One of the most fascinating biogeochemical aspects on the west coast of India has been the seasonal occurrence of two phytoplankton blooms of the different phylum. First, there are winter mixing driven blooms of *Noctiluca* scintillans (hereafter *Noctiluca*), a mixotrophic dinoflagellate that occur during winter in the northern Arabian Sea (Prakash et al., 2008; do Rosario Gomes et al., 2008; Rixen et al., 2020). And almost at the same time, there are massive cyanobacterial blooms of *Trichodesmium* ($N_2$ fixers) in the central – eastern Arabian Sea during March-May until summer monsoonal turbulence brings it down (Capone et al., 1998; Gandhi et al., 2011; Kumar et al., 2017).

The occurrence of *Noctiluca* blooms was first discovered in the early part of this century and seemed to have displaced the previously occurring diatom blooms in this region (do Rosario Gomes et al., 2008; Sarma et al., 2018). These blooms create a biogeochemical divide – making the northern Arabian Sea more productive than its southern part (Prakash et al., 2008). These massive outbreaks of *Noctiluca* blooms were reported to be fueled by an unprecedented influx of oxygen-deficient waters into the euphotic zone (do Rosário Gomes et al., 2014). However, such claims have been refuted (Prakash et al., 2017) and proved that they are naturally driven by changes in nutrient stoichiometry (Lotliker et al., 2018; Sarma et al., 2018).

Once nutrients supply driven by the winter mixing is consumed and the ocean begins to stratify, *Trichodesmium* blooms start to appear by early spring in the central Arabian Sea (Capone et al., 1998; Mulholland and Capone, 2009). These become so massive in the eastern Arabian Sea that they fix up to 34 mmol N m$^{-2}$ d$^{-1}$, which is the highest reported rate of $N_2$ fixation ever among the world oceans (Gandhi et al., 2011; Kumar et al., 2017). In fact, when similar conditions prevail during fall intermonsoon immediately after the summer monsoon upwelling, the $N_2$ fixation rate makes a surplus contribution to the nitrogen-nutrients to fuel primary production in the eastern Arabian Sea (Singh et al., 2019).

$N_2$ fixers are associated with excess phosphate (compared to NO3- if normalized as per the Redfield Ratio) concentration (Deutsch et al., 2007). Summer upwelling of oxygen-deficient waters along the shelf break is the major process regulating the biogeochemistry on the west coast (Gupta et al., 2016). Summer upwelling, driven by high primary production, is followed by the occurrence of denitrification (a nitrogen loss process) at subsurface layers (500-2000 m) in the eastern Arabian Sea (Naqvi et al., 2000), which would make these layers phosphate-rich. Hence, in this cycling, the upwelling would intrude phosphate-rich water to the sea surface (Sudheesh et al., 2016). The notion is that once parts of upwelled nutrients are utilized by autotrophs in the sunlit layers, it will create a niche for $N_2$ fixers. However, recent studies suggest that $N_2$ fixers can also occur in eutrophic conditions (Landolfi et al., 2015).

*2.7.5 Variable impact of coastal upwelling on the biogeochemistry of east and west coasts of Arabian Sea*

There are distinct differences in the observed biogeochemical processes between eastern and western upwelling regions of the Arabian Sea. The major differences are:





(1) Along the western Arabian Sea, the upwelling is more vigorous, with the surface temperature reaching as low as 16°C (Swallow and Bruce, 1966). These waters are enriched with macronutrients (the near-surface $NO_3^-$ recorded up to 15-20 µM (Smith and Codispoti, 1980; Morrison et al., 1998), which triggers large phytoplankton blooms; these upwelled waters transport quickly to the offshore due to strong Ekman flow. This leads to the extent of upwelling induced fertilization and the high phytoplankton bloom to a distance exceeding ~1000 km offshore (Naqvi et al., 2003, 2006).

(2) The eastern part is net heterotrophic whereas its western counterpart is net autotrophic (Sarma, 2004) due to the upwelled waters in the western Arabian Sea having a high initial oxygen concentration, and a vigorous upwelling and narrow shelves do not allow the upwelled waters to stay long enough to develop intense hypoxia (Naqvi et al., 2010).

(3) Unlike the eastern Arabian Sea, the influence of freshwater is very weak in the western Arabian Sea, and therefore the nearshore regions are not strongly stratified. This leads to sufficient ventilation to the sub-pycnocline waters to a more oxygenated condition.

The progression of upwelling over the eastern Arabian Sea is slow, and the upwelled waters sustain 4-6 months over the shelf (Gupta et al., 2016); i.e., a wider shelf over the eastern Arabian Sea allows the upwelled waters to persist long enough till the oxygen is completely utilized and seasonally cover the entire shelf (area ~200,000 km$^2$), making it the largest shallow water oxygen-deficient zone in the world (Naqvi et al., 2000). The intensely oxygen-depleted environment favors the development of diverse microbial populations that utilize anaerobic pathways to derive energy, mediating elemental transformations that are of immense geochemical significance (Wright et al., 2012). Denitrification is one of the classical examples for this kind, which makes the eastern Arabian Sea upwelling system one of the 'hot spots' of $N_2O$ production in the world ocean with $N_2O$ saturations up to 8250% (Naqvi et al., 2005). Moreover, these upwelling regions are also characterized by a high production of other climate-relevant trace gases such as $CH_4$ and dimethyl sulfide (Naqvi et al., 2010; Shenoy et al., 2012). Further, spring *Trichodesmium* blooms seem to be responsible for the emission of volatile organic compounds, such as isoprene – a precursor of ozone formation in the troposphere, in the eastern Arabian Sea (Tripathi et al., 2020a,b). The upwelling biogeochemistry of this seasonal oxygen-deficient zone also significantly impacts the cycling of several other micronutrients, like manganese, iron, etc. (Breitburg et al., 2018).

The variability in magnitude and intensity of upwelling and the characteristics of upwelled waters play a major role in shaping biogeochemistry of the eastern Arabian Sea shelf that designates it as a 'Hotspot' for greenhouse gas production during the summer monsoon. The upwelled waters are hypoxic in the south (Gupta et al., 2016) and suboxic in the central-eastern Arabian Sea (Sudheesh et al., 2016). This combines with strong thermohaline circulation leading to high oxygen demand over the central shelf, relative to the south, making the region extremely oxygen-depleted and sulphidic with $H_2S$ levels goes up to ~15 µM in the nearshore waters (Naqvi et al., 2006, 2009). While the hypoxic upwelling over the southern shelf restricts the denitrification to sediment, the anoxic/sulphidic conditions over the central shelf extend its occurrence to



the water column as well (Sudheesh et al., 2016). Though the waters over the western Arabian Sea shelf are net autotrophic
(Sarma, 2004), moderately low surface pH (<7.9) was recorded during the summer monsoon (Takahashi et al., 2014). Being
limited with very few studies on carbon dynamics over both east and west coasts, the temporal evolution of surface ocean
acidification is still not clear, although Kanuri et al. (2017) reported $p$CO$_2$ up to 630 µatm along the southeastern Arabian
Sea shelf and even levels exceeding 1000 µatm are common during peak upwelling (Sudheesh, 2018). Refuting the charges
levied by huge productions of CO$_2$ and N$_2$O, massive methane loss through anaerobic oxidation by sulphate in the nearshore
waters of the eastern Arabian Sea during late summer monsoon upwelling (Sudheesh et al., 2020) is a great relief to the
environment as the potential greenhouse effect is naturally diluted by converting methane to CO$_2$ (the latter is almost 300
times less potential compared to former).
On comparable lines of intensification of oxygen deficiency in the western Arabian (Piontkovski and Al-Oufi, 2015), eastern
Arabian Sea shelf was also earlier reported with such intensification (Naqvi et al., 2000, 2006), but the comparison of
monthly studies for one year in 2012 with similar data set from July 1958 to January 1960 (Banse, 1959) revealed remarkably
little change in oxygen concentrations (Gupta et al., 2016; Figure 16) with inter-annual variations between the years
supported by global climatic events such as IOD, ENSO, etc., as these warm years impact upwelling intensity and prevents
the anoxia formation on the shelf (Parvathy et al., 2017).
The productivity of the western Arabian Sea has earlier been shown to increase over the years (Goes et al., 2005) due to the
warming of the Eurasian landmass, but such a trend was not discernible over the eastern Arabian Sea (Prakash and Ramesh,
2007) as neither wind speeds nor SSTs could show any significant change. Although no information available on such recent
trends, the dissolved oxygen concentrations of recent years were comparable with that of five decades ago over the southwest
coast of India (Gupta et al., 2016) despite the period gained significant developmental activities on the hinterland co-
occurred with a steep rise in Arabian Sea warming. In the absence of climatology data, the maintained dissolved oxygen
levels can be considered as a proxy to show the sustained upwelling intensity and biogeochemistry of this region. Further,
the upwelling intensity and consequent biological production over its eastern part are several-fold less than the western
region. Yet, the famous and thickest Arabian Sea oxygen minimum zone (OMZ) is closer towards the eastern side,
underlying the importance of circulation in OMZ formation and source water characteristics for upwelling induced primary
production. Though the upwelling over both east and west coasts is progressed from south to north during the summer
monsoon, the coast of Somalia is pronounced with significant gradients in biological production – several folds high in the
north during the advanced phase of summer monsoon when nutrients from local upwelling as well as advected from south
support enhanced growth of phytoplankton (Lakshmi et al., 2020). In contrast, the productivity of eastern upwelling is higher
in the south due to relatively intense upwelling compared to its north (Gupta et al., 2016; Shankar et al., 2019). Though
upwelling over the west coast is much intense, it never experienced strong oxygen-depleted conditions, unlike its east coast.
The strong biological pump (Ekman transport) operating from the west coast transports organic matter too far off distances



beyond the central Arabian Sea and pushes the OMZ towards the east (Naqvi et al., 2003, 2006). Being closer, these OMZ
waters feed the eastern Arabian Sea upwelling and develop hypoxic/anoxic conditions there (Gupta et al., 2016; Sudheesh
et al., 2016, 2020). The upper 300m water column of the western Arabian Sea (Oman region) has witnessed warming by
~1.5°C from 1960 to 2008; it lost dissolved oxygen by ~1 ml L$^{-1}$ (at 100m) and became near anoxic with oxycline shoaled
at ~19 m per decade during this period (Piontkovski and Al-Oufi, 2015). While it was hypothesized that the upper ocean
warming reduces ocean mixing and biological production in the western Arabian Sea (Roxy et al., 2016), it was quickly
refuted as a northward shift in monsoon low-level jet can orient the wind angle to the Oman coast in such a way that the net
upwelling increases and so the primary production (Praveen et al., 2016). In the scenarios of such increasing upwelling and
shoaling of oxycline, if more deoxygenated/near anoxic waters upwell, it may turn the future of the west coast comparable
to the present-day east coast in terms of biogeochemistry under seasonal hypoxia/anoxia.
The impact of upwelling on oxygen concentration has a profound socio-economic impact too as it directly affects living
resources and biodiversity (Panikkar and Jayaraman, 1966; Naqvi et al., 2006). Though the available information from the
Arabian Sea is scanty, the mesopelagic fish populations appear to be impacted by a reduction in suitable habitat as respiratory
stress increases due to deoxygenation (Naqvi et al., 2006). Benthic ecosystems along the eastern Arabian Sea affected worst
owing to the unusually large area of continental margins being exposed to hypoxic/anoxic waters (Helly and Levi, 2004).
During this period, the density and diversity of larger benthic fauna (prawns, crabs, mollusks, etc.) become insignificant,
and groups that are sensitive to hypoxia, like echinoderms, are either absent or least abundant (Parameswaran et al., 2018).
However, macro-infaunal communities are overwhelmingly dominated by deposit-feeding opportunistic polychaetes,
particularly the proliferation of juveniles (Abdul Jaleel et al., 2015). The upwelling region of the Arabian Sea is a major
ground for fishery potential in terms of their eggs laying and recruitment succession. The upwelling induced high primary
production supports higher trophic level productivity but with less biodiversity. It is found that upwelling intensity and
coastal currents during summer monsoon are highly influencing fish eggs transport, their recruitment success rate, and
juveniles transport.
**2.8 South Coast of Sri Lanka**
The upwelling off the southern coast of Sri Lanka (which is slightly tilted towards the north in the east) begins with the
onset of the SWM, during the last week of May or during the first of June, and continues through October.  The coastal
upwelling here is primarily caused by summer monsoon winds, which have a strong alongshore component along the
southern   coast of Sri Lanka (Vinayachandran et al., 2004). The SMC that flows eastward to the south of Sri Lanka and
northeastward to the east of Sri Lanka (Vinayachandran et al., 1999, 2018; Webber et al., 2018; Rath et al., 2019) influences
the advection of cold upwelled water. During the early part of the SWM, some advection of cooler water occurs towards the
south, away from the coast. During the later half, most of the upwelled water flows into the BoB along with the SMC



(Vinayachandran et al., 2004; Das et al., 2018; Vinayachandran et al., 2020). Numerical simulations have successfully
reproduced the upwelling along the southern coast of Sri Lanka (Vos et al., 2014).

Satellite-derived chlorophyll data (**Figure 17)** during summer monsoon clearly shows that the coastal upwelling has a clear
expression on the biogeochemistry (Vinayachandran et al., 2004; Vinayachandran 2009). The chlorophyll concentration is
high near the coast in response to upwelling. In addition, the advection by SMC spreads water from near the coast towards
the east of Sri Lanka, impacting a larger region. In situ sampling to quantify the physical and biological impacts of upwelling
around Sri Lanka is yet to take place. The physical impact on the ecosystem in this upwelling zone is complex, owing to the
simultaneous influence of multiple factors.  The upwelled water advects to the southern coast of Sri Lanka from the southern
tip of India and the Gulf of Mannar. There is additional advection along the path of the SMC. Finally, the currents along the
east coast of Sri Lanka is southward during summer, being the eastern arm of the cyclonic gyre, associated with Sri Lanka
Dome (SLD, Vinayachandran and Yamagata, 1998). There are indications from model simulations that the pCO2
distribution is impacted by the combined influence of upwelling and advection (Chakraborty et al., 2018). On the whole,
satellite-derived SST and chlorophyll data clearly show an active upwelling zone along the southern coast of Sri Lanka,
which draws out a definite response from the ecosystem and biogeochemistry.

Using shipboard observations, Jyothibabu et al. (2015) suggest that capping of the upper layer by low salinity water in this
region can restrict the chlorophyll concentration in the near-surface layers.  Using the glider data set, Thushara et al. (2019)
has provided in situ observational evidence for the chlorophyll blooms associated with SLD. The observed bloom followed
a period of Ekman suction caused by cyclonic wind stress curl, and the decay was caused by the arrival of Rossby waves
from the east. Model simulations (Thushara et al., 2019) support these processes and suggest that the Ekman pumping is
capable of enriching the euphotic zone with nutrients, but there is a lack of corresponding in situ observations that are much
needed to validate these processes.

**2.9. East Coast of India**
**2.9.1 Physical Processes**
Circulation in the Bay of Bengal (BoB) is driven by a rather intricate combination of local winds over the BoB and remote
forcing originating from the equatorial Indian Ocean. During the southwest monsoon, strong southwesterly winds along the
western boundary of the BoB (WBoB) makes conditions favorable for coastal upwelling (Shetye et al., 1991; Shankar et al.,
1996; McCreary et al., 1996; Vinayachandran et al., 1996; Shankar et al., 2002; Thushara and Vinayachandran, 2019). The
winds are northeasterly during the northeast monsoon, which is favorable to coastal downwelling (Shetye et al., 1996). BoB
is also known for high SST (average temperature greater than 28 C) (Vinayachandran and Shetye, 1991; Shenoi et al., 2002)
and the formation of several low-pressure systems (Sikka 1980; Gadgil et al., 2004). A significant amount of freshwater
influx from major river sources like Ganga, Brahmputra, etc., plays a dominant role in stratifying the upper layer affecting





the strength and intensity of coastal upwelling (Vinayachandran et al., 2002; Behara and Vinayachandran, 2016; Thushara
and Vinayachandran, 2016). Additionally, coastal processes in the WBoB are influenced by complex bathymetry, shallow
mixed layer, the formation of mesoscale eddies, and propagations of large-scale planetary waves (Mukherjee et al., 2017).
Detailed description of the East India Coastal Current (EICC) and its variability, based on moored observations, are given
in Mukherjee et al. (2014) and Mukhopadhyay et al. (2020).

Unlike the Arabian Sea, which experiences intense upwelling along the Somalia and Kerala coasts in the western and
southeastern coasts, respectively, during the summer monsoon, the Bay of Bengal experiences feeble upwelling near India
and Sri Lanka (Shetye et al., 1991; Vinayachandran et al., 2004). The first evidence of coastal upwelling along WBoB was
observed between 1952–1965, during IIOE. The first published report, although insufficient to present evidences of coastal
upwelling or downwelling for a season, along WBoB using hydrographic data, was by La Fond (1954, 1957, 1958, 1959).
Evidence of coastal upwelling during summer monsoon along the east coast of India was reported by Varadachari (1961).
Murty and Varadachari (1968) found stronger upwelling at Visakhapatnam compared to Chennai during both spring and
summer seasons. Similarly, upwelling at the northern part of the east coast of India (**Figure 18**) was also reported by several
investigators (Murty, 1958; Murthy, 1981; Gopalakrishna and Sastry, 1984; Rao et al., 1986, Shetye et al., 1991).

Satellite altimeter data show that mesoscale eddies (both upwelling (cyclonic) and downwelling (anticyclonic) favorable)
play a dominant role in causing coastal upwelling/downwelling in the BoB (Ali et al., 1998; Gopalan et al., 2000; Chen et
al., 2012; Nuncio and Kumar, 2012; Cheng et al., 2013; Mukherjee et al., 2019). However, the vertical structure of mesoscale
eddies along WBoB is still unknown due to the lack of appropriate in-situ measurements. As cyclonic eddies upwell cold
water from its lower base to upper depth and enhance vertical mixing (Falkowski et al., 1991; Kumar et al., 2004), and
vertical structure of eddies affect the strength of upwelling and associated transport of heat, salt, and nutrients in the ocean
(Chaigneau et al., 2011; Dong et al., 2014), it is required to understand the role of eddies in the upwelling along the east
coast of India.

Model simulations that began in the 1990s to investigate EICC found local wind-driven coastal upwelling along WBoB
during the summer season compared to spring and northeast monsoon (McCreary et al., 1996; Shankar et al., 1996;
Vinayachandran et al., 1996). During spring, seasonal sea level variability along WBoB is dominated by remote forcing that
originating from interior Ekman pumping in the BoB, the equatorial Indian Ocean, and alongshore wind along the eastern
and northern boundary of the BoB (McCreary et al., 1996; Vinayachandran et al., 1996; Aparna et al., 2012; Mukherjee et
al., 2017). During the winter, seasonal coastal downwelling occurred due to the northeasterly winds (McCreary et al., 1996;
Shetye et al., 1996). Based on satellite and in-situ observations and models, Shankar et al. (2002) showed that the dynamics
of sea level and associated upwelling along WBoB at seasonal time scales could be explained using linear theory.




At interannual time scales, dynamics of sea level and associated upwelling are dominated by El Niño-Southern Oscillation
(ENSO) and Indian Ocean Dipole (IOD) (Saji et al., 1999). During ENSO and IOD events, interannual variability of sea
level is influenced by remotely propagating waves from the equatorial Indian Ocean (Clarke and Liu, 1994; Rao et al., 2009;
Aparna et al., 2012; Mukherejee and Kalita, 2019). At intraseasonal time scales, coastal upwelling or downwelling is
dominated by mesoscale eddies formed due to instability of the ocean (Nuncio and Kumar, 2012; Chen et al., 2012; Cheng
et al., 2013; Mukherejee et al., 2017).

Recent studies also showed that Andaman and Nicobar Islands (ANIs) play a dominant role in the dynamics of sea level and
associated upwelling along WBoB (Chatterjee et al., 2017; Mukherjee et al., 2019) by influencing the wave propagation.
While propagating in the interior BoB, the Rossby wave is significantly modified in the presence of ANIs (Chatterjee et al.,
2017) and generates coastal upwelling by the formation of mesoscale eddies in the WBoB (Mukherjee et al., 2019). Another
significant force for modifying coastal upwelling comes from freshwater discharge by rivers (Behara and Vinayachandran,
2016). Owing to the presence of fresh river water, barrier layer formation is common in the northern Bay of Bengal
(Vinayachandran et al., 2002), which has the potential to weaken upwelling (Behara and Vinayachandran, 2016). However,
the impact of river runoff inhibits upwelling only towards the end of the summer monsoon (**Figure 19**), and the local winds
sustain upwelling for most of the summer monsoon (Thushara and Vinayachandran, 2016)

In summary, coastal upwelling along WBoB is not simply local wind-driven but affected by several oceanic processes,
which includes mesoscale eddies, remote forcing from equatorial Indian Ocean & interior BoB, freshwater forcing from
rivers, etc. At the seasonal time scale, coastal upwelling along WBoB is dominated by linear processes either by local wind
or remotely propagating waves. At interannual time scales, sea level variability along WBoB is dominated by remotely
propagating waves from the equatorial Indian Ocean. At intraseasonal time scales, mesoscale eddies dominate sea level
variability. More in-situ observations are necessary in order to understand the vertical structure of coastal upwelling at
intraseasonal, seasonal, and interannual time scales. Additionally, ocean models need to be better parameterized for
resolving vertical processes near the coast related to mixed layer, thermocline, barrier layer, vertical stratification, etc, based
on in-situ observations.

**2.9.2 Ecosystem Response & Biogeochemical impacts**
*Biogeochemistry of the Bay of Bengal:* Despite being situated at similar latitudes and experiencing similar monsoonal force,
the Bay of Bengal is a low productive basin compared to the Arabian Sea. The large influx of freshwater leads to a strong
salinity gradient that stratifies the upper layer and also leads to the formation of the salinity-driven "barrier layer,"
particularly during the peak discharge season in the summer monsoon (Vinayachandran et al., 2002). This restricts
entrainment of nutrients into the upper sunlit layer. The inorganic nutrient (nitrate and phosphate) transport through rivers



draining into the bay is also abysmal (Sengupta et al., 1981; Sengupta and Naqvi 1984). The salinity driven stratification is
so strong that monsoonal winds are unable to erode them and inject nutrients from the subsurface layer. The surface
chlorophyll concentration is therefore, always low in the Bay of Bengal.  However, the basin is characterised by the perennial
presence of sub-chlorophyll maximum (SCM), which is located at 40-90 m depth (Prasanna Kumar et al., 2007; Thushara
et al., 2019). Cyclonic old-core eddies, which are predominantly present in the Bay of Bengal, do pump nutrients into the
upper layer and can enhance the productivity by more than two-fold (Prasanna Kumar et al., 2007; Singh et al., 2015).
Anticyclonic eddies, on the other hand, recharge the subsurface layer with dissolved oxygen and restrict the strengthening
of the OMZ. Episodic atmospheric disturbances such as depressions and cyclones also erode the stratification by churning
up the ocean and inject nutrients into the upper sunlit layer to fuel productivity (Gomes et al., 2000; Vinayachandran and
Simi 2003; Sarma et al., 2013; Vidya et al., 2017).
One of the most intriguing aspects of the BoB is that its organic carbon export fluxes are comparable to that of the Arabian
Sea though the bay is less productive (Ittekot et al., 1991; Lee et al., 1998). Various theories have been proposed to explain
observed high fluxes in the bay, such as the ballasting effect due to high terrigenous input by the river (Ittekot et al., 1991),
low respiration rates (Naqvi et al., 1996), high new production (Sanjeev Kumar et al., 2004) to list a few. The ballasting
effect helps flux material to coagulate, enhances the particle sinking rate (Ramaswamy et al., 1991), and decreases the
remineralisation in the upper layer, thereby decreasing the biological oxygen demand in this layer. These all collectively
impact the dissolved oxygen concentration in the bay and does not make it denitrifying.
*Upwelling in the Bay of Bengal:* Though feeble, the upwelling in the Bay of Bengal does drive regimes of high productivity
in the southwestern region during the Southwest monsoon (Vinayachandran et al., 2004) and in the northeastern region
during the northeast monsoon (Vinayachandran, 2009).  The nitracline, usually situated at a depth of ~75 m, below the
stratified layer, shoals upward by poleward flowing EICC during the pre-southwest monsoon and enhances productivity
(Gomes et al., 2000). But despite nutrient enrichment due to upwelling, the highest productivity was found in the eddy
region along the coast during the pre-monsoon. During the post-monsoon, although the wind-driven upwelling and river
discharge  increased the column chlorophyll concentration by nearly five-fold, the productivity decreased to half (Gomes et
al., 2000) due to light limitation.
*OMZ in the Bay of Bengal:*  Like the Arabian Sea, the Bay of Bengal also hosts a thick oxygen minimum zone in its core
despite having low surface production. However, unlike the Arabian Sea, which has significant seasonal variability in the
OMZ, the Bay of Bengal OMZ is seasonally stable but has significant short-term temporal variabilities (Johnson et al.,
2019). The high temporal variability can be attributed to the supply/injection of oxygen to the deeper layer by mesoscale
eddies and sporadic cyclonic activities (Sarma and Bhaskar, 2018; Joshnson et al., 2019), which can help break the surface
layer stratification. The freshwater driven stratification has profound effects on the oxycline variability as well. Prakash et
al. (2013) showed, using Argo-Oxygen data, that the oxycline in the Bay of Bengal is shallower than the Arabian Sea and
has a strong correlation with sea level anomaly. They also suggested that such a strong correlation is possible only when the





physical properties played a dominant role in defining the Oxygen minimum zone. The biological activity, however, can
determine the strength of the OMZ. Sarma et al. (2013) showed that freshwater discharge during the southwest monsoon
and associated organic debris input can help intensify the OMZ in the bay.
The high surface productivity may not necessarily translate into high export production, and this probably explains the
absence of seasonality in the Bay of Bengal OMZ. Anand et al. (2018) found that export production at 100 m depth in the
BoB was much less compared to that in the Arabian Sea, which indicated high utilization of organic matter by the
heterotrophs (Anand et al., 2017). This has implications for dissolved oxygen concentration in the BoB. The BoB, though
harbors low oxygen water in its core, its concentration is slightly above compared to its counterpart, the Arabian Sea, and
therefore was not denitrifying yet.  Bristow et al. (2017), based on dissolved oxygen measurements using STOX sensors,
suggested that dissolved oxygen concentration in the bay is much lower than earlier believed and suggested that any further
increase in primary productivity in the basin in the climate change scenario may lead to the formation of dead zones and
severely impact fisheries. Sridevi and Sarma (2020) argue that the observations of Bristow et al. (2017) may be biased due
to the selection of sites for sampling as most of the samples were collected from the outer boundary of the anticyclonic
eddies. Sarma and Bhaskar (2018), however, showed using Bio-Argo data that eddies recharge the subsurface water with
dissolved oxygen and is probably one of the reasons why the bay is still not denitrifying. Johnson et al.  (2019) further
argued, based on the Argo float-derived long-term oxygen and NO3-data, that highly variable dissolved oxygen
concentration, owing to injection of oxygen by eddies, helps BoB remain above the threshold limit to support denitrification.
Any climate-driven change in eddies distribution and the number of occurrences may alert this balance and drive bay towards
a denitrifying basin. At the same time, increasing occurrences of cyclonic activity may counterbalance an apparent decrease
in the number of eddies in the future under the climate change scenario.
*Primary and New production:* The established belief is that BoB is less productive (344 ± 164 MgC/M2/d; Gauns et al.,
2005) compared to the Arabian Sea (1,032 ± 260 MgC/m2/d; Barber et al., 2001). A recent study of Anand et al. (2018)
however, reported that productivity in the bay (936 ± 350 MgC/m2/d) is comparable to that in the AS and suggested that
the earlier reports underestimated the productivity partly due to less spatial coverage. Saxena et al. (2020) also recently
observed ~5-fold increase in productivity compared to the earlier reports (Madhupratap et al.,2003) and suggested that the
increase in productivity over the past two decades is in-line with the observed increase in the global marine primary
production. The lower concentration of nutrients, particularly NO3-, does not explain the observed high productivity. Sarma
et al. (2019) showed that 40-75% of the column productivity was observed in the upper 25m of the water column where the
inorganic component of nutrient was limiting (0.2uM), but dissolved fraction was available at much higher concentration
(>6uM). Therefore, uptake of DON by plankton may be responsible for the observed high productivity.
The new production, measured using [15]N tracer, in the BoB was higher during the pre-monsoon (5±4 mmol N/m2/d)
compared to the post-monsoon (2.6 mmol N/m2/d) (Kumar et al., 2004; Gandhi et al., 2010). The f-ratio, which provides
information about the fraction of primary production that can be exported to the deep, has been estimated to be high in the
BoB (Gandhi et al., 2010): it was higher during the pre-monsoon (~0.7) compared to post-monsoon (0.5). The high *f*-ratio



was in good agreement with the earlier estimates of high export fluxes in the bay (Ittekot et al., 1991). Sarma et al. (2018b), however, Anand et al. (2018) argued that the estimated high f-ratio needs to be revisited since most of the organic matter produced in the upper sunlit layer is remineralised and the export below 100m depth is quite low (<2% of the production). They reported f-ratio, estimated using the isotopic signature of nitrogen in the particulate organic matter, to be less than 0.31 even at the core of the cyclonic eddies; f-ratio was further lower in non-eddy and anticyclonic regions.

*$N_2$ fixation:* Bio-available NO3-being below the detection limit in the upper water column of BoB and an apparent nitrogen deficit of $4.7\pm 2.4$ Tg N/yr indicates the importance of dinitrogen fixation in the basin (Loscher et al., 2020). Though the presence of Trichodesmium in the bay has been reported by several authors, direct measurement of $N_2$ fixation rates was available from this basin till recently. Indirect estimates of $N_2$ fixation in the bay, using other proxies such as atmospheric soluble iron deposition, phosphorus limitation, and mass balance, range between 0.4 –1 TgN/Yr (Bikkina and Sarin 2013). Low $\delta^{15}N$ in the sinking matter also indicates a considerable amount of dinitrogen fixation in the bay. A recent report based on a field survey conducted in 2014 during the winter monsoon did not find detectable dinitrogen fixation in the bay despite having $N_2$ fixer community in the water column (Loscher et al., 2020). Since the sampling was carried out in the Oxygen minimum zone layer below the photic zone (>60m) and mostly diazotrophs are found in near-surface waters, the fixation rates were below the detection limit. The isotopic analysis of the settling flux also did not indicate significant $N_2$ fixation. Saxena et al. (2020) recently reported first N2 fixation rates from the bay using $^{15}N_2$ tracer gas experiment. Despite getting the signature of N2 fixation in the particulate organic matter ($\delta^{15}N$ ~0-6‰), the $N_2$ fixation rates were very low (4 to 75 μmol N/m2/d): the maximum fixation rates were measured at the surface, which decreases with the depth. Saxena et al. (2020) further argued that fixation rates are too low to support high nitrogen demand for the observed primary productivity rates and contributes to only 1% of the observed productivity of the bay. A considerable NO3- deficit, undetectable NO3-concentration in the upper water column, presence of diazotrophs capable of doing $N_2$ fixation, and yet only 1% contribution towards the primary productivity of the bay indeed calls for a more focussed approach to measure fixation rates in this important basin.

**Modeling Biogeochemical processes in the Bay**: Vinayachandran et al. (2005) made the first attempt to use a four-component ecosystem model coupled with a general circulation model to simulate the evolution of phytoplankton bloom in the bay during the northeast monsoon. The biogeochemical simulation successfully captured the bloom evolution supported not only by the entrainment of nutrients but also through upward transport of significant amounts of chlorophyll from the sub-surface layer. It highlights the contribution of deep chlorophyll maxima in the observed surface bloom. Thushara and Vinayachandran (2016) later studied the evolution of phytoplankton bloom in the northwestern bay during the summer monsoon. Their chlorophyll simulations could successfully capture the seasonal distribution of biomass, including the coastal blooms at major river discharge mouths, and were in good agreement with the satellite-derived chlorophyll data. The bloom intensity, however, is more realistic in the east of Srilanka and also in parts of the Andaman Sea. At other places, models tend to underestimate the chlorophyll values in comparison with the satellite chlorophyll. Thushara and Vinayachandran (2016) argued that the negative bias might be due to overestimation in the satellite-derived chlorophyll.

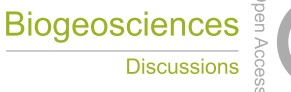

Their biogeochemical simulations showed that as the river plumes were pushed away due to coastal upwelling, they did not
change the biological production in model simulations. Surface winds appear to have significant control over governing
bloom in the southwestern bay during the summer monsoon. Chakraborty et al. (2018), a high-resolution biogeochemical
model coupled with Regional Ocean Modelling System (ROMS), investigated the CO2 dynamics of the Sri Lanka dome
region, which experiences intense upwelling during the southwest monsoon and showed that biological processes in the
upwelling region contribute towards draw-down of the pCO2. Their simulations indicated that biological processes dominate
over the physical upwelling in terms of the CO2 outgassing and leads to a net decrease (~11uatm) of pCO2. Shallower
nitracline in the region pumps more nutrients into the upper layer and fuels biological production that compensates for the
CO2 outgassing. In fact, the region becomes a sink for CO2 despite having significant upwelling. These new results vouch
for more concerted modeling efforts in the bay, a basin that has received insufficient modelling attention.
To summarize, recent studies indicate that the bay still remains largely an unexplored region for the following reasons: 1.
sources of nutrients such as dissolved inorganic nitrogen and its contribution to the primary production is heavily
underestimated, 2. The historical data and understanding of the quantum of primary productivity also have been questioned
in the new estimates, 3. A large influx of debris and dissolved organic matter and their decomposition have huge
contributions to the outgassing. They have ramifications on the carbon budget for the bay and also on the ocean acidification
at least near the zones of such discharges and 4. Finally, all of these have an impact on the dissolved oxygen content at the
mid-depth  layer in the bay. New state of the art instruments, measurement techniques, and sustained long term observations
can help understand and predict if the bay will be the "next hotspot" waiting to explode under climate change.
**2.10 Sumatra and Java**
The upwelling off the southern coasts of Sumatra and Java Islands is a remarkable eastern boundary upwelling system
(EBUS) in the Indian Ocean. Comparing with the other EBUS in the Pacific and Atlantic Oceans (Chavez et al., 2009),
however, the upwelling system in the Sumatra-Java coast had been overlooked until recently despite its important roles in
climate and ecosystem dynamics. This is because the average magnitude of the upwelling signals is smaller compared to the
other major EBUS in the world, due partly to a strong seasonality associated with monsoonal wind forcing and partly to the
existence of the Indonesian throughflow to the east and south of the upwelling region (Qu et al., 1994; Du et al., 2005). In
addition, the insufficient availability of in-situ data and complex geometry near the Indonesian Seas make detailed
investigations difficult.
The Sumatra-Java upwelling region is embedded in rather complex upper-layer circulations in the southeastern tropical
Indian Ocean between Indonesia and Australia (**Figure 20**). Seasonally changing South Java Coastal Current is associated
with the monsoonal wind reversal and is directly linked with the upwelling system (Quadfasel and Cresswell, 1992; Sprintall
et al., 1999). The westward flowing South Equatorial Current, a part of which is fed by the Indonesian throughflow from



the Indonesian Archipelago, is located to the south of the upwelling region (e.g., Qu and Meyers, 2005a). Since the Sumatra-
Java upwelling region sits close to the equator (from the equatorial region down to around 9°S), the equatorial and coastal
waveguide affects variability in upwelling signatures significantly.

A major feature of the upwelling in this region is the seasonal variation associated with the monsoonal wind along the coasts;
the upwelling favorable southeasterly winds prevail during boreal summer while the northwesterly winds appear during
boreal winter, which generates downwelling condition. Wyrtki (1962) is the first to demonstrate that the Ekman upwelling
along the coast of Sumatra and Java occurs during the boreal summer associated with the local southeasterly monsoon over
the region. Upwelling signatures in this region are well observed in in-situ measurements and satellite remote sensing
observations as in the other upwelling regions; e.g., cooler SST and subsurface temperature, shallower thermocline depth
and lower sea surface height, and higher chlorophyll-a and nutrients concentrations compared to the surrounding area (e.g.,
Wyrtki, 1962; Susanto et al., 2001; Susanto and Marra, 2005; Iskandar et al., 2017) **(Figure 20)**. The upwelling signatures
propagate to the west in association with the westward movement of the along-shore winds (Susanto et al., 2001). However,
locations of maximum amplitude of the upwelling signatures may differ in space among the variables due to dynamical
upper-ocean responses to the wind forcing. One such example can be seen in a phase relation between the winds and SST
along the Java coast; strong winds appear in the western area of the Java coast while the SST signal comes further east
(Naulita et al., 2020). In addition to this local wind forcing, the Sumatra-Java coastal area is affected by remotely forced
Kelvin waves propagating from regions further northwest along the Sumatra coast and from the equatorial Indian Ocean.
Several studies have shown that both the local wind forcing and this remote wave influence play key roles in determining
the magnitude and area of the Sumatra-Java upwelling (e.g., Cheng et al., 2016; Horii et al., 2016; Delman et al., 2018).

The local and remote influences vary year-to-year, generating interannual variability of upwelling strength and spatial
coverage. The most significant interannual variability in the tropical Indian Ocean is the Indian Ocean Dipole Mode (IOD),
whose center of action in the eastern pole appears over the Sumatra-Java upwelling region. During the positive IOD, the
southeastern Indian Ocean, particularly along the Sumatra-Java coasts, are occupied by negative SST anomaly, which tends
to be phase-locked to the seasonal upwelling during the boreal summer to fall. While the cool SST in seasonal variation is
pronounced along the Java coast (see **Figure 20**), the interannual SST anomaly appear in both Sumatra and Java coastal
regions with relatively stronger signal off the coast of Sumatra. Besides, the upwelling variability in the interannual time-
scale is also related to ENSO phenomena in the Pacific Ocean (Susanto et al., 2001; Susanto and Marra, 2005), partly due
to co-occurrence of ENSO and IOD in some years and also to atmospheric teleconnections from the Pacific for modifying
the strength of along coast component of the wind stress over Sumatra and Java. Oceanic teleconnections from the Pacific
through the Indonesian archipelago via the Indonesian throughflow may also affect the upwelling variability.

There have been attempts to investigate the ocean processes responsible for the seasonal and interannual variations in the
mixed-layer or upper-layer temperature using heat/temperature budget analyses. Both the seasonal and interannual variations



are dominated by vertical processes associated with the Ekman upwelling, with significant contributions from horizontal
advection, including the one from the Indonesian throughflow (e.g., Qu et al., 1994; Du et al., 2005, 2008). The barrier layer
is also affecting the seasonal SST variability, especially in the region off the Sumatra coast (Du et al., 2005; Qu et al.,
2005b). For interannual time-scales, the SST variability is driven by both the local and remote wind forcing, both of which
are strongly related to the IOD and to a lesser extent to ENSO, while the thermocline depth variations are mostly due to the
remote wave influences from the equatorial eastern Indian Ocean (Chen et al., 2016). There are several studies, including
those under the IIOE-2 program, focusing on the roles of variability in the upwelling region on the evolution of IOD events.
Initiation of positive IOD events is related to anomalous cooling off the coast of Sumatra-Java, which may be generated by
local winds, particularly along the coast of Sumatra Island (Delman et al., 2016, 2018; Kämpf and Kavi, 2019), or remotely
forced Kelvin wave signal originated in the equatorial Indian Ocean (Horii et al., 2008). During the mature stage of the
positive IOD, vertical upwelling processes, as well as horizontal advection, contribute to keeping anomalous cooling of the
SST off the coasts of Sumatra and Java (Du et al., 2008; Chen et al., 2016). While the seasonal March to the northwesterly
monsoon condition terminates this maintaining process forced by local and remote winds, oceanic eddy heat flux associated
with mesoscale eddies generated by enhanced instability during the height of the positive IOD is also shown to facilitate the
disappearance of the anomalous conditions in the strong events (Ogata and Masumoto, 2010, 2011).

The upwelling off Sumatra and Java is also affected by intraseasonal variability in the ocean and atmosphere. This
intraseasonal variability is, as in the case of seasonal and interannual variations, due both to the local winds and to remotely
forced oceanic Kelvin waves (Iskandar et al., 2006; Horii et al., 2016). Even during the height of the positive IOD in 2008,
strong intraseasonal upwelling signals are observed in temperature and salinity profiles obtained by Argo floats (Horii et al.,
2018). In addition, a long-term trend in the upwelling strength is studied recently. For example, Varela et al. (2016) suggest
the weakening of the upwelling although the SST shows a cooling trend due to cooler subsurface temperature. Sources of
this upwelled water mass are not clearly determined yet. A study using a numerical model proposes the Indonesian
throughflow as a possible source for the upwelling water off the Java coast (Valsala and Maksyutov, 2010), while another
study suggests that water mass from the northwest via South Java Current could be upwelled in this region (Varela et al.,

1390    2016).


As in the other EBUS, high biological productivity is expected in the Sumatra-Java upwelling region during boreal summer
and fall (Wei et al., 2012), with high nutrient and chlorophyll-a concentrations along the coasts (e.g. Wyrtki 1962; Tranter
and Newell, 1962; Susanto et al., 2001; Asanuma et al., 2003; Iskandar et al., 2009). Spatial distributions and temporal
variations of various biogeochemical parameters have been detected from in-situ observations, coral records, and sediment
cores for the present situations and paleo-oceanographic conditions (e.g., Grumet et al., 2004; Murgese and De Deckker,
2005; Andruleit 2007; Andruleit et al., 2008; Baumgart et al., 2010; Ehlert et al., 2011). Recent in-situ observations in the
Sumatra-Java upwelling region conducted during the IIOE-2 period indicate different phytoplankton compositions and



assemblages between upwelling and non-upwelling regions (Gao et al., 2018) and physical and biological processes that
determine aragonite saturation state (Xue et al., 2016). Efforts to develop and improve biogeochemical models of the
upwelling systems are also in progress (e.g., Sreeush et al., 2018). These researches on biogeochemistry are important to
understand key processes operating in the Sumatra-Java upwelling system. However, these results are rather fragmented at
this stage, and integrated studies on biophysical interactions, ecosystem dynamics, and the marine food web in the Sumatra-
Java upwelling region would be strongly required.

During the IIOE-2 period (2015 - 2020), Argo float measurements and satellite remote sensing data have been accumulated
significantly, and there were several in-situ observations of physical and biogeochemical aspects of the upwelling system.
These data sets provide us key evidences for a better understanding of physical processes responsible for the upwelling
variability in various time-scales and of distributions of biogeochemical variables. However, in-situ measurements are still
quite limited to obtain a synthetic view of the upwelling system off Sumatra-Java coasts, particularly on biogeochemical
parameters. Mixed-layer dynamics and mixing processes in this unique region for relating subsurface oceanic variability to
SST need to be investigate in more detail. Further observations and accumulation of additional evidences are necessary to
obtain a comprehensive view of the upwelling system off the Sumatra-Java coasts.

**2.11 West coast of Australia**
**2.11.1 Overview**
Unlike other eastern boundary systems, such as the highly productive Humboldt and Benguela upwelling systems, the west
coast of Australia features a downwelling current that carries tropical water southward along the shelf-break (Pearce, 1991).
In the late nineteenth century, the presence of tropical corals at the Abrolhos Islands (28°- 29°S 114° E) was observed by
Saville-Kent (1897), and from sea temperature measurements, he postulated that there was an offshore, warm, southward-
flowing current. A drift-card study conducted near Rottnest Island (32°S, 115°E) confirmed that during the austral winter,
there was a southward flowing current (Rochford, 1969), and Gentilli (1972) explored the seasonal southward progression
of "rafts" of warm water off the west coast of Australia.
The Leeuwin Current (LC) was named and described by Cresswell and Golding (1980) from the trajectories of satellite-
tracked buoys and measurements of surface temperature and salinity from the shelf and slope stations. Other early studies
of the LC, which included using current meters, shipboard measurements, satellite imagery, steric height gradients, wind
stress calculations, and modeling, identified the seasonal nature of the current, origins along the North West Shelf, an
eastward extension to the Great Australian Bight, the frequent presence of meanders and eddies and low nutrient status
(Godfrey and Ridgway, 1985; Holloway and Nye, 1985; Pearce, 1991; Smith et al., 1991; Thompson, 1984; Thompson and
Veronis, 1983; Weaver and Middleton, 1989). Essentially, the alongshore steric height gradient is set up by the Indonesian
throughflow (which delivers warm, less saline waters from the Pacific into the Indian Ocean) and surface heat loss at higher
latitudes (Smith et al., 1991). Later, using remote sensing and modeling, research attention centered on understanding the



influence of the LC on the recruitment of puerulus larvae of the economically-important rock lobster *Panulirus cygnus* (e.g.,
Griffin et al., 2001; Phillips and Pearce, 1997).
More recent shipboard studies along the entire west coast of Australia (Weller et al., 2011; Woo and Pattiaratchi, 2008)
provided more detailed information on the trajectory and features of the LC, including the chemistry, primary production,
zooplankton and larval fishes (Buchanan and Beckley, 2016; Holliday et al., 2012; Lourey et al., 2012; Sutton and Beckley,
2016; Thompson et al., 2011). Further, remote sensing and modeling have confirmed the seasonal nature of the LC and the
influence of the El Niño Southern Oscillation with stronger LC flows occurring during La Niña years (Domingues et al.,
2007; Feng et al., 2003). The ecological significance of the LC eddy field was investigated with two dedicated voyages
(Paterson et al., 2008; Waite et al., 2007b). The most recent ecological work explored the significance of the LC and its
eddy field on the ecology of the planktonic phyllosoma of *P. cygnus* in the wake of a drastic decline in puerulus settlement
(Saunders et al., 2012; Säwström et al., 2014; Waite et al., 2019).
Many of the early studies on the LC noted the occurrence of inshore northward-flowing currents during the summer months
(e.g., Thompson and Veronis, 1983; Thompson, 1984) with Holloway and Nye (1985) specifically mentioning upwelling
along the northwest coast. Subsequent studies highlighted regional upwelling nodes (see below) and, using an upwelling
index developed from 15 years of satellite data, (Rossi et al., 2013b) produced a synopsis covering the development of
sporadic upwelling events (generally lasting 3-10 days) along the entire western coast of Australia (**Figure 21 and 22**).
Although such upwelling generally occurs from September to April (austral summer) sporadic events can occur at any time
north of 30°S (**Figure 23**). Upwelling favorable winds, local topography, and the characteristics of the LC such as onshore
geostrophic flow, stratification, mesoscale eddies, and meanders influence the intensity of intermittent upwelling. For this
review of upwelling along the western coast of Australia, we have separated the coast into three nodes of upwelling, namely,
the South West (35°-30° S), Central (30°-25° S), Gascoyne (25°- 22° S) and will also cover upwelling in the eddy field.

**2.11.2 Upwelling nodes**
**South West**
A northward-flowing inshore current along parts of the southwest coast was indicated by several early LC studies (Cresswell
et al., 1989; Cresswell and Peterson, 1993; Cresswell and Golding, 1980; Pearce, 1991; Thompson, 1987). For example,
Cresswell and Peterson (1993) noted in the austral summer of 1986-87 that a cold upwelling plume (~17.5°C) extended
westward along the shelf from the southern coast of Western Australia around Cape Leeuwin and northward to Cape
Naturaliste. They speculated that the absence of the LC south of 34°S might have allowed upwelling-favorable easterly
winds on the south coast to drive this upwelling. From a detailed analysis of satellite imagery (1987-1993) and environmental
data, Pearce and Pattiaratchi (1999) described the narrow, northward flowing counter-current between Cape Leeuwin and
Cape Naturaliste during the austral summer months and named it the Capes Current. They indicated that strong northward
wind stresses between November and March slowed the LC and drove the Capes Current and that localized upwelling also
contributed to it. This upwelling was examined by Gersbach et al. (1999) using XBT, CTD, nutrient, and ADCP data from





summer sections off Cape Mentelle (located between Cape Leeuwin and Cape Naturaliste) and several sections between
Albany and Perth, as well as wind data and satellite imagery. They concluded that northward wind stress in summer could
overcome the alongshore steric height gradient on the continental shelf, inducing the thermocline to lift and sporadic
upwelling to occur 5-9 times per summer. Interestingly, the T/S characteristics of the water upwelled at Cape Mentelle were
slightly different from that of the current setup from the south (Gersbach et al., 1999).
Rossi et al. (2013b) indicated that, overall, the transient upwelling events in this southwest region last 3-10 days, and shelf
regions between 34°S and 31.5°S exhibit up to 12 upwelling days per month during the austral spring/summer (Figure 22).
Historical current measurements near Perth suggest that the Capes Current continues northwards past Rottnest Island, and
there may also be links with shelf counter-currents well past the Abrolhos Islands at 29°S (Cresswell et al., 1989).

**2.11.3 Central coast**

Along the central Western Australian shelf inshore of the Leeuwin Current, there is a general northward flow during the
austral summer months based on current measurements across the shelf near the Houtman Abrolhos Islands (Cresswell et
al., 1989; Rochford, 1969). Rossi et al. (2013b) indicated a high upwelling index along the central coast with locations
around 28°S and 26° S producing elevated mean numbers of upwelling days per year. Interestingly, both show peaks in
upwelling from March to May, with upwelling at 28° S continuing through the austral winter.

**2.11.4 Gascoyne coast**

The Gascoyne coast is characterised by a northward flowing inshore current known as the Ningaloo Current. Various studies
have revealed that the Ningaloo Current consists of water sourced from upwelling of shallow water (~100 m) from the base
of the LC (Hanson et al., 2005; Taylor and Pearce, 1999; Woo and Pattiaratchi, 2008; Woo et al., 2006a, 2006b). Previously,
it was understood to be strongly seasonal in the austral summer, but recent investigations have shown autumn upwelling
events as well (Lowe et al., 2012; Xu et al., 2013; Rossi et al., 2013a). The source water in autumn may be from a greater
depth (150m) under the increased mixed layer depth (Rossi et al., 2013a). The Ningaloo upwelling region around 22.5°S
has the highest number of upwelling days per year (140), and the events are often longer in duration than elsewhere on the
west coast (Rossi et al., 2013b).

**2.11.5 Biogeochemical & ecological impacts**
**Nutrients, primary productivity, pro- and micro-eukaryotes**
In the next section, we explore how the regional dynamics along the west coast of Australia control the spatio-temporal
variability of biogeochemical cycles, such as primary productivity and nutrient cycles in general. Water column productivity
along the west coast of Australia (generally <200 mg C m$^{-2}$ d$^{-1}$; Hanson et al., 2005) peaks at the deep chlorophyll maximum





(DCM), which is closely aligned with the base of the nutricline. Productivity at the DCM in this system is strongly influenced
by the mixed layer depth (MLD), with deeper DCMs having lower productivity and shallower DCMs resulting in higher
productivity rates (Hanson et al., 2007a; Johannes et al., 1994).
Furnas (2007) argued that intermittent bursts of high productivity could occur in specific locations or under certain
circumstances along this coast. The strength of the LC is controlled by the weakening of southerly winds in the austral
autumn and winter. Modeling results from Feng et al. (2003) suggest that an increase in the southward transport of the LC
has been linked to an erosion of the thermocline, which then brings $NO_3^-$ into the euphotic zone, thereby enhancing primary
productivity in early autumn (Feng et al., 2009; Rousseaux et al., 2012). Satellite observations across the shelf and LC
confirmed these results with the highest chlorophyll levels in autumn and winter (Lourey et al., 2006; Lourey et al., 2012).
Similarly, in summer, the wind-driven Capes Current locally enhances productivity near the shelf (Pearce and Pattiaratchi,
1999), yet, generally, the oligotrophic nature of this system limits $NO_3^-$ driven new production throughout the year (Hanson
et al., 2005; Twomey et al., 2007). The recycling of organic matter, however, via microbial regeneration has been shown to
primarily control the rates of primary production in this system (Hanson et al., 2007b; Pearce et al., 2006; Twomey et al.,
2007) rather than the strong upwelling as seen in other eastern boundary systems, such as the Humboldt and Benguela
systems (Hood et al., 2017).
The inputs of new N derived from $N_2$ fixation is also an important pathway supporting primary productivity in this region.
Along the Western Australian shelf from 32°S northwards to 12°S, and west to 110°E, the contribution of new N from $N_2$
fixation towards the total dissolved inorganic nitrogen (DIN) assimilation pool can be ~ 20% and up to 50% during the
winter months (with $N_2$ fixation rates ranging from 0.01 up to 12 $nmol^{-1}$ $L^{-1}$ $h^{-1}$), making $N_2$ fixation equal to $NO_3^-$ in terms
of N assimilation into this ecosystem (Raes et al., 2015; Raes et al., 2014). Waite et al. (2013) and Raes et al. (2015) also
noted that the small diffusive deep-water $NO_3^-$ fluxes are not able to support the measured $NO_3^-$ assimilation rates. Their
data suggest that nitrification above the nutricline (referred to as "shallow nitrification"; see also Thompson et al., 2011)
could be an important pathway of the N-cycle along the southwest coast of Australia. Waite et al. (2016) suggested that the
persistent layers of low oxygen, high dissolved nitrogen (LDOHN; $O_2$ ~150 $\mu mol$ $L^{-1}$ and $NO_3^-$ ~2-10 $\mu mol$ $L^{-1}$) just below
the euphotic zone (~150-250m; Thompson et al., 2011; Weller et al., 2011) are hotspots for the mineralization of organic
material from local sources (< 500 km away). In addition, Waite et al. (2016) noted that the depletion of oxygen in these
isolated layers along with the release of NO3-, could happen on a time-scale of ~2 weeks. Warren (1981), on the other hand,
originally suggested that the isolated nature of the lower oxygen features is created by density gradients, which prevent the
mixing of deep $O_2$ rich water. According to Thompson et al. (2011) and Weller et al. (2011), the source of the oxygen
minimum layer is associated with multiple water masses further upstream, possibly at lower latitudes north of Australia.
Overall, a number of studies point to the conclusion that an active microbial loop (Azam et al., 1983) controls the biogenic
C and N fluxes through heterotrophic recycling via ammonification, nitrification, and $N_2$ fixation in this vast region (Hanson
et al., 2007b; Raes et al., 2015; Waite et al., 2016).



The west coast of Australia has been suggested to have a subtropical phytoplankton cycle, with a winter bloom, similar to
the open ocean waters of the subtropical South Indian Ocean (**Figure 24).** Picoplankton (unicellular cyanobacteria and
prochlorophytes) have been shown to contribute >40% of the pigment biomass (Hanson et al., 2007b). In terms of bio-
volume, the Dinophyceae, including large gymnoids and other Dinophyceae (e.g., *Gyrodinium* spp., *Prorocentrum* spp.),
are the most abundant microplankton and can account for up to 50% of the microplankton component in this region (Raes
et al., 2014). Sightings of $N_2$-fixing microorganisms (such as *Trichodesmium*) in the oligotrophic waters off the west coast
of Australia date back to voyages of Captain Cook and Charles Darwin (Cook et al., 1999; Darwin, 1889). *Trichodesmium*
occurrences have been measured at the Australian National Reference stations from the tropics (Darwin) to the temperate
waters off Rottnest Island.

Amplicon sequencing of the nitrogenase (*nifH*) gene, however, has shown a low diazotrophic evenness across a transect
along the shelf from Perth (32$^o$S) to Darwin (10°S). One operational taxonomic unit (OTU) made up 65–95% of the *nifH*
enzyme diversity along the transect, and was identified as a Gamma 4 proteobacteria (Raes et al., 2018). This dominant *nifH*
OTU was nearly identical (one nucleotide difference) to the gamma 4 proteobacteria (HM201363.1) found by Halm et al.
(2012) in the oligotrophic South Pacific Gyre. The ubiquitous finding of these gamma proteobacterial *nifH* genes is
consistent with the results from Schmidt et al. (1991) and Langlois et al. (2015) in open, oligotrophic oceanic waters.

**2.11.6 Zooplankton**

Ecological studies about zooplankton community structure along the west coast of Australia are few, and most studies have
examined specific taxa (e.g., larval fishes, chaetognaths, or krill) particularly in relation to the effect of the LC on dispersal
(Beckley et al., 2009; Buchanan & Beckley, 2016; Holliday et al., 2012; Sutton & Beckley, 2016). Although inshore stations
(50 m depth) were sampled during the voyage when most of these studies were made (extending from 22°S - 34°S), there
was no evidence of coastal upwelling, likely because it was conducted in the austral autumn (May 2007). Recently, meso-
zooplankton abundance, composition, and diversity data from the three years (2010-2012) that the IMOS Australian
National Reference Stations (Ningaloo, Rottnest Island, and Esperance) were concurrently sampled were analyzed
(McCosker et al., 2020).  Besides the obvious influence of the LC in winter, there were clear dissimilarities between the
copepod assemblages, particularly during the summer months when coastal upwelling-associated currents such as the Capes
and Ningaloo Currents influenced the biota.
Specific effects of coastal upwelling on zooplankton have not been explored in the South West, but concurrent with the
phytoplankton study of Koslow et al. (2008) across the Two Rocks transect north of Perth, mesozooplankton assemblages
were examined (Strzelecki and Koslow, 2006). During the summer, the inshore shelf stations were found to have
significantly higher zooplankton abundance than the offshore sampling stations, but this was reversed in the winter months
when the LC was flowing strongly. Copepod production ranged from 0.4-10 mg C m$^{-2}$ d$^{-1}$ (Strzelecki and Koslow, 2006),



which is low compared to upwelling regions elsewhere in the world but comparable to copepod production in the North
West Cape region (McKinnon and Duggan, 2003). Along the same cross-shelf transect, Muhling and Beckley (2007) and
Muhling et al. (2008b) found clear seasonal differences in the diversity and abundance of inshore larval fish assemblages
when the cool Capes Current was flowing northwards during the austral summer compared to the austral winter months
when the LC strongly influenced larval fish assemblages on the continental shelf.
Little is known about the effect of coastal upwelling on zooplankton along the central part of the Western Australian coast,
and the only extensive zooplankton survey in the region targeted the phyllosoma larvae of the rock lobster, *Panulirus cygnus.*
Nevertheless, the study highlighted the presence of the Abrolhos front separating the tropical waters of the LC from the
dominant oligotrophic subtropical surface water (STSW), and the LC waters had much higher chlorophyll *a* and zooplankton
concentrations than the STSW (Säwström et al., 2014).
In the north, the coastal copepod communities at Ningaloo are diverse (> 120 species; McKinnon and Duggan 2001). They
are characterized by small "upwelling- ready" species, which can react quickly to pulses of sporadic upwelling and
phytoplankton blooms, but, unlike the high primary production rates, copepod production rates are generally low (~ 13 mg
C m$^{-2}$ d$^{-1}$; Hanson and McKinnon, 2009). Interestingly, *Calanoides carinatus*, a copepod that is characteristic of upwelling
regimes elsewhere, was absent, and they proposed that upwelling was too infrequent and episodic to sustain zooplankton
specific to upwelling regimes. Of the macro-zooplankton, krill, especially, *Pseudeuphausia latifrons* has been investigated
in coastal waters at Ningaloo (Wilson et al., 2003), and seasonal occurrence of whale sharks has been linked to aggregations
of this species during the austral autumn months (Hanson and McKinnon 2009).

**2.11.7 Fisheries**

Investigations into the spawning of sardines (*Sardinops sagax*) off southwestern Australia have highlighted advective
transport (Fletcher et al., 1994; Gaughan et al., 2001b) and variation in the growth rate of larvae from areas with different
levels of productivity (Gaughan et al., 2001a). Muhling et al. (2008a) showed that, although adult sardines had a winter
spawning peak coinciding with the seasonal peak in chlorophyll *a* (Koslow et al., 2008), it also matched the seasonal peak
in the southward flow of the LC, resulting in low retention of the early life history stages. Thus, egg and larval concentrations
were lower than expected in winter but higher in summer when retention conditions were more favorable. They postulated
that, as larval growth rates were actually high, the insignificant catches of adults in the fishery compared to other eastern
boundary upwelling systems was due to a combination of suppression of large-scale upwelling and the modest seasonal
maximum in primary productivity occurring during the time least favourable for pelagic larval retention.
There has been commentaries on the role of the Capes Current in assisting migrations of south coast fish species such as
*Arripis truttaceus* and *Arripis georgianus* in their migrations to autumn spawning areas in southwestern Australia and
subsequent return transport of early life stages by the LC during winter (Pearce and Pattiaratchi, 1999). Both Caputi et al.
(1996) and Lenanton et al. (2009) have reviewed the importance of the LC with respect to Western Australian fisheries and



have noted the likely role of the Capes Current for several species, including the economically important rock lobster.
Through modeling, Feng et al. (2010) examined dispersal and retention areas along the west coast. Although the LC was
dominant in winter, northward flow in summer was linked with recruitment success of scallops (*Amusium balloti)*, abalone
(*Haliotis roei),* and tropical sardines (*Sardinella lemuru).*

**2.11.8 Eddies**
The vast eddy-field associated with the LC is well-known and has been investigated by numerous oceanographers and
shown to influence regional biogeochemistry and pelagic ecology (e.g., Andrews, 1977; Feng et al., 2007; Waite et al.,
2007b; Moore et al., 2007; Paterson et al., 2008; Holliday et al., 2011; Dufois et al., 2014). Though the warm, anticyclonic
eddies have been explored in greater detail than the cyclonic eddies, it is the latter which are cold-core upwelling systems
and deserve mention here as they have been shown to drive a significant fraction of cross-shelf transport and enhance local
and regional productivity (Waite et al., 2016).

A study contrasting a dipole pair of eddies off Western Australia revealed many differences in the biota between the two
eddies (Muhling et al., 2007; Strzelecki et al., 2007) as a result of the differences in physical and chemical properties. Warm-
core eddies (WCEs; anticyclonic) have greater surface chlorophyll signatures compared to cold-core eddies (CCEs;
cyclonic) in the eastern Indian Ocean (Dufois et al., 2014; Waite et al., 2016). Yet, Waite et al. (2019) showed that CCEs
actually have greater depth-integrated primary productivity as their shallower mixed layers are closer to the nutricline and
across pycnocline mixing then increases the upward flux of dissolved inorganic nutrients. This results in greater flagellate
and dinoflagellate populations in CCEs, which provide a high-quality food source for zooplankton, and consequently
increases their lipid stores (Waite et al., 2019). Earlier work showed no significant differences between the fractionated
isotopic zooplankton analyses between CCEs and WCEs but highlighted that micro-heterotrophs are positioned on a trophic
level as third and fourth-order consumers (Waite et al., 2007a). The high position of micro-heterotrophs again confirms the
rapid recycling of particulate organic matter in this system in general, as outlined previously (Hanson et al., 2005; Raes et
al., 2014; Twomey et al., 2007). Further, upwelling eddies generated by the Leeuwin Undercurrent in the Perth Canyon have
been implicated in the abundance of krill in the area and consequent feeding of migrating blue whales (Rennie et al., 2007).
**2.12 Seychelles-Chagos Thermocline Ridge**
The Seychelles-Chagos Thermocline Ridge (SCTR, Xie et al, 2002; Hermes and Reason, 2008; Yokoi et al., 2008; Vialard
et al., 2009) is an upwelling region across the southern tropical Indian Ocean in the latitude band 5-15°S (**Figure 25**). It is
characterized by a thin mixed layer (~30m) and a relatively shallow thermocline. The ridge, and the upwelling associated
with it, is set up by wind stress curl patterns, and it has significant variability on seasonal and interannual time scales due to
both remote and local and forcing (Xie et al., 2002; Hermes and Reason, 2008; Yokoi et al., 2008; McPhaden and Nagura,



2014; Nyadjro et al., 2017). It is coincident with the southernmost latitudes of monsoon-driven circulation in the Indian Ocean, south of which a steadier trade wind regime prevails (**Figure 26**). During boreal winter, the Intertropical Convergence Zone (ITCZ) is located over the SCTR. The ITCZ and associated rainfall migrate northwards to the Indian subcontinent as the year progresses, where it is the source of precipitation during the summer monsoon.

Upwelling along the SCTR affects sea surface temperature (SST, **Figure 26**), biogeochemistry (**Figure 26**), and fisheries (**Figure 27**), and drives strong ocean-atmosphere coupling (Vinayachandran and Saji, 2008, Vialard et al., 2009; Resplandy et al., 2009; Robinson et al., 2010; Dilmahamod et al., 2016). As discussed previously, upwelling centers in the monsoon-dominated Indian Ocean are found in off-equatorial regions because the mean winds along the equator are westerly, unlike in the easterly trade wind-forced Pacific and Atlantic Oceans (Schott et al., 2009; Wang and McPhaden, 2017). The SCTR is the largest and most persistent upwelling region in the Indian Ocean.

The SCTR represents the ascending branch of the subtropical circulation cell in the southern hemisphere, where upwelling is balanced primarily by meridionally divergent flow in the surface layer (Lee, 2004; **Figure 28**). Horizontal flow in the upper ocean circulates cyclonically around the SCTR axis, with the westward-flowing South Equatorial Current (SEC) to the south and the eastward-flowing South Equatorial Countercurrent (SECC) to the north (**Figure 25**). The westward-flowing SEC in the SCTR region provides the conduit for interbasin exchanges that link the Pacific Ocean to the Atlantic Ocean through the Indonesian Seas and the Agulhas Current.

SST varies substantially in the SCTR on intraseasonal to interannual time scales because the shallow mixed layer is sensitive to changes in upwelling, vertical mixing, air-sea heat fluxes, and horizontal advection (Vialard et al., 2008; Foltz et al., 2010). On intraseasonal time scales, pronounced SST variations in the SCTR happen in response to forcing from the Madden-Julian Oscillation (MJO, Madden and Julian 1972), which is generated in this region. This variability feeds back to the atmosphere, which helps to organize the MJO convective cells. Large SST variations on interannual time scales are associated with the El Nino Southern Oscillation (ENSO) and the Indian Ocean Dipole (IOD; Webster et al., 1999; Saji et al., 1999). This year-to-year SST variability affects the frequency of Indian summer monsoon rainfall (Izumo et al., 2008), tropical storms in the southwestern Indian Ocean (Xie et al., 2002), and the climate of East Asia (Yamagata et al., 2004). The IOD also profoundly affects the tuna fishery in the Indian Ocean, which is well developed in the SCTR during normal years (**Figure 27**). However, during the positive IOD events, when upwelling is weakened in the SCTR and strengthened off the coast of Java and Sumatra, tuna migrate eastward, apparently in search of more favorable foraging grounds (**Figure 27**, Robinson et al., 2010).

Observational studies have documented concentrated tuna fishing activities at locations where surface phytoplankton blooms had been observed in satellite observations 2-3 weeks previously (Fonteneau et al. 2008), demonstrating a strong connection between the food webs that respond to SCTR blooms and the prey required by large tuna. In contrast,





biogeochemical modeling results indicate that neither phytoplankton biomass nor carbon export from the euphotic zone
changes significantly in response to seasonal and interannual variability of the SCTR thermocline depth (Resplandy et al.,
2009). These contrasting results represent a paradox in the current understanding of ecological dynamics in the SCTR
upwelling region, which can neither be so subtle as to produce no biogeochemical signals from the episodic nutricline
intrusions into surface waters nor so strong and efficient as to produce an almost instantaneous population response of the
large, long-lived prey that tuna consume. The extent to which iron may be a limiting primary production in the SCTR region
is also unknown, though independent modeling studies and remote sensing-based analyses both suggest it may be (Wiggert
et al., 2006; Behrenfeld et al., 2009). Finally, there is still considerable uncertainty in whether the Indian Ocean is a net
source or sink of carbon to the atmosphere because the variability in pCO2 fluxes across the air-sea interface is poorly
constrained by existing observations, particularly in active upwelling zones like the SCTR
**3. Summary**

The unique features of the oceanography of the Indian Ocean and the complexities associated with its circulation, boundary
currents, climate, and ecosystem response, driven and modulated by the monsoons, have been a matter of extensive
discussion in the past reviews of the Indian Ocean (Shetye and Gouveia, 1998, Schott and McCreary, 2001 Shankar et al.,
2002, Hood et al. 2015). The coastal upwelling, despite its importance for the ecosystem and economic impacts, however,
has not received sufficient attention (Hood et al., 2015) till the recent past. Several new programs were launched in the last
decade, which has shed considerable new light on the coastal upwelling system in the Indian Ocean. The WIOURI was
initiated to study nine upwelling systems in the western Indian Ocean (Roberts et al., 2015). Similarly, EIOURI was planned
to study a large spectrum of processes affecting the upwelling in the eastern half of the Indian Ocean (Yu et al., 2016). Along
the coasts of India, an array of ADCP mooring deployed since 2008 (Mukhopadhyay et al., 2017, Chaudhari et al., 2020)
complemented intense programs such as WIOURI and EIOURI. Such programs have contributed significantly to enhancing
our knowledge of the science of the upwelling in the Indian Ocean, its science, ecosystem impacts, and sensitivity to changes
in the environment. The prime goal of this paper is to review the upwelling in the Indian Ocean extending from the Agulhas
region to the western coast of Australia.

*The Upwelling:* While some of the upwelling systems, such as that along the Somali coast, were surveyed early (during
IIOE or before), others such as Mozambique were sampled much later. The surveys, particularly those in the recent period,
have revealed multiple processes that trigger and control upwelling, the combination varying for each of the systems. Salient
features of their progress are summarized here. The northeasterly monsoon winds are favorable for upwelling along the
western boundary, in the southern hemisphere, up to about 20˚S. Along the coast of Kenya, in addition to an Ekman type of
mechanism, shelf-break upwelling induced by topography is a driving force. Along the coast of Tanzania, the additional
forcing for upwelling is drawn from the shear instability of EACC. In the Mozambique channel, competing roles of local



winds and eddies drive upwelling in the channel. South of Madagascar, upwelling is caused by local winds, the interaction
of the currents with the continental margin and eddies.  Eddies associated Natal pulses cause subsurface upwelling in the
Agulhas region, and surface-reaching upwelling occurs in its inshore edge due to dynamical processes and wind forcing.

The distinct feature of the Somali upwelling system is the cold wedges. One wedge forms in May on the shoreward edge of
the Southern Gyre during May and the other along the northern flank of the Great Whirl, during the peak of the summer
monsoon.  The presence of multiple gyres and the intense current present a complicated upwelling system in this region. In
addition to alongshore winds, Rossby wave radiation from the east by Ekman pumping driven by anticyclonic wind stress
curl drive upwelling in this region. The downwelling of the thermocline due to the wind stress curl, however, can lead to a
weakening of the upwelling as the deepening reaches up to the coastal region during the fully developed phase of the SWM.
Consequently, upwelling is limited to frontal regions dominated by eddies. The coast of Oman, on the other hand, presents
a classical Ekman type of upwelling system. The intensity of the upwelling increases with the progress in the SWM.
However, the influence of Rossby wave radiation has been suggested to affect the timing of the peak phase of the SST
decrease associated with upwelling. Generation of eddies and filaments are well-known features associated with the currents
and upwelling along the coast of Oman.

Along the west coast of India, upwelling is more prominent along the southern part of the coast and begins about two months
before the onset of the summer monsoon. The alongshore winds are weak and are only partly responsible for the upwelling.
The major driving force is the coastally trapped wave propagation originating from the Bay of Bengal. The alongshore winds
are unidirectional, but the currents reverse, confirming the dominant role of remote forcing.  Winds along the southern tip
of India and along the southern coast of Sri Lanka drive Ekman type of upwelling during the summer monsoon. Upwelling
along the east coast of India is feeble and available evidences suggest the presence of upwelling during the summer monsoon.
The intricate combination of forcing by local winds, Kelvin waves originating from either EIO or the eastern boundary of
the BoB, Rossby wave propagation all affect the upwelling. At interannual time scales, ENSO and IOD dominate the
variability, whereas at intraseasonal time scales, mesoscale eddies appear to be important.

The upwelling along the Sumatra and Coasts is mainly driven by alongshore winds during the summer monsoon but affected
by Kelvin wave propagations and circulation in the Equatorial Indian Ocean, Indonesian throughflow and the subtropical
Indian Ocean. It is affected severely by the IOD events and modified significantly by intraseasonal events. The circulation
along the west coast of Australia is dominated by the LC but upwelling occurs at several nodes along the coast.  Transient
wind-driven upwelling that lasts for 3-20 days occurs along the southwest part of the coast. Along the central coast,
upwelling takes place during March-May. Along the Gascoyne coast, Ningaloo upwelling takes place during austral summer
and autumn.





*Ecosystem Impacts:* It is evident that in all regions, the upwelling stimulates an ecosystem response and the facilitation of
this response is achieved by different processes in different regions. In the Mozambique channel, peripheries of the cyclonic
eddies are found to be centers of biological activity in terms of increased productivity, aggregation of small organisms and
foraging bird populations. Along the southern coast of Madagascar, upwelling nodes enhance primary productivity, fish
catch and whale sightings. The interannual variability of the cyanobacteria bloom here is modulated by the detachment of
the South-East Madagascar current. The chlorophyll concentration is high along the coasts of Somalia and Oman, during
the summer monsoon, which has been known since a long time. Recent advances in this region have been slow and a
modeling study suggests that the influence of upwelling is restricted to limited areas and the strong currents spread the effect
tolarger spatial coverage. Off the coast of Oman, advection of nutrient-rich water can give rise to blooms in the offshore
region.

Recent research has revealed the high impact of upwelling on the biogeochemistry of the eastern Arabian Sea. Most
significantly, the upwelling affects the OMZ and its spatial and temporal limits, in spite of the upwelling itself being weaker
compared to that in the western Arabian Sea. This has an impact on the mesopelagic fish population, benthic ecosystems,
macro infaunal communities and biodiversity. The upwelling in the Bay of Bengal, on the other hand, is feeble and it is not
clear what the ecosystem responses are to upwelling; The productivity appears to be more under the control of eddies   and
the stratification imposed by rainfall and river runoff. The upwelling along the coasts of Sumatra and Java enhances
productivity and the phytoplankton composition here is distinctly different during upwelling compared to that during
downwelling.  Along the west coast of Australia, upwelling has a lesser role in controlling the rates of primary productivity
compared to that of remineralization. However, there are indications that summertime zooplankton biota is affected by
upwelling. The SCTR is a prominent open ocean upwelling region in the Southern Topical Indian Ocean that are caused
primarily by the persistent wind stress curl and this upwelling has a clear expression on the surface chlorophyll distribution.
This region also has a significant role in the air-sea interaction in this region.

*Future prospects:* Some of the upwelling zones have registered significant progress during the period of IIOE-2 (2015
onwards) while some others have rather been left behind. Agulhas current, Mozambique channel, Madagascar Coasts and
coasts of India, Sumatra-Java and Africa belong to the former category whereas Somali and Oman costs to the latter. In
addition, the northern coast of the Arabian Sea and the eastern boundary of the Bay of Bengal still remain poorly observed
and understood. The spatial and temporal variability of upwelling is not sufficiently documented for the most part of the
Indian Ocean coastline. This emphasizes the importance of sustained observations and modeling, and a combination of
them.

The new knowledge that has been acquired from the recent research has posed new questions and challenges. One of them
is related to the variability of upwelling.  There is a considerable gap in the space-time variability of upwelling in almost all



the regions, primarily owing to the lack of systematic long-term data sets with sufficient spatial resolution and coverage.
Second, the processes that drive upwelling are complicated in several regions and there is no consensus or quantities account
of the relative roles of each process; the role of eddies in the Mozambique channel, impact of currents along the southern
coast of Madagascar and coastally trapped waves are good examples for the dichotomy. A combination of focused modeling
studies and systematic observations are required to address such issues. The required in-situ observations need to be with
high temporal and temporal resolutions and with the capability for long-term monitoring. In addition, intensive process-
oriented observational programs are required to understand physical processes and their interconnection to the ecosystem.
Such observing strategies together with high-resolution regional and global models that include both physical and
biogeochemical/ecosystem components have the potential to develop strategies for sustainable uses of coastal resources. A
related and more sophisticated issue is the ecosystem response and fisheries. While definite progress has been made in the
Eastern Arabian Sea and off the coast of Australia, a complete picture regarding the dependence of marine biota on upwelling
is yet to emerge for the entire upwelling system along the periphery of the Indian Ocean.
**Author contributions:** PNV planned the outline of the paper and led the paper preparation. All authors contributed to the
paper preparation.
**Competing interests:** The authors declare that they have no conflict of interest.
**Acknowledgments**
This is a contribution from the Science Theme - 2 of the IIOE-2. Partial financial support from SCOR is gratefully
acknowledged. PNV acknowledges partial financial support from J C Bose National Fellowship, SERB, DST, Govt. India.
NIO contribution number of this paper is 10473. Thanks to Dr. D. Shankar for his comments on the manuscript.

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








**Figures**

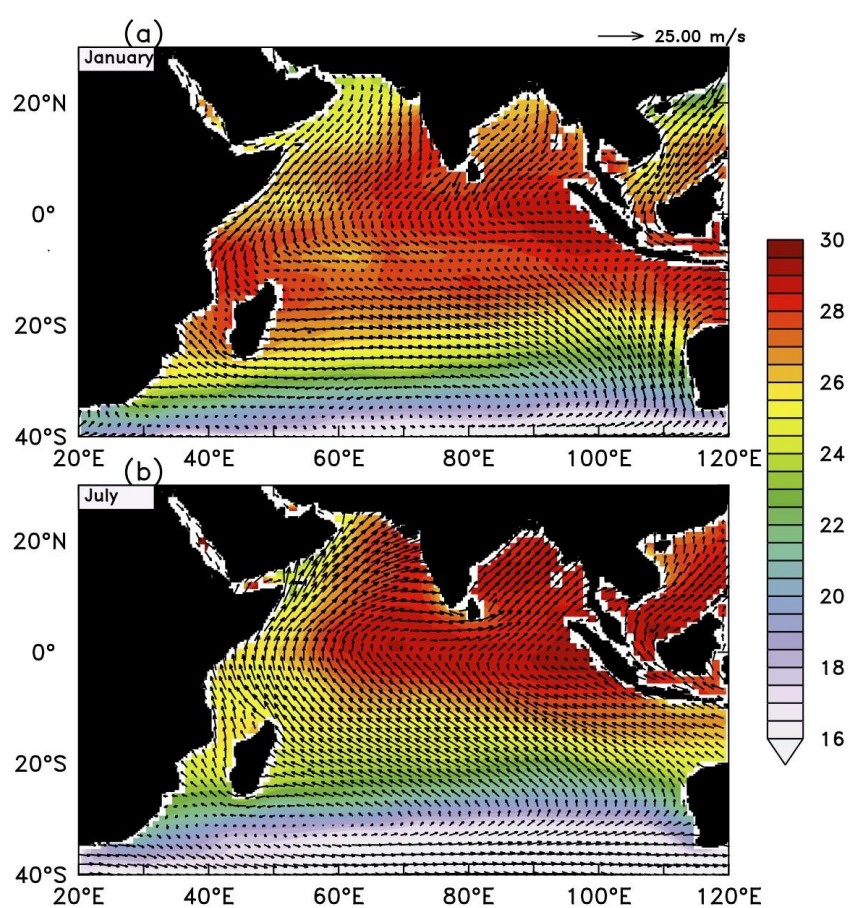


**Figure 1.** Climatological (Locarnini, 2018) SST averaged from surface to 50 m depth (shaded ) and winds (vectors m s
[-1]), data from QuikSCAT (http://apdrc.soest.hawaii.edu.) for the months of (a) January and (b) July.   The scale for SST is
given to the right of each panel and the scale vector for wind speed is given at the top of each panel.

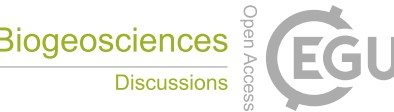







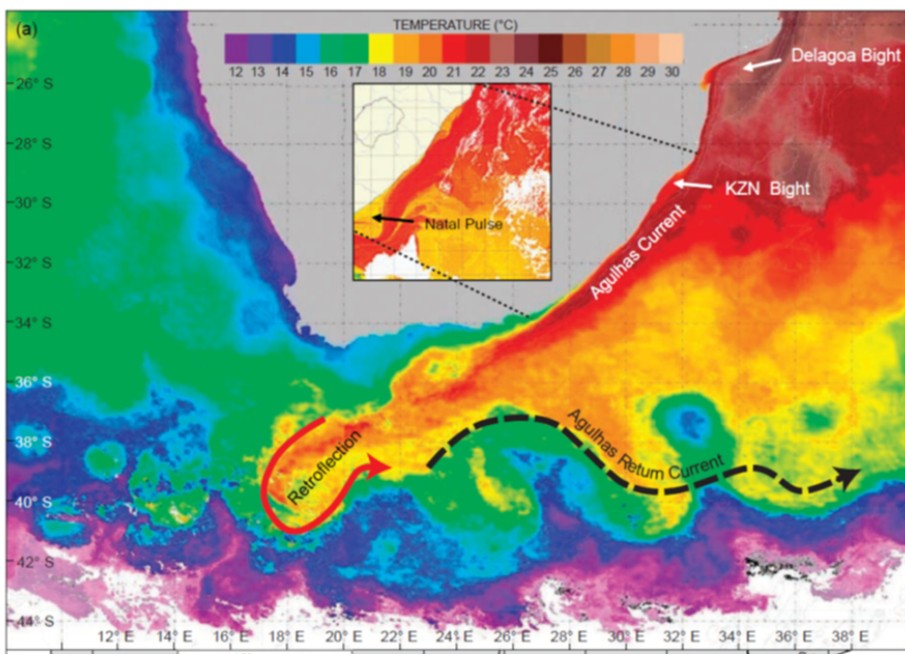




**Figure 2.** SST image highlighting the Agulhas Current flowing along the east coast of
South Africa. PE = Port Elizabeth, PA = Port Alfred. Insert highlights a south-westward
propagating Natal pulse (a singular meander in the trajectory) which has a cold core.
The shelf on the east coast is narrow with a steep continental slope. Exceptions are the
KZN Bight and the Agulhas Bank.







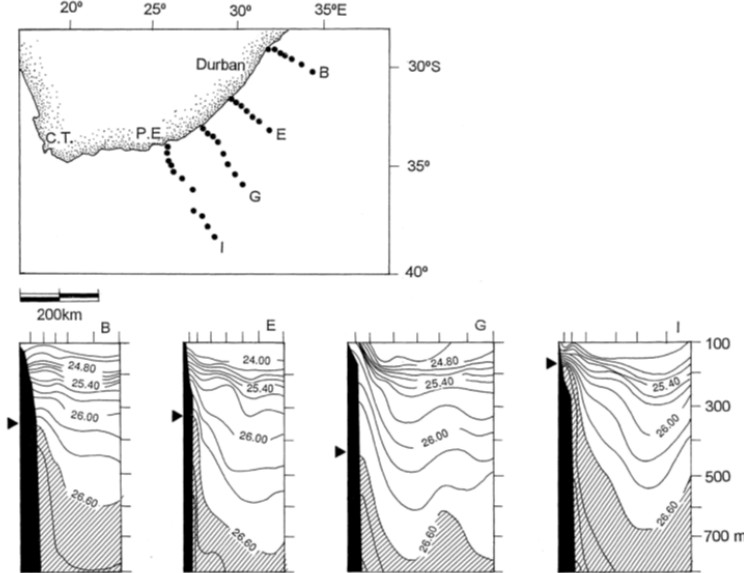


**Figure 3.** Sections across the Agulhas Current, showing sigma-t values obtained during March 1969 (after Harris and Van Foreest, 1978). All show water with a density greater than 26.60 upwelled along the inshore edge of the Agulhas Current.


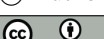

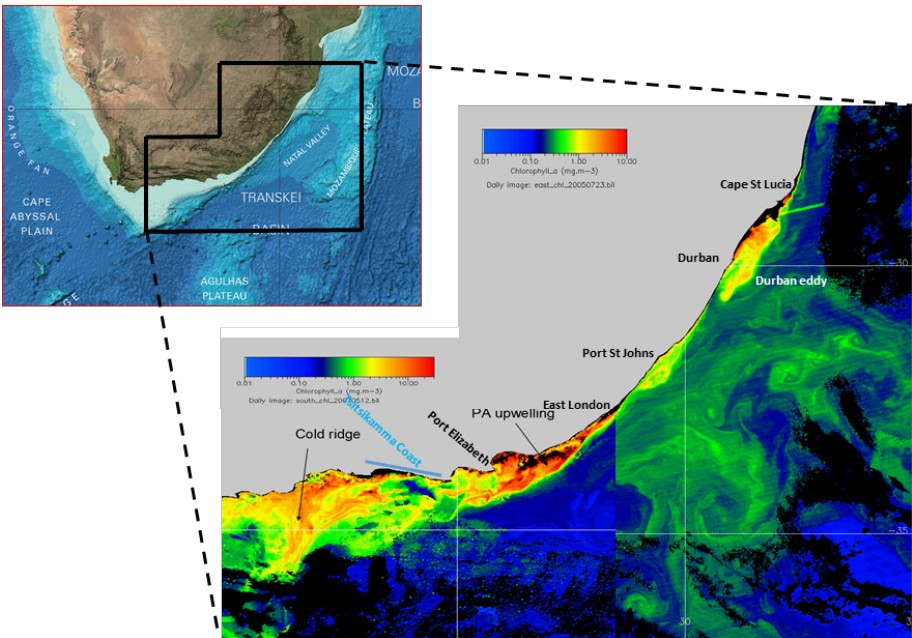

**Figure 4.** A composite chlorophyll satellite image chosen to highlight the main productivity features commonly found on the inshore edge of the Agulhas Current. Note the different chlorophyll scales applicable to the LHS and RHS parts of the composite. Highlighted are the cold ridge on the central Agulhas Bank (AB), Port Alfred upwelling extending onto the eastern AB, the Durban (break-away) eddy with a similar feature passing Port St Johns where a semi-permanent smaller cyclonic feature often exists.




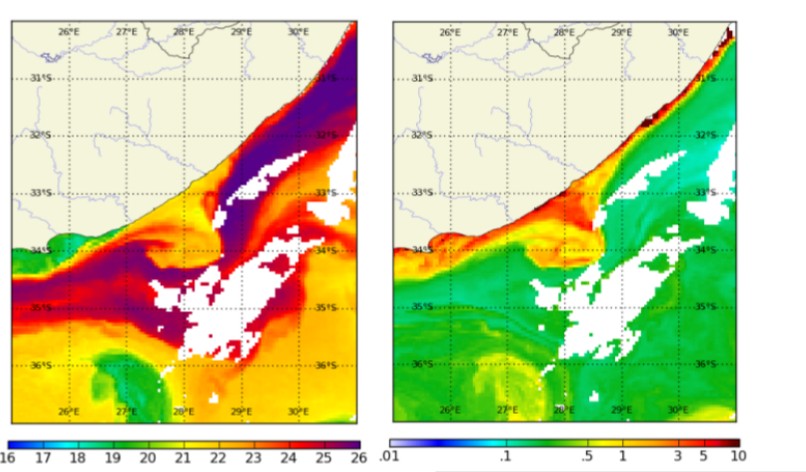


**Figure 5.** Satellite SST (LHS) and chlorophyll-a (RHS) images of a Natal pulse on 2


April 2010 off the narrow Transkei shelf. Note the high levels of chl-a on the eastern


side of the cyclone (meander) which protrude of the shelf.






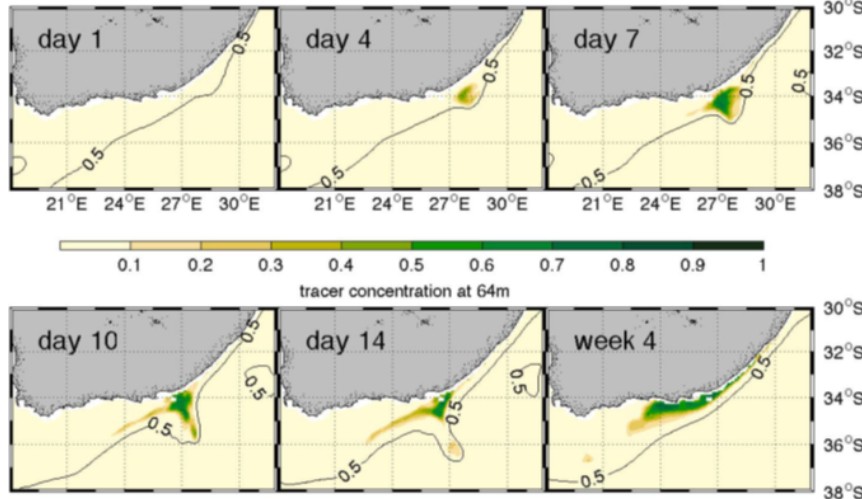


**Figure 6.** Tracer concentration at 64 m in a AGU-HYCOM to reveal shelf edge upwelling. Tracers were initialized in the Agulhas Current below 400 m over a 6-week period during a meander event in 2001 and used as a proxy for upwelling. The 0.5-m sea level contour is highlighted to show the inshore edge of the current as the meander propagates along the coast (after Malan et al., 2018).







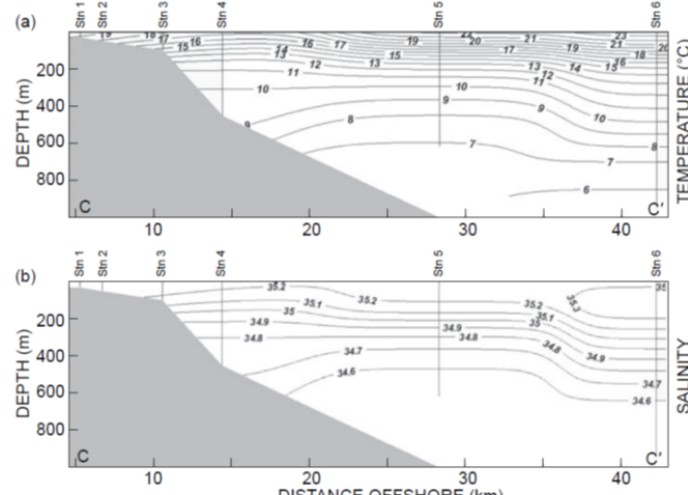


**Figure 7.** Vertical sections of CTD data collected along a trans-shelf transect off Port St Johns to a depth of 1 000 m near Port St Johns on 4 May 2005 (see Roberts et al., 2010). Both temperature (a) and salinity (b) show slope upwelling with a surface temperature of 16 °C near Station 4 in the centre of the Port St Johns–Waterfall Bluff cyclonic eddy. Graphic after Roberts et al. (2010).

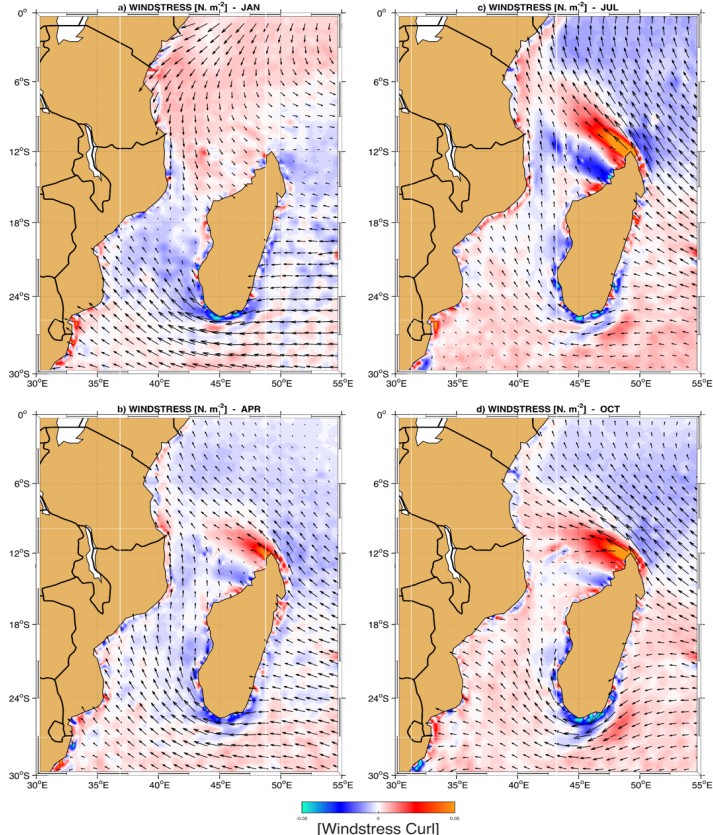


**Figure 8.** Climatological monthly means of wind stress (vectors) and wind stress curl (shading) during different seasons. Austral summer (a), fall (b), winter (c) and spring (d). Negative (blue) and positive (red) wind stress curl depict favourable upwelling and downwelling areas respectively. The data was extracted from Scatterometer Climatology of Ocean Winds (SCOW), described by Risien and Chelton (2008); mapped globally with a spatial grid resolution of 1/4∘ × 1/4∘, estimated from 10 years' period, ranging between September 1999 and August 2009, measured by NASA Quick Scatterometer (QuikSCAT).

3155





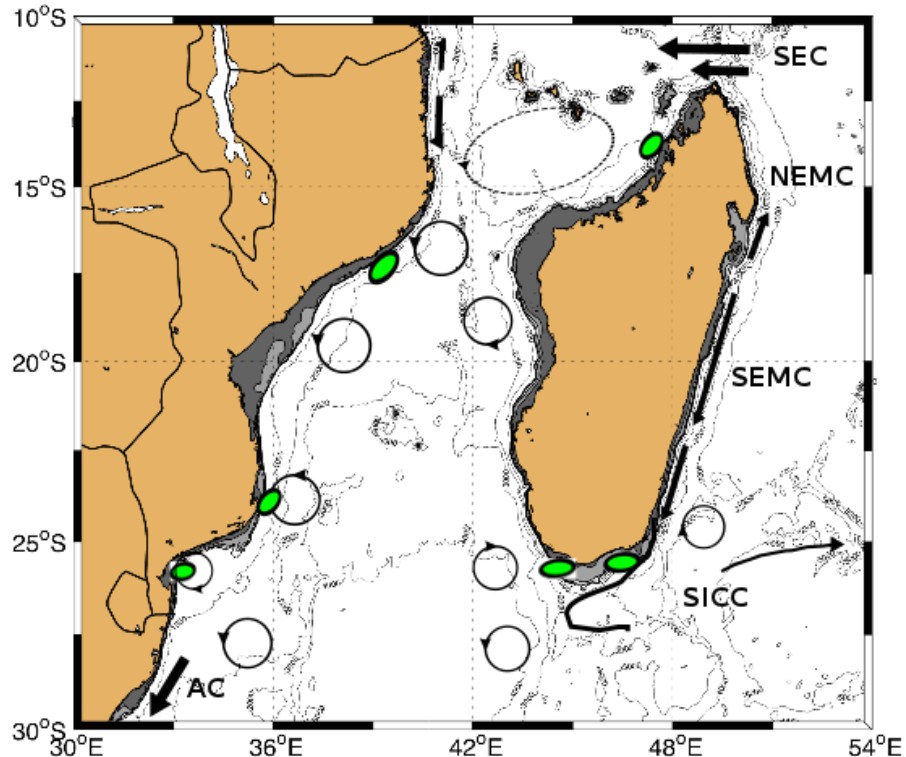

**Figure 9**. Bathymetry and major circulatory features in the Mozambique Channel and around Madagascar. Currents include the South Equatorial Current (SEC), Northeast and Southeast Madagascar Current (NEMC and SEMC), South Indian Countercurrent (SICC) and the Agulhas Current (AC). Shaded areas show the extent of the continental shelf to a depth of 200 m. Green ellipses denote upwelling areas.


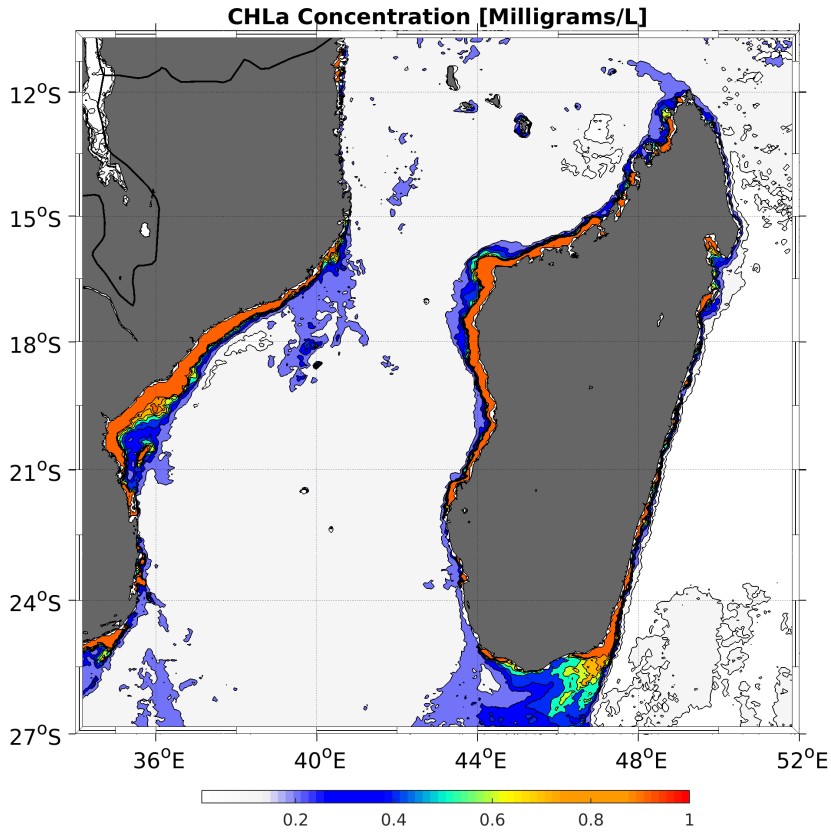


**Figure 10**. Monthly mean chlorophyll-a concentration for February 2003, derived from Moderate Resolution Imaging
Spectroradiometer (MODIS) Aqua satellite (https://oceancolor.gsfc.nasa.gov/data/aqua/). Intermediate values beyond the
continental shelf-edge highlight areas of elevated productivity off the Mozambique and Madagascar coasts that are primarily
upwelling-driven.

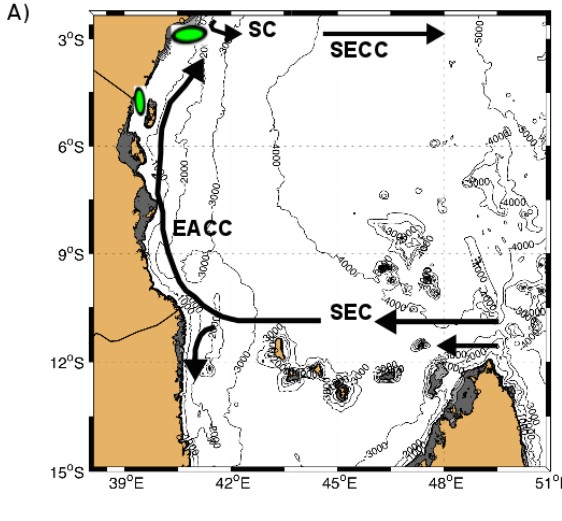


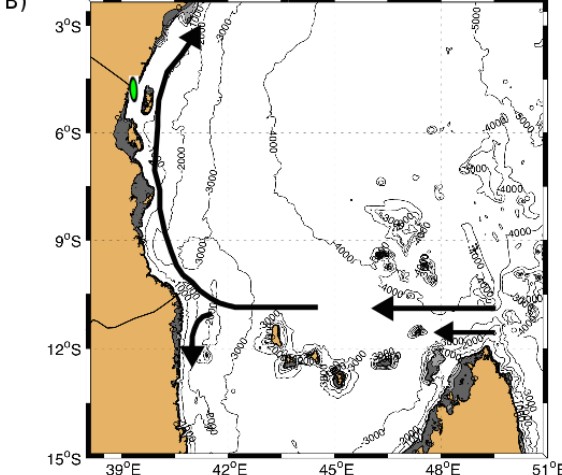

**Figure 11.** Circulation patterns during (A) the northeast monsoon and (B) the southeast monsoon, showing the Somali
Current (SC), South Equatorial Counter Current (SECC), East African Coastal Current (EACC) and the South Equatorial
Current (SEC). Green ellipses denote upwelling areas.





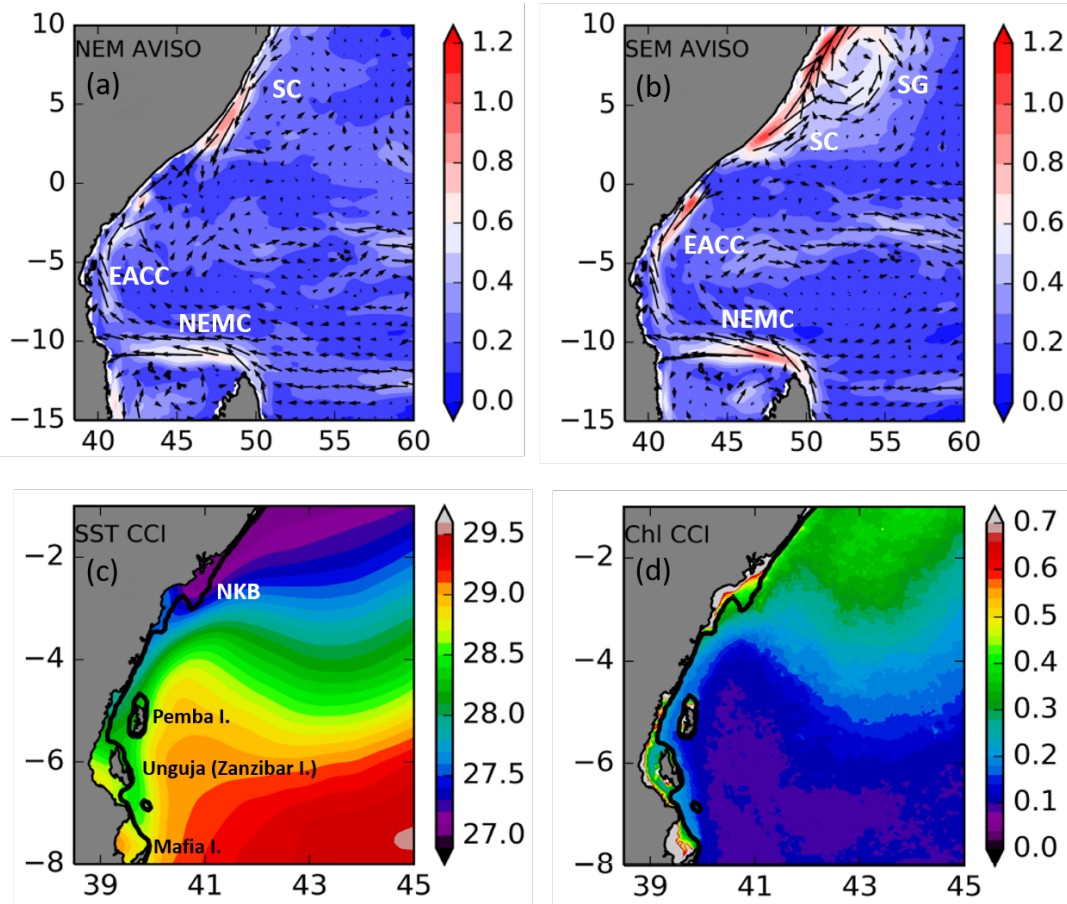


**Figure 12**. Decadal average of surface currents (m s$^{-1}$) during (a) the NEM (DJF) and (b) the SEM (MJJAS) from altimetry (AVISO), over the period 2001-2010. Features shown are the Somali Current (SC), the East African Coastal Current (EACC), the Northeast Madagascar Current (NEMC), the Somali Gyre (SG) and the North Kenya Banks (NKB). Decadal average of (c) SST (°C) and (d) surface chlorophyll (mg m$^{-3}$) during the NEM (DJF) from remote sensing (CCI) over the period 2001–2010. The thick black line represents the 200-m isobath. Adapted from Jacobs et al. (2020).




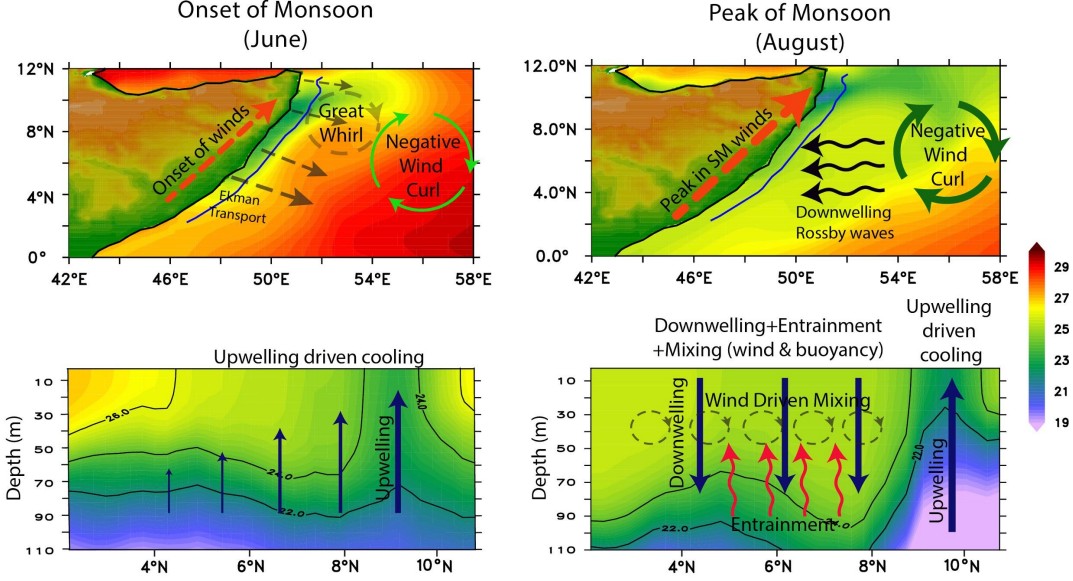


**Figure 13.** Climatological SST ( $^oC$, Panel a and b) and vertical section of temperature ( $^oC$, Panel c and d) along the vertical section aligned roughly around 1000 m isobath (blue contour along the Somali coast in the top panels) for the month of June (left panels) and August (right panels). The climatology is computed from model (Modular Ocean Model, Version 5.1) interannual simulations for 1993-2018 (reproduced from Chatterjee et al., 2019; Lakshmi et al., 2020). As the monsoon onset during early June, southwesterly winds blow along the coast of Somalia (red dashed arrow) leading to offshore Ekman transport (which is stronger in the south than the northern part; see Panel a black dashed arrows) driven coastal upwelling (upward blue arrows; Panel c). Though the offshore transport is strongest in the south, the maximum upwelling (upsloping of thermocline) is seen along the front of the Great Whirl north of ~8oN (Panel c). Notably, offshore wind stress curl turns negative south of the Findlater Jet axis favorable for open ocean downwelling. As the monsoon peaks, this negative wind stress curl radiates downwelling Rossby waves (Panel b) which propagate westward and upon reaching the Somali coast deepen the thermocline there against the upwelling favorable winds (downward blue arrows; Panel d). Further, stronger winds during peak monsoon enhance wind driven mixing which further deepen the thermocline in most parts of the Somalia coast. By late summer, the upwelling remains confined to the front of the Great Whirl in the northern part of the Somalia coast.

3198





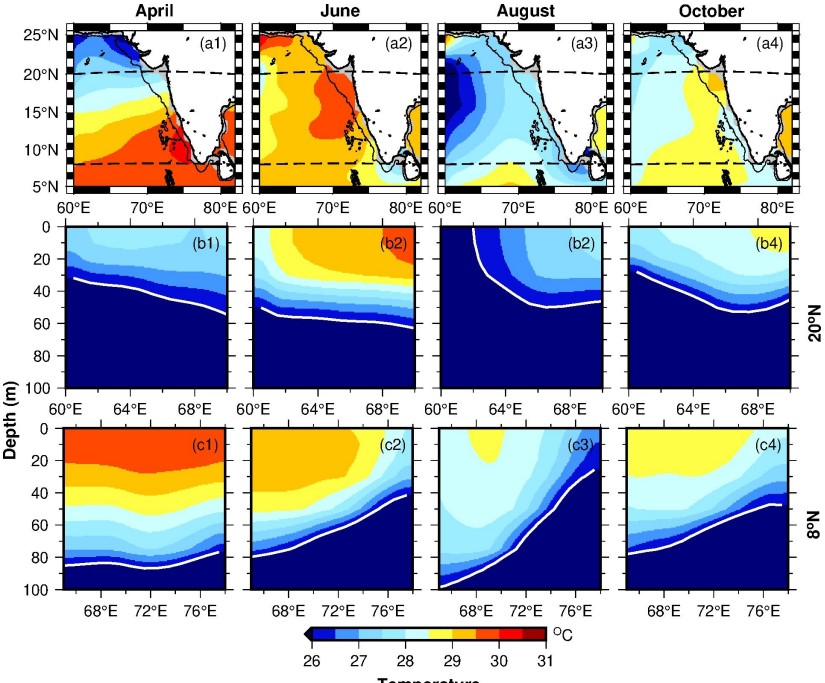

**Figure 14.** Monthly climatology of temperature from April to October. The data are from *North Indian Ocean Atlas* (Chatterjee et al. 2012). (a1-a4) Sea surface temperature from the eastern Arabian Sea. The black contour represents the 1000 m water-column depth, and the horizontal dashed lines are the 20°N and 8°N. (b1-b4) Vertical section of temperature at 20°N. (c1-c4) Vertical section of temperature at 8°N. The white contour is 26°C. The figure highlights how the upwelling evolves from pre-monsoon to post-monsoon season along the west coast of India. The upwelling sets earlier in the south and progresses slowly towards north. The upward tilt of the isopycnals, though weak, is evident at 20°N towards the end of the summer monsoon.





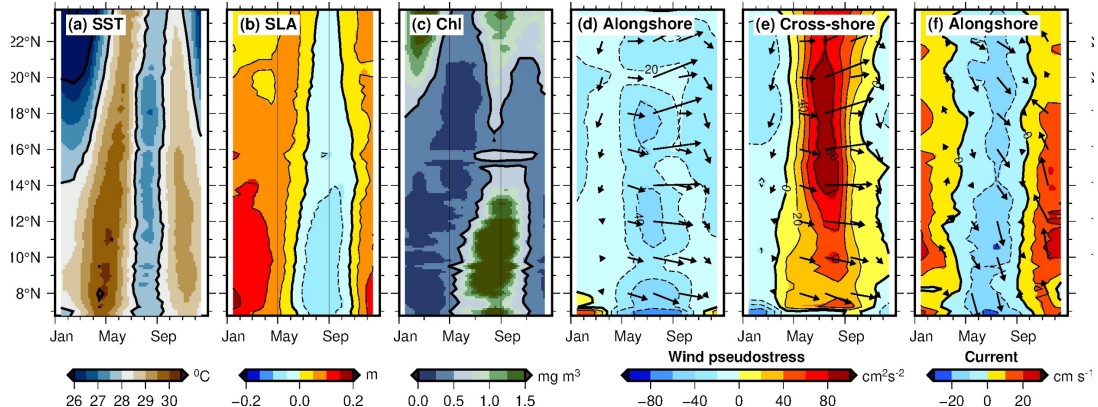

**Figure 15.** Climatology of (a) sea surface temperature from Terra MODIS, (b) sea-level anomaly from Aviso SSALTO/DUACS, (c) chlorophyll-*a* from SeaWIFS, (d) alongshore and (e) cross-shore wind pseudostress from QuikSCAT, and (f) alongshore current from OSCAR. The data were picked and the vectors were rotated based on the 1000 m contour (see Figure 15). Panels (a) and (c) are redrawn based on (Shankar et al., 2019).



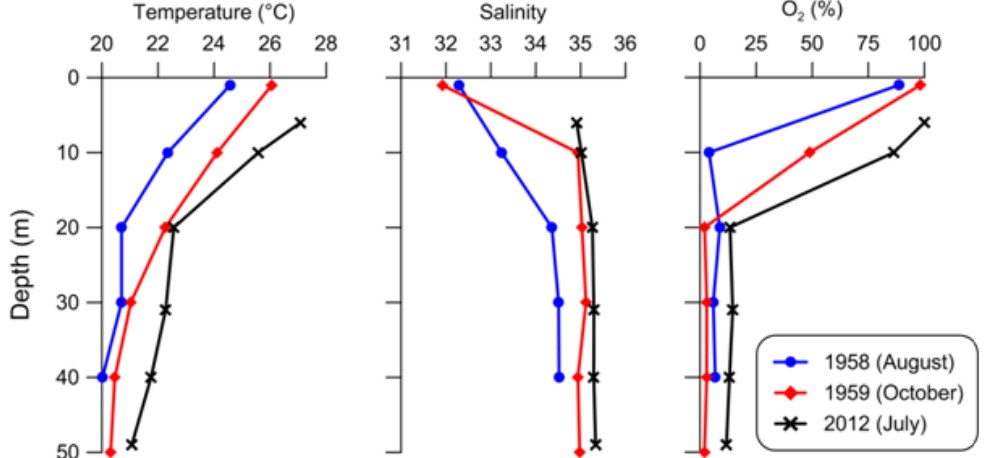

**Figure 16.** Comparison of historical profiles of temperature, salinity and dissolved oxygen corresponding to peak upwelling months over the inner shelf off Kochi, southwest coast of India (From Gupta et al., 2016).





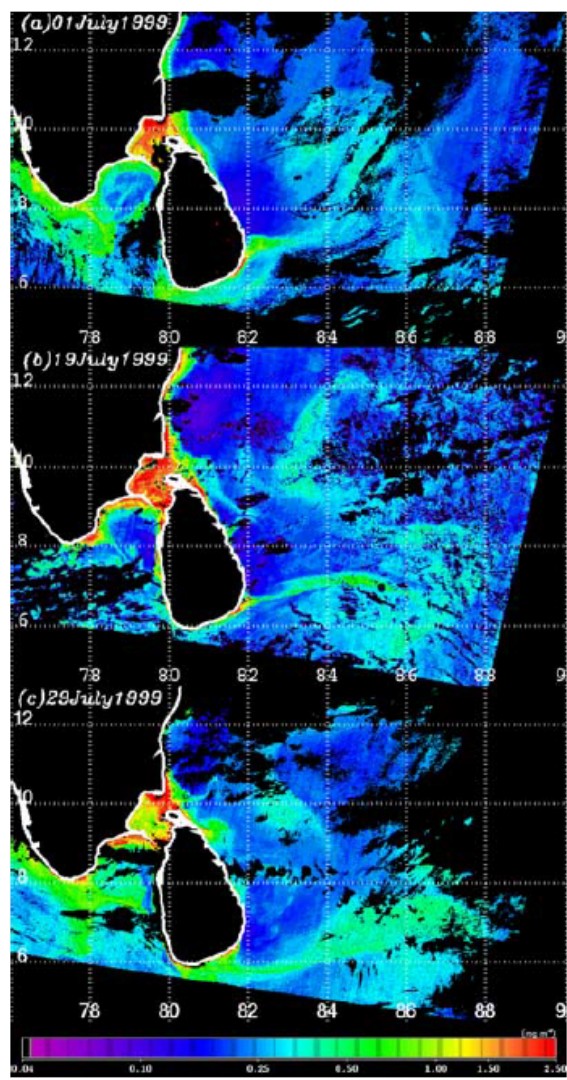

**Figure 17.** Chlorophyll a (mg m-3) images around Sri Lanka for (a) 1 July 1999, (b) 19 July 1999 and (c) 29 July 1999 obtained from OCM on board IRS-P4 Oceansat (From Vinayachandran et al., 2004).





3245

3246

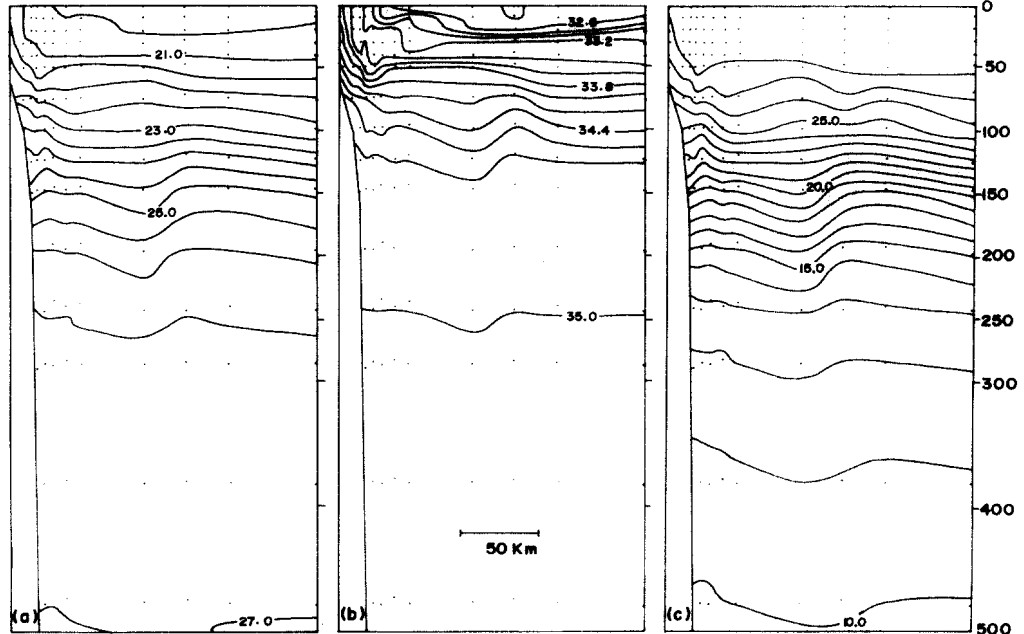

3247

3248

**Figure 18**. Hydrography along a section (normal to the coast) which lies approximately midway of the east cost of India.

(a) potential density (g cm$^{-3}$); (b) salinity (ppt); (c) temperature (°C). The scale shown in (b) also applies to (a) and (c).

(From Shetye et al., 1991**).**

3252



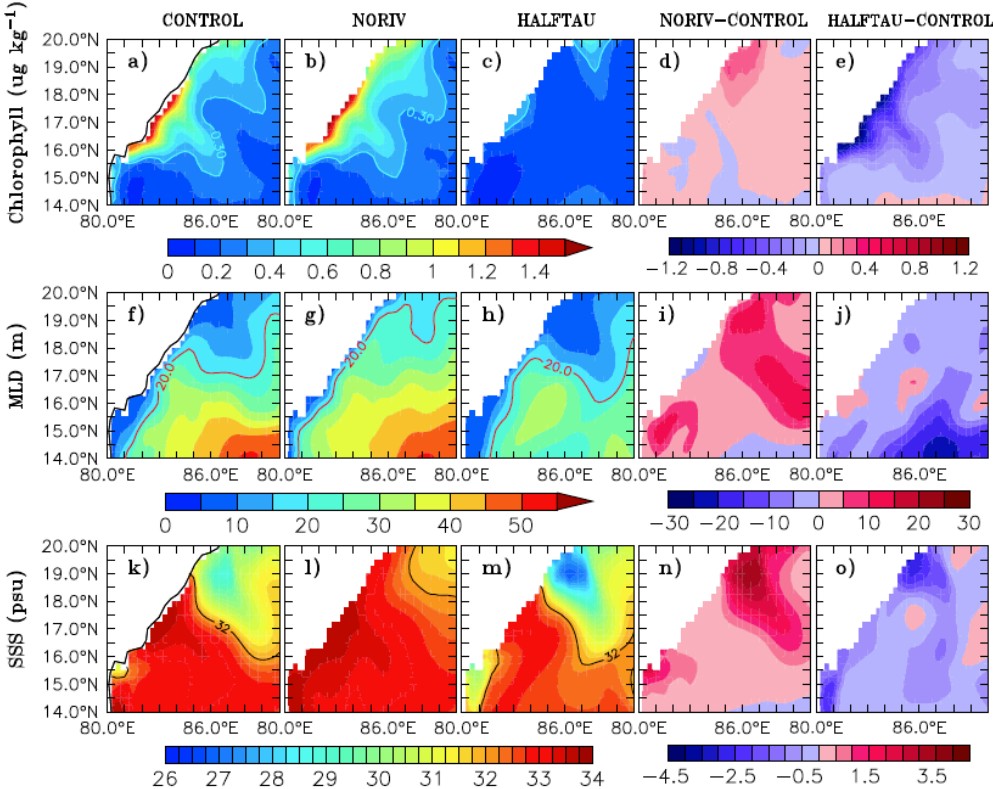

3253

**Figure 19.** Forcing mechanisms of upwelling induced chlorophyll distribution in the northwestern Bay of Bengal. Comparison of model simulated surface (a–e) chlorophyll, (f–j) MLD, and (k–o) SSS from CONTROL, NORIV, and HALFTAU experiments averaged for the month of August. Contours shown are for chlorophyll, MLD, and salinity of 0.3 ug kg21, 20 m and 32 psu, respectively. Shown are model simulations from a control run which included all the forcings (CONTROL), without river runoff (NORIV) and with the magnitude of wind stress reduced by 50% (HALFTAU) (From Thushara and Vinayachandran, 2016)

3260



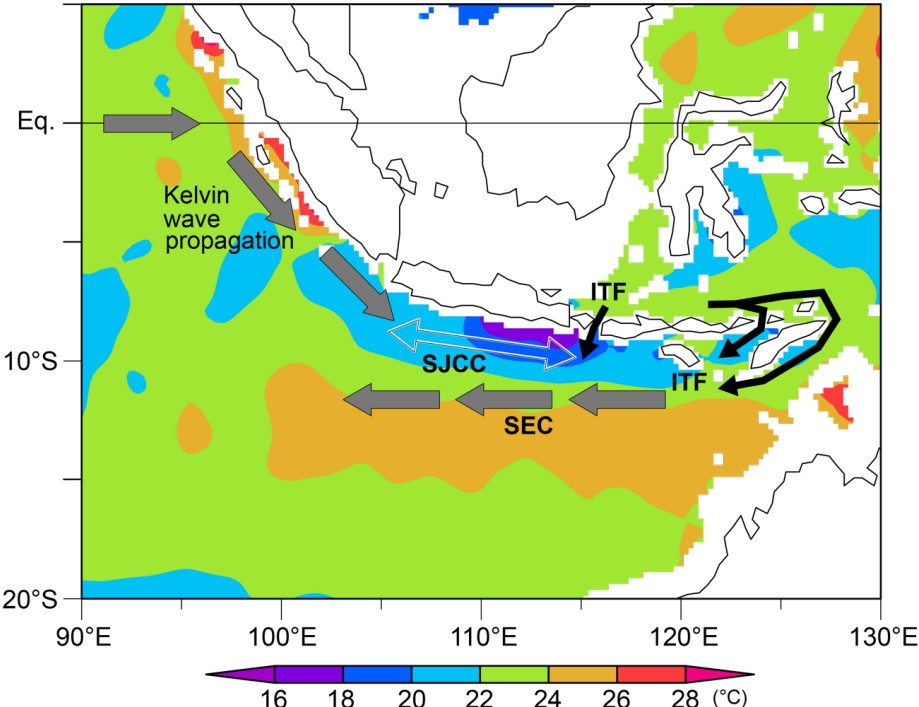

3261

**Figure 20.** Map of the Sumatra-Java upwelling region and surrounding area. Background color shade shows July-August-September mean climatological temperature at 100 m depth from World Ocean Atlas 2018 (Locarnini et al., 2018). Grey, black, and line arrows schematically indicate representative surface currents near the upwelling system (SEC: South Equatorial Current; SJCC: South Java Coastal Current; ITF: Indonesian throughflow) and a route of Kelvin wave propagation from the equatorial region down to the Sumatra and Java coasts.

3267

3268



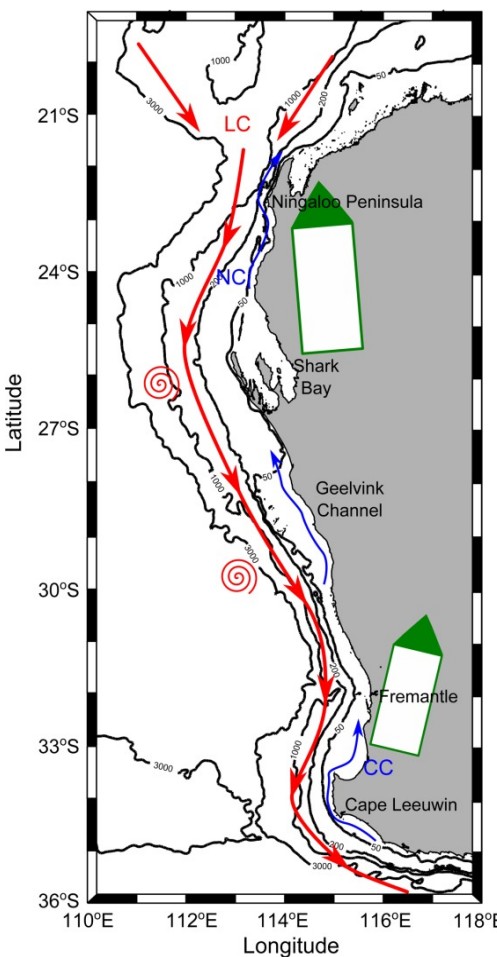

**Figure 21.** Map of the Western Australian coast with thin black contours showing the 50, 200, 1000 and 3000m isobaths. Green arrows represent mean surface winds, red arrows indicate the Leeuwin Current, red schematic vortices indicates meso-scale eddies and blue arrows indicate the Capes and Ningaloo Currents (from Rossi et al., 2013b)





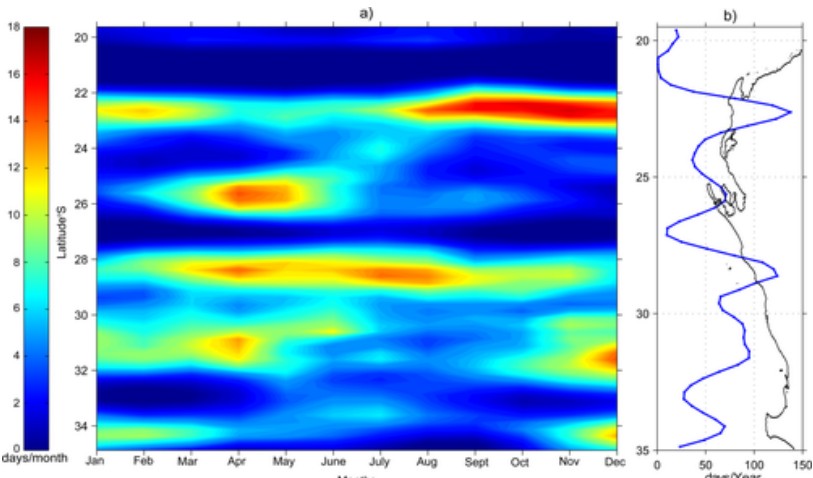

**Figure 22.** Climatological analysis of sporadic upwelling events. a) Hovmöller (latitude versus time) diagram of the mean number of "upwelling days" (CUI > 15 m/day during 3 days or more) per month and b) mean number of "upwelling days" per year, recorded from 1995-2010 (from Rossi et al. 2013b).





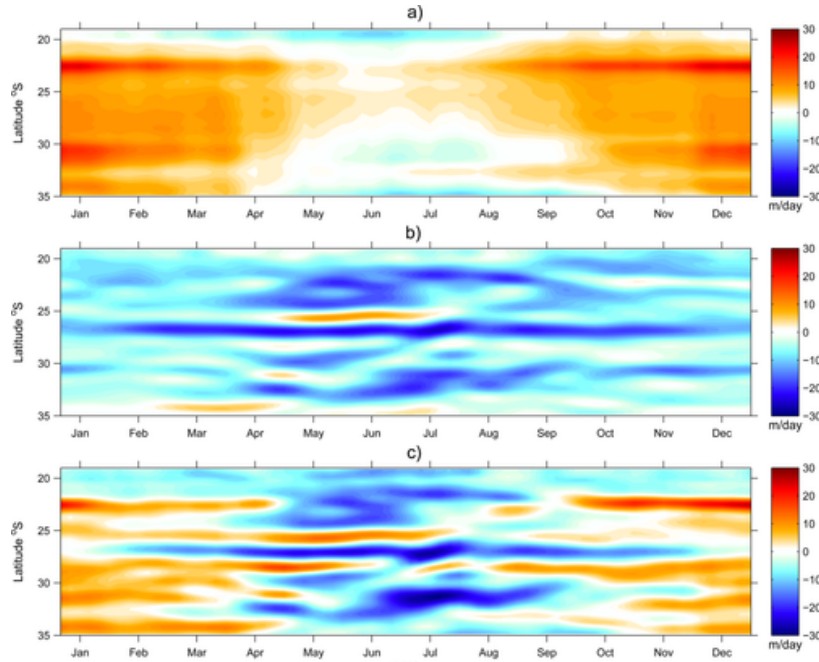

3279

**Figure 23.** Hovmöller diagrams (latitude versus time) of a) the Ekman upwelling index (m/day, equivalent to vertical velocities), b) the geostrophic upwelling index (m/day, equivalent to vertical velocities), and c) composite upwelling index (in m/day of vertical velocities, a combination of the two previous components). Red colours represent a balance of forces favouring upwelling events (from Rossi et al., 2013b).

3284





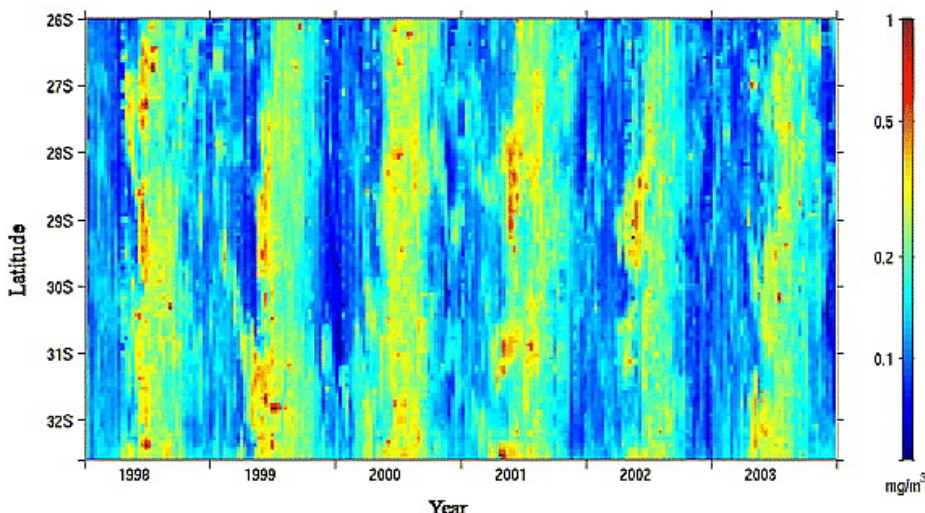

3285

3286     **Figure 24.** Annual distribution of chlorophyll estimated from SeaWiFS ocean colour data along the shelf break off the west

3287     coast of Australia from 26°- 32°S , 1998-2003 (from Koslow et al., 2008).

3288

3289


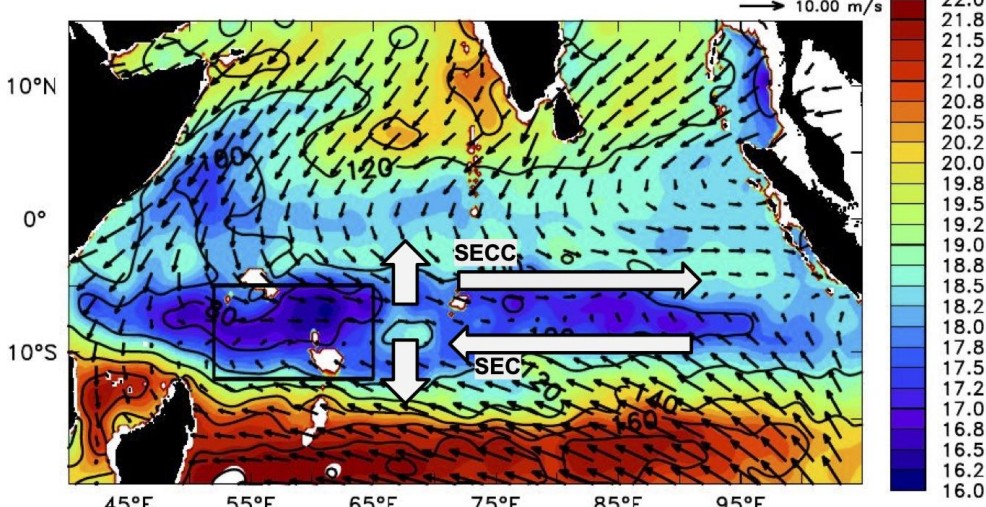

3290

**Figure 25.** Climatological (Locarnini, 2018) temperature (shading with scale shown to the right) averaged over 0-300m for
the months of January and February (shaded ) overlayed with wind vectors (m/s) from QuikSCAT Climatology (2000-
2008) and thermocline (depth of 20 degree C isotherm, m) depth as the black contour lines. Reference vector for winds
is given at the top right corner. The black box marked represents the Seychelles-Chagos Thermocline Ridge (SCTR) . The
surface flow indicated by upward and downward white arrows promotes upwelling leading to the formation of the SCTR.
The white arrows aligned left is the South equatorial current (SEC) and right is the South equatorial counter currents (SECC).
Redrawn after Vialard et al. (2009)

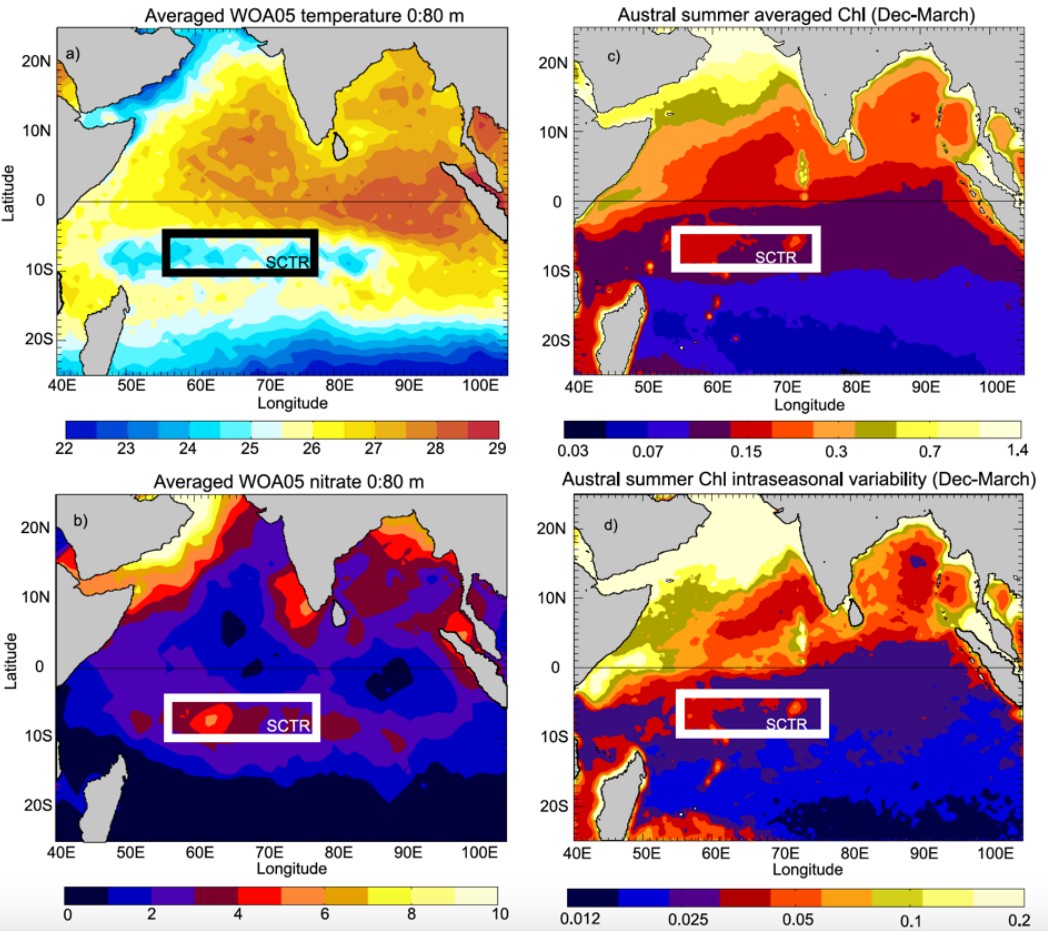

3298

**Figure 26**. Annual World Ocean Atlas (2005 ) (a) temperature and (b) nitrate concentration (in mmol N m$^{-3}$) averaged

between the surface and 80 m in the Indian Ocean. (c) SeaWiFS seasonal mean during austral summer (December–March)

(mg m$^{-3}$). (d) Intraseasonal variability of SeaWiFS Chl during austral summer estimated by the averaged RMS of (Chl-Chl*)

between December and March of years 1998–2007.   (from Resplandy et al. (2009).

3303

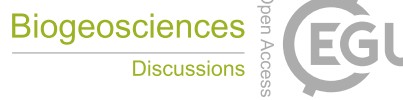

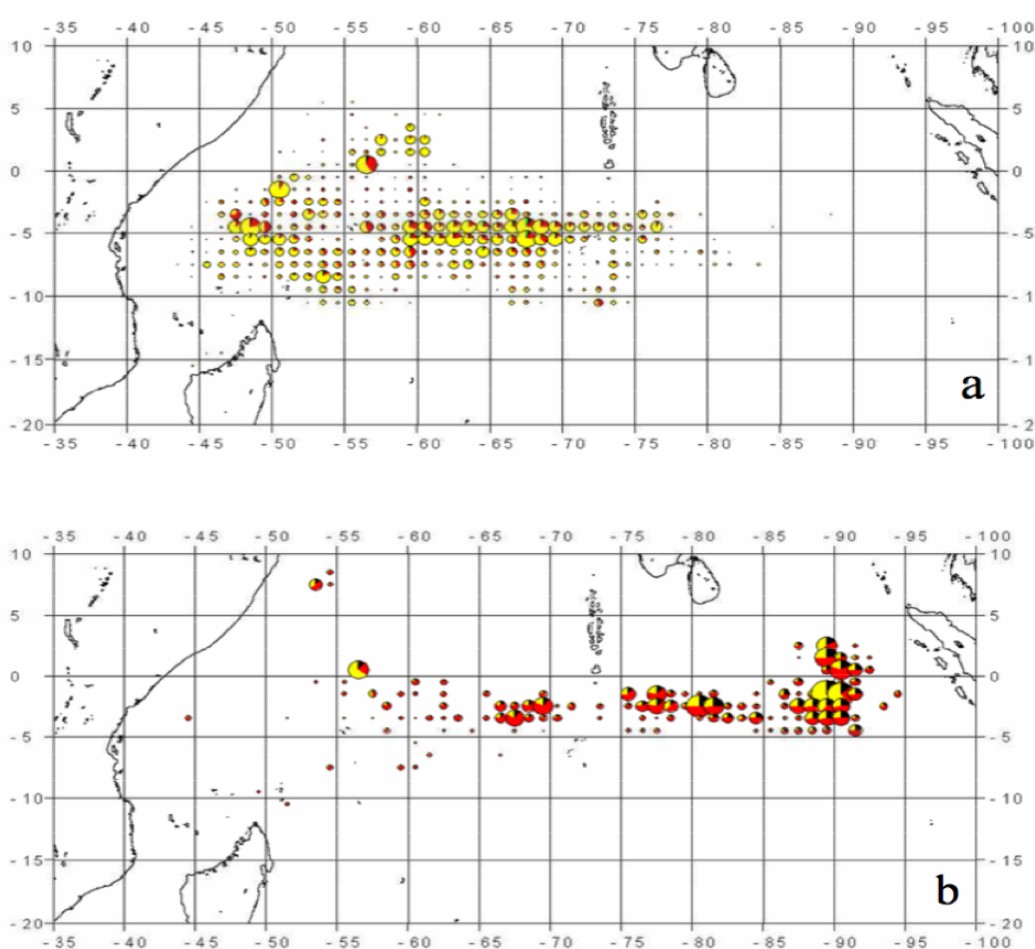

3304

**Figure 27.** Tuna catch in the Indian Ocean during the 1997/1998 IOD event (bottom panel) compared to catch in normal

years (top panel). From Robinson et al. (2010), Copyright Inter-Research 2010.





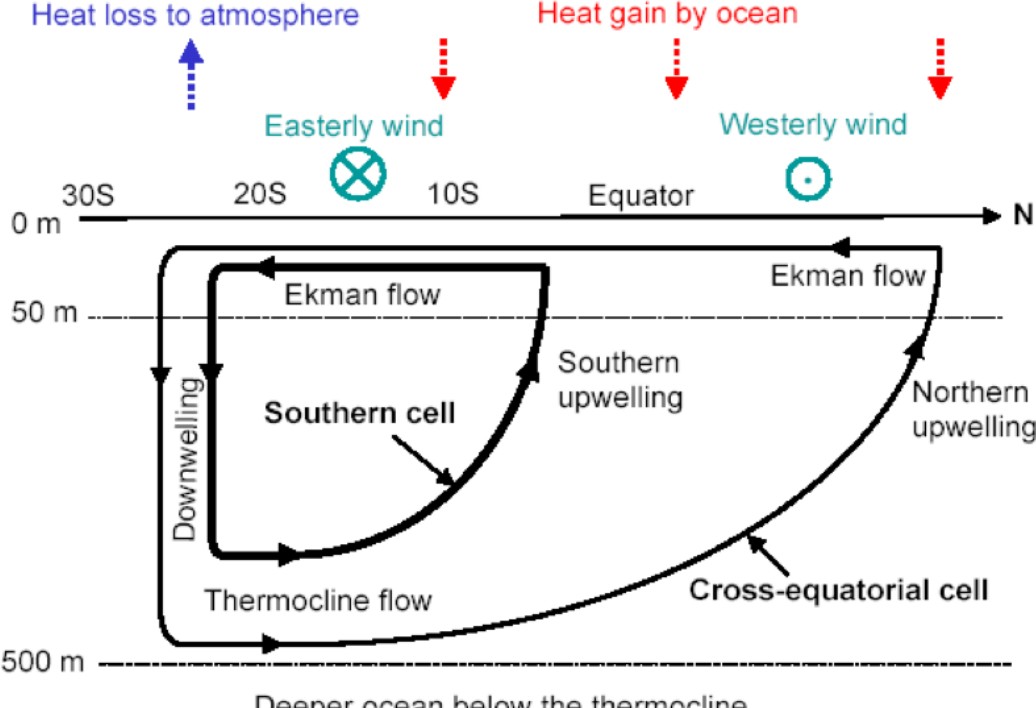

**Figure 28.** Conceptual illustration of the time-mean meridional overturning circulation of the upper Indian Ocean that consists of a southern and a cross-equatorial cell. The time-mean zonal wind and surface heat flux are also shown schematically. This flow is believed to partially supply the cross-equatorial thermocline flow. From Lee (2004)