# Peer review of "Reviews and syntheses: Physical and biogeochemical processes associated with upwelling in the Indian Ocean"

_Biogeosciences, 2020_

## Author Comment (AC1)

We thank Dr. Vinu Valsala for his comments and suggestions on the manuscript. The comments are reproduced in black font and the replies are given in blue below each comment.

This is an excellent and timely review of the Indian Ocean's physical and biogeochemical processes associated with the upwelling zones. There has been a considerable amount of observational and modeling studies in this area and are very well covered in this comprehensive review.

However, the Biogeochemistry about the carbon cycle in the upwelling zones is somewhat found less emphasized despite a considerable number of studies and research has gone into it. These studies are also useful to highlight potential gap areas in the observations of the Indian Ocean carbon cycle, pCO2, and acidification. A few are mentioned below, kindly incorporate them also in this review and synthesis effort.

We completely agree with Dr. Vinu Valsala that items in the above list are important topics. However, they are somewhat beyond the scope of this review. A reviewer has also recommended that the description be focussed on upwelling regions so as to reduce the length of the paper and increase its readability. Nevertheless, we have made our best efforts to include them. Please see replies to specific questions given below:.

*Lines-1034:1035: Being 1034 limited with very few studies on carbon dynamics over both east and west coasts, the temporal evolution of surface ocean acidification is still not clear.*

Takahahsi et al, (2014) compilation show the clear seasonal cycle of pH in the western Arabian Sea. Further, modeling studies show that the western Arabian Sea has been acidified from a pH of 8.12 (in 1960) to a pH of 8.05 (in 2010). The trend in pH over the western Arabian Sea is due to contributions from dissolved inorganic carbon (DIC) and SST at a value of 109% and 16%, respectively. The effect of alkalinity (ALK) is to buffer the trend in pH by -36% while salinity contribution is only +7%. Collectively, DIC and ALK contribute up to 73% to the net pH trend. SST warming alone contributes another 16%, which is quite alarming considering the intense warming of the western Indian Ocean (Roxy et al., 2016). This calls for the sustained observational efforts required for the Indian Ocean upwelling zones to monitor and model ocean acidification.

We have already cited Takahasi et al (2014) in the sentence previous to this line (i.e., line 1033). Detailed discussion on the factors acidifying the oceans is beyond the scope of this synthesis as our main focus is on ecosystem response. Moreover, our interest is more on coastal upwelling systems while the suggested papers mostly result from the studies conducted in the open ocean. To make this more clear, the term 'surface ocean acidification' at lines 1034-1035 will be changed to 'coastal acidification'.

*Lines-1401-1403: Efforts to develop and improve biogeochemical models of the 1401 upwelling systems are also in progress (e.g., Sreeush et al., 2018).*

In addition, Sreeush et al., (2020) showed improving biogeochemical models in upwelling zones using inversion of surface observation such as pCO2 and imposed constraints that can cascade through solubility and the biological pump in the upwelling

zones to retrieve valuable subsurface ocean parameters such as community compensation depth in models.

 Thanks for this suggestion. We have updated the text as follows:

"Recent in-situ observations in the Sumatra-Java upwelling region conducted during the IIOE-2 period indicate different phytoplankton composition and assemblages between upwelling and non-upwelling regions (Gao et al., 2018) and physical and biological processes that determine aragonite saturation state (Xue et al., 2016). Efforts to develop and improve biogeochemical model of the upwelling systems are also in progress (e.g. Sreeush et al., 2018). These researches on biogeochemistry are important to understand key processes operating in the Sumatra-Java upwelling system. However, these results are rather fragmentary at this stage and integrated studies on biophysical interactions, ecosystem dynamics, and marine food web in the Sumatra-Java upwelling region are needed."

*Lines-1665:167:  biogeochemical modeling results indicate that neither phytoplankton biomass nor carbon export from the euphotic zone changes significantly in response to seasonal and interannual variability of the SCTR thermocline depth (Resplandy et al., 2009)*

*Lines-1673:1674: Finally, there is still considerable uncertainty in whether the Indian Ocean is a net source or sink of carbon to the atmosphere because the variability in pCO2 fluxes across the air-sea interface is poorly constrained by existing observations, particularly in active upwelling zones like the SCTR*

Other studies also highlighted the variability of seasonal and interannual cycles of sea-to-air CO2 fluxes, pCO2 in the upwelling regions of the Indian Ocean. Valsala and Maksyutov (2013) identified that the interannual variability of western Arabian Sea sea-to-air CO2 fluxes and pCO2 are complementarily controlled by ENSO and IOD-related forcing and dynamics. In the south of Sri-Lanka, the interannual variability of the carbon cycle is controlled by variability in wind-induced upwelling dynamics of the dissolved inorganic carbon (Valsala and Maksytov, 2013). The western Arabian Sea is also home to intra-seasonal variability in sea-to-air CO2 fluxes and pCO2 due to the eddy dynamics associated with Great Whirl and Southern Gyre (Valsala and Murtugudde, 2015) as verifiable with limited observations of surface ocean pCO2. More observational efforts are required to understand such fine-scale variability of pCO2 in the Indian Ocean.

Recent studies pointed out that the south Java-Sumatra coast also exhibits interannual variability in sea-to-air CO2 fluxes and pCO2 due to upwelling variability linked to IOD, as identifiable from gap-filled observations using neural networks (Valsala et al., 2020, Lanschutzer al., 2016). The sea-to-air $CO_2$ fluxes, surface ocean partial pressure of $CO_2$ (p$CO_2$), the concentration of dissolved inorganic carbon (DIC), and ocean alkalinity (ALK) range as much as ±1.0 mole m$^{-2}$ yr$^{-1}$, ±20 μatm, ± 35 μmole kg$^{-1}$, and ± 22 μmole kg$^{-1}$ within 80$^o$E-105$^o$E, 0-10$^o$ S due to IOD. The DIC and ALK are significant drivers of p$CO_2$ variability associated with IOD. The roles of temperature (T) and biology are found negligible. A relatively warm T and extremely high freshwater forcing make the southeastern tropical Indian Ocean carbon cycle variability submissive to DIC and ALK

evolutions in contrast to the tropical eastern Pacific where changes in DIC and T dominate the $pCO_2$ interannual variability (Valsala et al., 2020).

Thanks for these suggestions. We do agree that these are very important topics. However,  We are unable to include details about CO2 fluxes in this paper as this is beyond the scope of this review. Reviewer-2 has also suggested minimizing this section.

*Lines-733-736: On the other hand, studies with the help of more complex models, in the last couple of decades, suggest that phytoplankton growth in this region are prone to iron limitation (Wiggert et al., 2006; Wiggert and Murtugudde, 2007) and also likely to be silicate stressed (Koné et al., 2009; Resplandy et al., 2011).*

Anju et al, (2020) used a 13-component silicate limiting biogeochemistry ecosystem model to study the impact of Silicate limitation in the western Arabian Sea. The new production represents 80% of the total primary production in the AS and implicitly controls 70% of total zooplankton production annually. The regenerated production augments small phytoplankton (by ~50%; e.g., flagellates) and small zooplankton (by ~20%; e.g., ciliates) growth with negligible effects on large phytoplankton (e.g., diatom) and predatory zooplankton (e.g., copepods). The diatom production remains within the observed range due to silicate limitation, which is fundamental in the models for realistic simulation of sub-surface chlorophyll maxima.

Thanks for pointing out the silicate limitation. We have revised the paragraph as follows:

"Observations of the coastally upwelled water off Oman during US JGOFS indicates iron stress with N:Fe ratio ranging between 20,000-30,000 during the early phase of the summer monsoon. Later, this Fe limitation was also confirmed based on in-situ observations off the Oman shelf by Moffett et al. (2007) and Naqvi et al. (2010). Naqvi et al. (2010) further argued that this iron limitation fueled a shift in the phytoplankton communities from diatoms to smaller phytoplankton species which favours vertical export to the offshore deep ocean via lateral advection by the offshore currents (McCreary et al. 2013). In contrast, Rixen et al. (2006), based on sediment transport observations, suggests intense grazing in the silicon-rich near-shore upwelled water limits the diatom bloom off Oman coast. Further, modelling studies based on coupled physical-biogeochemical ocean models, suggests that iron (Wiggert et al., 2006 and Wiggert and Murtugudde, 2007) and silicate (Koné et al., 2009) are the most limiting nutrients that inhibit the growth of diatoms off the coast of Arabia.

**References:**

Anju M., Sreeush M.G, **Valsala V.,** Smitha B.R., Hamza F., Bharathi G., Naidu C.V., Understanding the role of nutrient limitation on plankton biomass over the Arabian Sea via 1-D coupled biogeochemical model and Bio-Argo observations, Journal of Geophysical Research: Oceans, 125: e2019JC015502, June 2020, DOI:10.1029/2019JC015502, 1-28

Landschützer, P., Gruber, N., and Bakker, D. C. E. ( 2016), Decadal variations and trends of the global ocean carbon sink, *Global Biogeochem. Cycles*, 30, 1396–1417, doi:10.1002/2015GB005359.

Sreeush M. G., R. Saran, **V. Valsala**, S. Pentakota, K.V. S.R. Prasad, R. Murtugudde (2019): Variability, trend and controlling factors of Ocean acidification over Western Arabian Sea upwelling region, Marine Chemistry, doi.org/10.1016/j.marchem.2018.12.002.

Sreeush, M. G., Vinu Valsala, Halder Santanu, Sreenivas Pentakota, K.V.S.R. Prasad, C.V. Naidu, Raghu Murtugudde (2020), Biological production in the Indian Ocean upwelling zones - Part 2: Data-based estimates of variable compensation depth for ocean carbon models via cyclo-stationary Bayesian Inversion., Deep-Sea Research Part II: Topical Studies in Oceanography, 179, 2020,https://doi.org/10.1016/j.dsr2.2019.07.007.

Takahashi, T., Sutherland, S.C., Chipman, D.W., Goddard, J.G., Newberger, T., Sweeney, C., (2014). Climatological distributions of pH, $pCO_2$, Total $CO_2$, Alkalinity, and CaCO3 Saturation in the Global Surface Ocean. ORNL/CDIAC-160, NDP-094. Carbon Dioxide Information Analysis Center, Oak Ridge National Laboratory, U.S. Department of Energy, Oak Ridge, Tennessee. https://doi.org/10.3334/CDIAC/OTG.NDP094.

**Valsala, V.,** S. Maksyutov, (2013), Interannual variability of the air-sea CO2 flux in the north Indian Ocean, Ocean Dynamics,  DOI:10.1007/s10236-012-0588-7, 1-14

**Valsala, V**., and R. Murtugudde, (2015), Mesoscale and Intraseasonal Air-Sea CO2 Exchanges in the Western Arabian Sea during Boreal Summer, Deep Sea Research-I, doi:10.1016/j.dsr.2015.06.001

**Valsala, V.**, M. G. Sreeush, and K. Chakraborty, (2020), IOD impacts on Indian the Ocean Carbon Cycle, Journal of Geophysical Research, https://doi.org/10.1029/2020JC016485

**Valsala V**., (2009), Different spreading of Somali and Arabian coastal upwelled waters in the northern Indian Ocean: A case study, J. Oceanography, Vol.65, Page 803-816.

---

## Author Comment (AC2)

**Comment on "Reviews and syntheses: Physical and biogeochemical processes associated with upwelling in the Indian Ocean"**

**by Puthenveettil Narayana Menon Vinayachandran, Yukio Masumoto, Mike Roberts, Jenny Hugget, Issufo Halo, Abhisek Chatterjee, Prakash Amol, Garuda V. M. Gupta, Arvind Singh, Arnab Mukherjee, Satya Prakash, Lynnath E. Beckley, Eric Jorden Raes, Raleigh Hood**

We thank the reviewer for his comments and suggestions. In the following, the comments by the reviewer are reproduced in black font and replies are given in blue.

**General comments:**

This review provides a comprehensive overview on the physical dynamics in upwelling regions and the resulting interplay with biogeochemical processes in the entire Indian Ocean based on observations and model results. Variability of upwelling has a strong impact on climate-relevant trace gases as well as biogeochemical processes both having an effect in many ways on one of the most densely populated regions. Upwelling processes in the Indian Ocean are still less well understood than in the other major oceans, therefore the topic is highly relevant.

The manuscript is well organized, structured and (mainly) easy to read. The authors address main upwelling regions in the Indian Ocean giving a historical background, an overview about recent observations (characterstic of upwelling and impact on physical parameter), present status of modelling these upwelling systems and their upwelling mechanisms, and their impact on marine ecosystems and biogeochemistry. My main concerns are related to the figures, which lack in care and precision and make the text sometimes hard to follow. I would also highly recommend a schematic figure for the different monsoon phases, that summarizes the main upwelling regions together with the monsoonal winds.

We are thankful to the reviewer for reading the manuscript carefully and offering several comments. We have revised the manuscript to enhance its readability and the major change is the organization of each of the upwelling systems into same organizational structure, based on the comments by the other reviewer. We have also included an additional figure (Figure 1B) showing upwelling regions and a schematic of circulation. The figures have been revised as per the suggestions provided by the reviewer. Reply to each of the specific comments are given below.

I recommend publication of this manuscript after minor revision. I will leave my comments below in the specific comments.

**Specific comments:**

-line992: Upwelling cannot be driven by primary production, it is the other way round.

This has been corrected as : "Summer upwelling, which drives by high primary production, …"

Figure 2: Port Elizabeth (PE), Port Alfred (PA) and Agulhas Bank are mentioned in the figure caption but not in the figure. It would be easier to follow the descriptions in the text, if some geographical points would be marked in the figure or the latitude-longitude needs to be mentioned in the text.

PE and PA are marked in the figure. Labels for the places are placed at their respective geographic location.

Figure 3: "C.T" and "P.E." are marked in the figure but not explained in the text/figure caption.

Corrected: "C.T and P.E indicates Port Elizabeth and Cape Town respectively." This has been added to the figure caption.

Figure 4: X-axes (longitude) and y-axes (latitude) should be labelled. The font/scales of the colorbars are hard to read/recognize. Acronyms "LHS" and "RHS" should be explained.

The X and Y axes have been labelled. The labels of the color bar has been revised for better readability. The acronyms LHS and RHS have been removed.

Figure 5: Please mark the Transkei shelf in the figure.

Transkei shelf is marked on the top left panel of the figure.

Figure 6: I would recommend to label the x-axes of the lower panels as well.

This figure is now replaced with a different figure.

Figure 8: The titles of the figures are hard to read and a scale vector for the wind stress is missing. I also would recommend to mark the respective months ("JAN", "JUL", "APR", "OCT") in the figures. The colorbar is very tiny as well as the font/scales. The unit for the wind stress curl is missing as well.

The corrections suggested by the Reviewer has been made to all panels in the Figure and a scale vector has been place below bottom left panel.

Figure 9: This figure would improve a lot and the paragraph relating to this figure would be much easier to follow if some geographical points would be marked in the figure such as: Angoche (or Nampula), Sofala Bank (or Ponta Zavora), Inhambane, Delagoa Bright, Maputo.

Geographical points have been printed on the Figure at appropriate locations.

Figure 10: Please mark in the figure the geographical points you are referring to in the text.

All geographical points that are referred in the text have been marked on the figure.

Figure 11: Please mark in the figure the geographical points (Tanzania, Kenya) you are referring to in the text. Please label the currents also in the lower panel.

Labels of Tanzania and Kenya have been added to the figure.

Figure caption: correct "southeast" to "southwest". Please refer to the contour lines.

The correction 'southwest' has been made. Contours are described in the figure caption.

Figure 12: 12a, b: What does the color shade show? A scale vector for the surface currents is missing. 12b: correct "SEM" to "SWM".

The color panels are appropriately labelled in the revision. A separate vector for the currents are not provided because the color panel clearly shows a scale for the speed.

12c, d: Please also mark the respective monsoon phase (NEM) in the figures as in 12a,b. The text would be easier to follow if country borders are drawn in the figures. I would also recommend to label the colorbars with the respective unit.

The respective seasons have been marked on the figure as well as clearly stated in the Figure caption. The country borders have been marked on the maps. Color bars have been labelled along with respective units.

Figure 13: Please mark a-d in the figure as you are referring to in the figure caption and in the text.

The markings a-d has been made on the figure.

Figure 15: The x-axes of 15c should have the same scale as 15a-b, d-e.

 The x-axis of 15c is the same as in other panels of the figure in the revised version.

Figure 17: The font of the colorbar is hard to read and the unit is missing.

The font of the color bar has been revised to make it readable and the  unit has been placed.

Figure 18: Could you give the approximate latitude of the hydrographic sections? "approximately midway of the east coast" is not very precise.

The approximate latitude has been mentioned in the figure caption.

**Technical corrections:**

Thanks to the reviewer for pointing out several minor errors. All of them have been corrected.

-line 121: typo: correct "Northeasterly" to "northeasterly": Corrected

-l156: typo: correct "IIOE-2 (2015-2010)": Corrected

-l236: typo: "day1" should say "day$^{-1}$" : Corrected

-l289: typo: correct "Current" to "current" : Corrected

-l429: Do you mean "vertical suction"?: : Corrected

-460: The acronym MC has not been explained yet.: The acronym has been defined

-l482: typo: correct "water" to "Water" : Corrected

-l495: The terms "northeast orientation" and "southeast orientation" are misleading. Better: "northeasterly winds" and "southeasterly winds" or "southwest directions" and "northwest direction":        Corrected as  "the winds are northeasterly along the southeastern coast, and southeasterly  along the southwestern coast".

-l515: The acronym EMC has not been explained yet. :  EMC has been replaced with "East Madagascar Current"

-l565: typo: correct "southeast" to "southwest" : Corrected

-l566 and more: correct "SEM" to "SWM": Corrected

-l594: typo: correct "North-East Monsoonal" to "northeast monsoonal" or use the acronym "NEM": Corrected

-l635: correct "move" to "moves": Corrected

l636: correct "recirculate" to "recirculates": Corrected

-l643: correct "forms" to "form": Corrected

-l654: correct "migrate" to "migrates": Corrected

-l664: correct "result" to "results": Corrected

-l680: typo: delete "the" in "the another": Deleted

-l702: typo: correct "study" to "studies" : Corrected

-l710: correct "of" to "off": Corrected

-l747: correct "crosshore" to "cross-shore": Corrected

-l788: typo: correct "21oC": Corrected

-l790: correct "in the shelf" "on the shelf" : Corrected

-l930: The term "peak" is associated with a maximum, maybe "minima" is a better term. : Corrected

-l998: The paragraphs have the wrong order: paragraph 2.7.5 precedes 2.7.4.: Corrected

-l1044: "Figure 16" should be in bold. : Corrected

-l1225: typo: correct "Joshnson" to "Johnson" : Corrected

-l1229: correct "Oxygen minimum zone" to "oxygen minimum zone" or "OMZ" : Corrected

-l1249: "belief" – I would prefer "assumption": "belief" is more appropriate here

-l1294: correct "Srilanka" : Corrected

-l1356: correct "appear" to "appears" : Corrected

-l1425: "Other early studies…" This sentence is very long and hard to follow, please rephrase.: The sentence has been split into two for easy reading.

-l1635: Figure 26 does not show wind.: Changed to Figure 1

-l1740: correct "tolarger": Corrected

-1747: correct the semicolon: Corrected

-l1752: correct "are caused" to "is caused": Corrected

-l1771: What do you mean with "high temporal and temporal resolution", please rephrase. : Corrected as "high spatial  and temporal resolutions"

---

## Author Comment (AC3)

Reply to the comments by Reviewer#2

We thank the reviewer for reading through the manuscript and offering several comments and suggestions. In this reply, comments by the reviewer are given in black font and replies in blue.

Reviews and syntheses: Physical and biogeochemical processes associated with upwelling in the Indian Ocean

Vinayachandran et al. described twelve upwelling regions in the Indian Ocean. The description starts with upwelling systems associated with coastal currents in the southwestern part of the Indian Ocean, followed by those largely driven by the Asian Monsoon and ends with those associated with coastal currents off Australia in the southeastern part of the Indian Ocean. Additionally, the authors discussed the Seychelles-Chagos Thermocline Ridge (SCTR), which is an upwelling system that develops in the open ocean between 5 and 15°S in the western part of the Indian Ocean.

Writing a review and synthesis paper on upwelling systems in the Indian Ocean is a great idea and I am very pleased by the efforts of the authors. However, to my understanding, a review and synthesis paper should provide a comprehensive overview without overloading the reader with details and aspects, which are interesting but not directly related to the topic. Considering this demand, the individual chapters of the paper reveal pronounced disparities. For instance, there are chapters such as the Introduction and chapter 2.6 which do not reflect the state of the art whereas e.g. chapters 2.1, 2.9, and 2.12 include so much additional information that they could most be stand alone papers on the Agulhas Current, the Bay of Bengal and the oceanography along the Australian west coast. Hence, to my understanding, the current version of the manuscript needs a major revision.

We thank the reviewer for reading the manuscript carefully and offering several constructive comments. We have revised the paper in the light of the reviewer's comments and all the issues raised by the reviewer have been addressed. Reply to each of the reviewer's comments are given below.

A revised version of the paper should include an Introduction clarifying terms, which thereafter are used coherently throughout the manuscript. Furthermore, chapters on the individual upwelling regions should focus on upwelling and exclude regional characteristics, which are not directly related to upwelling. It would also be helpful to summarize discussions on upwelling mechanisms since after reading about it from two different perspectives (modeling vs. observations), the general conclusion often remains unclear. In sum, I believe that uniform subdivision of all chapters into three sections such as Background, Mechanisms, and Productivity and Ecosystem, as seen e.g. in chapter 2.4, would help to shorten the manuscript, which is with 128 pages quite long. Simultaneously, it would increase the readability of the manuscript.

We agree with the reviewer on including definition of all clarifying terms; in the revised version of the paper, all such terms have been defined appropriately. We have also limited the descriptions to those regional features that are relevant to the upwelling. In the revised version, uniform subdivision of chapters into the general framework

suggested by the reviewer based on Background, Mechanisms and Productivity and Ecosystem has been adopted for describing the upwelling systems.

In the following I will provide arguments and suggestions whereas I will start with the Instruction and continue, thereafter, with chapter 2.

**Introduction**

The authors subdivided the Introduction into three sections describing upwelling (i), biogeochemical implication of upwelling (ii) and the study area (iii). This is a convincing structure but the section on upwelling contains only two references; one from 1905 and the other one from 1937. This is insufficient to introduce the reader into our current understanding of upwelling, which includes the formation of coastal parallel jets, meso-scale eddies and offshore-directed filaments, as well as interactions of these features with remotely and locally trigged waves. After reading an introduction to upwelling, I would expect that the reader exactly knows what the authors mean with 'classical upwelling dynamics' (see lines 745 – 746). Furthermore, it is crucial to define the following terms in order to better understand the description of the individual upwelling regions and, in particular, the differences between these regions:

Ekman upwelling, wind-driven coastal upwelling, core upwelling, shelf-edge upwelling, slope upwelling, divergent upwelling, dynamic upwelling, topographic-driven upwelling, dynamic boundary upwelling, current-driven upwelling, upwelling wedge, current-induced upwelling, upwelling Kelvin waves and upwelling node.

We respect the reviewer's suggestion that the introduction could be updated to introduce the current understanding of the topic.  The introduction has been revised to include coastal parallel jets, mesoscale eddies and offshore-directed filaments and the interaction of them with wave propagations. The terms listed by the reviewer has also been defined in the paper.

To emphasize the role of upwelling on marine ecosystems and ecosystem services such as fisheries, the authors refer to Eastern Boundary Upwelling Systems (EBUS) in the second section of the Introduction. This is convincing since it emphasizes the relevance of upwelling in general, as well as the uniqueness of the Indian Ocean where such an EBUS is not established. However, this approach suffers from the poor discussion of EBUS. In the following, a couple of references are listed which provide to my understanding nice overviews on EBUSs:

We thank the reviewer for pointing out the relevant reference on EBUS. The description on the EBUS has been expanded into a separate paragraph to include the features of upwelling systems mentioned in the review. References about production in the EBUS has also been included in the revision.

Kämpf, J. & Chapman, P. in Upwelling Systems of the World    (Springer International Publishing, 2016).

Messié, M., Chavez, F.P., 2015. Seasonal regulation of primary production in eastern boundary upwelling systems. Progress In Oceanography, 134, 1-18.

Chavez, F.P., Messie, M., Pennington, J.T., 2011. Marine Primary Production in Relation to Climate Variability and Change. Annual Review of Marine Science, 3, 227-260

Messié, M., Ledesma, J., Kolber, D.D., Michisaki, R.P., Foley, D.G., Chavez, F.P., 2009. Potential new production estimates in four eastern boundary upwelling ecosystems. Progress In Oceanography, 83, 151-158.

Carr, M.-E., 2001. Estimation of potential productivity in Eastern Boundary Currents using remote sensing. Deep Sea Research Part II: Topical Studies in Oceanography, 49, 59-80.

Carr, M.-E., Kearns, E.J., 2003. Production regimes in four Eastern Boundary Current systems. Deep Sea Research Part II: Topical Studies in Oceanography, 50, 3199-3221.

Bakun, A., 2017. Climate change and ocean deoxygenation within intensified surface-driven upwelling circulations. Philosophical Transactions of the Royal Society A: Mathematical, Physical and Engineering Sciences, 375, 20160327.

Brady, R.X., Lovenduski, N.S., Alexander, M.A., Jacox, M., Gruber, N., 2019. On the role of climate modes in modulating the air–sea $CO_2$ fluxes in eastern boundary upwelling systems. Biogeosciences, 16, 329-346.

Zuidema, P., Chang, P., Mechoso, C.R., Terray, L., 2011. Coupled ocean-atmosphere-land processes in the tropical Atlantic. Joint edition of the newsletter of the Climate Variability and Predictability Project (CLIVAR) exchanges and the CLIVAR variability of the American Monsoon System Project (VAMOS), 55, 12 - 14.

Chavez, F.P., Toggweiler, J.R., 1995. Physical estimates of global new production: The upwelling contribution. In C.P. Summerhayes, K.-C. Emeis, M.V. Angel, R.L. Smith, B. Zeitschel (Eds.), Upwelling in the ocean, modern processes and ancient records (pp. 313-320). Chichester: Wiley & Sons.

Regarding the study area, I would suggest to go beyond mentioning papers and describe climate, ocean currents, and climate anomalies such as Madden-Julian Oscillation, El Nino Southern Oscillation (ENSO) and the Indian Ocean Dipole.

This indeed is a very good suggestion. However, we feel this this would make the paper much too long. Moreover, there have been reviews and synthesis of the above topics published earlier (Xie et al., 2009), the most recent one by Hood et al., (2015).

Furthermore, it would be extremely helpful to include maps showing the general circulation and primary production as well as the location of the upwelling region, which the authors discuss in chapter 2. Also, in this context it is crucial to define and standardize the used terms. For instance, select only one term for: northeast monsoon; NE monsoon; northeast Monsoon; NEM; North-East Monsoon which should also indicate whether it refers to boreal or austral regime. This is just one of many examples making it difficult to follow the description of the upwelling systems.

A new figure (Figure 1B) has been included which shows a schematic of the circulation during summer and winter. The background of this Figure is satellite derived chlorophyll data which presents the major upwelling regions along the coast. We have also adhered to the terms of northeast monsoon (NEM) and southwest monsoon (SWM) throughout the paper.

**Chapter 2** Coastal Upwelling Systems

General remarks:

Since the Seychelles-Chagos Thermocline Ridge (SCTR) is an upwelling system that develops in the open ocean, I would suggest adding an additional chapter on upwelling systems in the open ocean. In this chapter the SCTR could be discussed, as well as the equatorial upwelling in the Indian Ocean. So far this upwelling region has not sufficiently been considered by the authors, even though same papers on this upwelling system have been cited.

We have moved SCTR to section 3 under open ocean upwelling. We could not find sufficient material to warrant a separate section for the equatorial Indian Ocean. The equatorial upwelling is restricted to the eastern boundary and this has been included in Section 2.11

Instead of describing the remaining elven upwelling regions in dependence of their geographical location, it would also be helpful for the reader if the author could subdivide chapter 2 into sections, which relate to e.g. driving forces. Such overarching sections could be: (i) monsoon-driven upwelling systems, (ii) upwelling systems associate with coastal currents and (iii) upwelling systems strongly influenced by freshwater fluxes such as those in the Bay of Bengal and off Sumatra/Java. This is just a suggestion, but I am convinced that defining overarching types of upwelling will help the reader to better understand upwelling in the Indian Ocean.

We were unable to find a better organization in terms of driving forces. Therefore, we have opted to retain the present organization.

Within chapters 2.5 to 2.9 contents are mixed up. For instance, the authors discussed iron-limitation in chapter 2.5 line 697 – 699 and cited Naqvi et al. 2010. According to the reference list, this is a paper entitled 'Marine hypoxia/anoxia as a source of CH4 and N2O' but I guess the authors mean:

Naqvi, S.W.A., Moffett, J.W., Gauns, M.U., Narvekar, P.V., Pratihary, A.K., Naik, H., Shenoy, D.M., Jayakumar, D.A., Goepfert, T.J., Patra, P.K., Al-Azri, A., Ahmed, S.I., 2010. The Arabian Sea as a high-nutrient, low-chlorophyll region during the late Southwest Monsoon. Biogeosciences, 7, 2091-2100.

Thanks for pointing out this mistake. This has been rectified in the revised version. We have also organized these sections and repetitions have been eliminated.

This paper discusses data measured off the Arabian Peninsula and along a transect across the central Arabian Sea. Hence, if the authors want do discuss iron limitation, it

belongs to chapter 2.6. Furthermore 'iron limitation' is a controversially discussed issue. Similar to results derived from numerical models, also studies based on observations are coming to opposing conclusions regarding iron limitation. To my understanding, such opposing conclusion should have also been considered by the authors. See e.g. Rixen, T., Goyet, C., Ittekkot, V., 2006. Diatoms and their influence on the biologically mediated uptake of atmospheric CO2 in the Arabian Sea upwelling system. Biogeosciences, 3, 1 - 13.

The reviewer has rightly pointed out that the conclusion of Naqvi et al. (2010) was based on observations made along a transect from the coast of Arabia to the central Arabian Sea. Hence, including this statement in the Section for the "Somali upwelling system" may cause confusion. This statement was made to elaborate on possible mechanisms that limit the phytoplankton bloom in this region. We noted that most of the observation-based hypotheses on productivity limitation were a spin-off from the JGOFS observation and therefore, limited to the coast of Arabia and in the central Arabian Sea (see, for example, Rixen et al., 2006; Naqvi et al., 2010 ). Whereas, model studies suggest that the iron limitation is not only limited to the coast of Arabia but also along the coast of Somalia (Wiggert et al., 2006; Wiggert and Murtugudde, 2007). Hence, suggesting that the conclusion based on observation collected off the coast of Arabia may also be true for the coast of Somali. The citation of Naqvi et al. (2010) was, in fact, in this sense mentioned in the Somali region.
However, we agree that the statement was misleading a bit, primarily due to lack of a link between the discussion on model studies and the observation-based studies. Since, in the revised version, we intend to re-organize the structure of the sections by combining the observation and model-based studies, all the limiting processes discussed in the literature (the original version discussed quite extensively about other limiting processes as well) will be arranged coherently for a better flow and more definite conclusions. Nevertheless, we will include Rixen et al. (2006) and its conclusions in the revised texts.

In chapter 2.6 (line 806) the authors wrote: 'a detailed discussion on biophysical interactions along this coast is discussed in Section 2.7.5.' It makes really no sense to discuss biophysical interactions along this Arabian Coast in a chapter on upwelling along the Indian coast! Please spilt chapter 2.7.4 and 2.75 and discuss biogeochemical implications in association with the respective upwelling regions. In chapter 2.9.2 the authors discuss general biogeochemical characteristics of the Bay of Bengal, but it remains open to which extent this relates to upwelling! To my understanding this is also a main problem of chapters 2.1 and 2.11 as mentioned already before.

We agree with this suggestion by the reviewer and the contents of sections 2.6 and 2.7 have been re-organized to be self-contained. Chapters 2.1, 2.9 and 2.11 also will be revised to eliminate topics that are not relevant to upwelling.

In the following, I will focus on chapter 2.6 and 2.8 because these chapters suffer from missing contents and describe upwelling system, which are comparable to EBUS. Upwelling in the western Arabian Sea is, perhaps in addition to the one off Somalia, the only major upwelling system in the Indian Ocean with an impact on marine productivity,

which is comparable with those of EBUS. The upwelling region off Indonesia, in turn, is the only large upwelling system associated with an eastern boundary current in the Indian Ocean.

The authors introduce chapter 2.6 with a hint to the enormous work, which was carried out in the framework of JGOFS with respect to upwelling in the western Arabian Sea. This included modelling and fieldwork whereas the fieldwork had also a strong focus on biogeochemical implications of upwelling. I guess that > 6 'Arabian Sea special issues' have been published. Sharon Smith edited many of these special issues and worked a lot on zooplankton dynamics in upwelling system. Other groups studied phytoplankton successes including their link to nutrient dynamics in surface waters, primary production and export production, as also recently summarized by Rixen et al. 2019a,b and references given therein. Hence, moving this topic to chapter 2.7.5 is incomprehensible and weakens the discussion on ecosystem shifts. To my opinion, ecosystem shifts as well as overfishing, are among the largest threads to upwelling systems and aspects which should be considered in the discussion of each upwelling system in case the availability of data allows it. Goes and Gomes published a number of papers on ecosystem shifts, and the one belonging to chapter 2.6 have been cited in chapter 2.7. Please see also their latest work: Goes et al. 2020.

We thank the reviewer for these thoughtful comments. Yes, we agree that the separation of biogeochemical response to the Section 2.7.5 distracted the discussion. In the revised version, we propose to scrap section 2.7.5 and bring the relevant discussions in Section 2.5-2.7.
Regarding a comprehensive review of the studies based on US JGOFS observations, the review paper by Schott and McCreary (2001) and Hood et al. (2017) provide great detail on these findings. Hence, in this manuscript, we tried to focus on some of the aspects that were not quite highlighted in the earlier reviews and keep repetition as minimum as possible. The same was stated upfront in the introductory paragraph of this section in the original submission.
We agree that the topics like "ecosystem shifts" and "overfishing" were not fairly discussed in the original submission. Considering that these are the pressing challenges in the fast-changing world, we also believe that a better discussion is necessary. We will try to address this in the revised version and would like to thank the reviewer for providing some of the important references in this
Goes, J.I., Tian, H., Gomes, H.d.R., Anderson, O.R., Al-Hashmi, K., deRada, S., Luo, H., Al-Kharusi, L., Al-Azri, A., Martinson, D.G., 2020. Ecosystem state change in the Arabian Sea fuelled by the recent loss of snow over the Himalayan-Tibetan Plateau region. Scientific Reports, 10, 7422.

Rixen, T., Gaye, B., Emeis, K.-C., 2019. The monsoon, carbon fluxes, and the organic carbon pump in the northern Indian Ocean. Progress In Oceanography, 175, 24-39.#

Rixen, T., Gaye, B., Emeis, K.C., Ramaswamy, V., 2019. The ballast effect of lithogenic matter and its influences on the carbon fluxes in the Indian Ocean. Biogeosciences, 16, 485-503.

Chapter 2.10. According to authors 'The upwelling off the southern coasts of Sumatra and Java Islands is a remarkable eastern boundary upwelling system (EBUS) in the Indian Ocean'. To my understanding this is not correct. Even though upwelling associated with an eastern boundary current develops off Indonesia, it reveals fundamental differences to other EBUS going beyond aspects discussed by the authors. The development of EBUSs is associated with the development of deserts on land. In contrast to this coevolution, Indonesia is characterized by extremely high river discharges. Freshwater fluxes establish e.g. capping effects similar to those in the Bay of Bengal and an ENSO-driven seesaw linking upwelling off Indonesia in the Indian Ocean to those in the equatorial Pacific Ocean. These are important aspects, which, to my understanding, should be considered in a review paper.

We thank the reviewer for these suggestions, the distinguishing characteristics of upwelling along the Sumatra and Java coasts have been described in Chapter 2.10, by revising this section.

---

## Author Response (AR2)

We thank the editor for accepting the manuscript.

We have noted that technical corrections are required in the manuscript, but no specific comments have been received.

The manuscript has been checked again and some minor typing errors have been corrected.